# MARS: Unleashing the Power of Variance Reduction for Training Large Models

**Huizhuo Yuan** [* 1 2]  **Yifeng Liu** [* 1]  **Shuang Wu** [3]  **Xun Zhou** [3]  **Quanquan Gu** [1 2]

## Abstract

Training deep neural networks—and more recently, large models demands efficient and scalable optimizers. Adaptive gradient algorithms like Adam, AdamW, and their variants have been central to this task. Despite the development of numerous variance reduction algorithms in the past decade aimed at accelerating stochastic optimization in both convex and nonconvex settings, variance reduction has not found widespread success in training deep neural networks or large language models. Consequently, it has remained a less favored approach in modern AI. In this paper, to unleash the power of variance reduction for efficient training of large models, we propose a unified optimization framework, MARS (**M**ake v**A**riance **R**eduction **S**hine), which reconciles preconditioned gradient methods with variance reduction via a scaled stochastic recursive momentum technique. Within our framework, we introduce three instances of MARS that leverage preconditioned gradient updates based on AdamW, Lion, and Shampoo, respectively. We also draw a connection between our algorithms and existing optimizers. Experimental results on training GPT-2 models indicate that MARS consistently outperforms AdamW by a large margin.

## 1 Introduction

Adaptive gradient methods such as Adam (Kingma & Ba, 2015) and AdamW (Loshchilov & Hutter, 2019) have become the predominant optimization algorithms in deep learning. With the surge of large language models, the majority of the renowned models, including GPT-2 (Radford et al., 2019), GPT-3 (Brown, 2020), PaLM (Chowdhery et al., 2023) and Llama 3 (Dubey et al., 2024) are

trained with adaptive gradient methods. Numerous efforts have been made to improve adaptive gradient methods from both first-order and second-order optimization perspectives. For example, You et al. (2019) introduced LAMB, a layerwise adaptation technique that boosts training efficiency for BERT (Devlin, 2018). Using symbolic search, Chen et al. (2023) developed Lion, achieving faster training and reduced memory usage. Liu et al. (2023) designed Sophia, leveraging stochastic diagonal Hessian estimators to accelerate training. Gupta et al. (2018) proposed Shampoo, which performs stochastic optimization over tensor spaces with preconditioning matrices for each dimension. Anil et al. (2020) further refined Shampoo to give a scalable, practical version. Recently, Vyas et al. (2024) showed that Shampoo is equivalent to Adafactor (Shazeer & Stern, 2018) in the eigenbasis of Shampoo's preconditioner, and introduced SOAP, which stabilizes Shampoo with Adam. Importantly, recent studies (Kaddour et al., 2024; Zhao et al., 2024) have shown that these optimizers perform on par with AdamW in LLM pretraining, yet do not outperform it. This suggests the ongoing challenge in developing adaptive gradient methods superior to Adam and AdamW for large-scale model training.

Since adaptive gradient methods face challenges of high stochastic gradient variance, and language model training inherently involves a high-variance optimization problem (McCandlish et al., 2018), it is natural to consider variance reduction techniques to address this challenge. There exists a large body of literature on variance reduction for stochastic optimization, such as SAG (Roux et al., 2012), SVRG (Johnson & Zhang, 2013), STORM (Cutkosky & Orabona, 2019), which can improve the convergence of stochastic optimization. However, variance reduction has not found widespread success in training deep neural networks or large language models. Defazio & Bottou (2019) discussed why variance reduction can be ineffective in deep learning due to factors such as data augmentation, batch normalization, and dropout, which disrupt the finite-sum structure required by variance reduction principles. Nevertheless, in training language models, data augmentation, batch normalization, and dropout are nowadays rarely used, which opens the door for applying variance reduction techniques in optimizing these models. This naturally leads to the following research question:

---
[*]Equal contribution [1]Department of Computer Science, University of California, Los Angeles, California, USA (This work was done during Yifeng's internship at ByteDance Seed) [2]ByteDance Seed, San Jose, California, USA [3]ByteDance Seed, Beijing, China. Correspondence to: Quanquan Gu <qgu@cs.ucla.edu>.

*Proceedings of the 42nd International Conference on Machine Learning*, Vancouver, Canada. PMLR 267, 2025. Copyright 2025 by the author(s).

*Can variance reduction technique be applied to improve the performance of training large models?*

In this paper, we answer the above question affirmatively by introducing a novel optimization framework called MARS (**M**ake v**A**riance **R**eduction **S**hine), which incorporates variance reduction into adaptive gradient methods. Notably, we introduce a scaling parameter into the stochastic recursive momentum (STORM) (Cutkosky & Orabona, 2019) to adjust the strength of variance reduction and define a new gradient estimator. This gradient estimator undergoes gradient clipping and is subsequently subjected to exponential averaging. When the variance reduction strength is set to 1, it recovers the vanilla STORM momentum. In addition, the second-order momentum update is defined by the reweighted intermediate variable. These together ensure optimization stability throughout the training process. We summarize our major contributions of this paper as follows:

- We propose a unified framework for preconditioned variance reduction, namely MARS. At its core, MARS comprises two major components: (1) a scaled stochastic recursive momentum, which provides a variance-reduced estimator of the full gradient for better gradient complexity; and (2) the preconditioned update, which approximates the second-order Newton's method for better per-iteration complexity. By combining preconditioned gradient methods with variance reduction, MARS achieves the best of both worlds, accelerating the search for critical points in optimization.

- The MARS framework is versatile, accommodating all existing full matrix or diagonal Hessian approximations. Under this framework, we utilize three distinct designs of the preconditioning matrix, resulting in three specific instances of our MARS framework: MARS-AdamW, MARS-Lion, and MARS-Shampoo. Each variant demonstrates compatibility with their corresponding preconditioning in AdamW, Lion, and Shampoo, showing that MARS can seamlessly integrate with and do variance reduction on these established methods.

- Empirically, we evaluated MARS on GPT-2 fine-tuning tasks using the OpenWebText dataset. It demonstrates superior performance on GPT-2 large: AdamW requires 50 billion tokens to reach a validation loss of 2.58, whereas MARS only requires 28 billion tokens, and it achieves a final validation loss of 2.51. Furthermore, on the downstream task Hellaswag, MARS improved accuracy to 44.64%, outperforming AdamW's 41.70% after training on 50 billion tokens. And the code is available at https://github.com/AGI-Arena/MARS.

**Notations** In this paper, we assume $\mathbf{x}_t$ denotes the parameter of the language model at step $t$ and $\boldsymbol{\xi}_1, ..., \boldsymbol{\xi}_T \in \Xi$ are a sequence of independent random variables which denote the training data for each step. For some objective function $f$ that is differentiable, we assume $\mathbb{E}[f(\mathbf{x}, \boldsymbol{\xi}_t)|\mathbf{x}] = F(\mathbf{x})$ for $\forall \mathbf{x}, \forall t$. In our algorithm, the training data of the current step $\boldsymbol{\xi}_t$ and previous step $\boldsymbol{\xi}_{t-1}$ are used for attaining different gradient for the same parameter $\mathbf{x}_t$, so we just explicitly indicate these variables for function $f$.

## 2. Preliminaries

In this section, we review the preliminaries of stochastic optimization, including standard stochastic gradient methods and variance reduction.

We consider minimizing an objective function $F(\cdot) : \mathbb{R}^d \rightarrow \mathbb{R}$ as follows:

$$\min_{\mathbf{x}} F(\mathbf{x}) = \mathbb{E}_{\boldsymbol{\xi} \sim \mathcal{D}}[f(\mathbf{x}, \boldsymbol{\xi})], \tag{2.1}$$

where $f(\mathbf{x}, \boldsymbol{\xi})$ is possibly nonconvex loss function, $\mathbf{x} \in \mathbb{R}^d$ is the optimization variable, $\boldsymbol{\xi}$ is a random vector (e.g., a training data point) drawn from an unknown data distribution $\mathcal{D}$. We assume the access to the first-order oracle, which returns an unbiased estimator of the gradient $\mathbb{E}[\nabla f(\mathbf{x}, \boldsymbol{\xi})] = \nabla F(\mathbf{x})$. The standard stochastic gradient descent (SGD) algorithm yields:

$$\mathbf{x}_{t+1} = \mathbf{x}_t - \eta_t \nabla f(\mathbf{x}_t, \boldsymbol{\xi}_t), \tag{2.2}$$

where $\eta_t > 0$ is the learning rate or step size. SGD needs $\mathcal{O}(\varepsilon^{-4})$ stochastic gradient evaluations (i.e., gradient complexity or incremental first-order oracle complexity) to find a $\epsilon$-approximate first-order stationary points, i.e., $\|\nabla F(\mathbf{x})\|_2 \leq \epsilon$ (Ghadimi & Lan, 2013).

To accelerate the convergence of SGD, variance reduction techniques have been extensively researched in both the machine learning and optimization communities over the past decade, resulting in numerous algorithms for convex optimization—such as SAG (Roux et al., 2012), SVRG (Johnson & Zhang, 2013), SAGA (Defazio et al., 2014), and SARAH (Nguyen et al., 2017a)—as well as for nonconvex optimization, including SVRG (Allen-Zhu & Yuan, 2016; Reddi et al., 2016), SNVRG (Zhou et al., 2020), SPIDER (Fang et al., 2018), and STORM (Cutkosky & Orabona, 2019), among others. Notably, for nonconvex optimization, SNVRG (Zhou et al., 2020), SPIDER (Fang et al., 2018) and STORM (Cutkosky & Orabona, 2019) can improve the gradient complexity of SGD from $\mathcal{O}(\varepsilon^{-4})$ to $\mathcal{O}(\varepsilon^{-3})$, demonstrating a provable advantage.

At the heart of variance reduction techniques is a variance-reduced stochastic gradient, exemplified by the method proposed by Johnson & Zhang (2013) as follows:

$$\mathbf{m}_t = \nabla f(\mathbf{x}_t, \boldsymbol{\xi}_t) - \nabla f(\widetilde{\mathbf{x}}, \boldsymbol{\xi}_t) + \nabla F(\widetilde{\mathbf{x}}),$$

where $\widetilde{\mathbf{x}}$ is an anchoring point (a.k.a., reference point) that updates periodically. This variance-reduced stochastic gradient can reduce the variance of the stochastic gradient by

adding a correction term $-\nabla f(\widetilde{\mathbf{x}}, \boldsymbol{\xi}_t) + \nabla F(\widetilde{\mathbf{x}})$ based on a less frequently updated reference point $\widetilde{\mathbf{x}}$ and its full gradient $\nabla F(\widetilde{\mathbf{x}})$. It can be shown that the variance $\mathbf{m}_t$ can be controlled by $\|\mathbf{x}_t - \widetilde{\mathbf{x}}\|_2$, which will diminish as both $\mathbf{x}_t$ and $\widetilde{\mathbf{x}}$ converges to the stationary points when the algorithm makes progress. Subsequent improvements in variance reduction techniques were introduced in SARAH (Nguyen et al., 2017a) and SPIDER (Fang et al., 2018), which get rid of the anchor point and result in the following momentum update:

$$\mathbf{m}_t = \nabla f(\mathbf{x}_t, \boldsymbol{\xi}_t) - \nabla f(\mathbf{x}_{t-1}, \boldsymbol{\xi}_t) + \mathbf{m}_{t-1},$$
$$\mathbf{x}_{t+1} = \mathbf{x}_t - \eta_t \mathbf{m}_t. \tag{2.3}$$

In the context of training neural networks, $\mathbf{x}_t \in \mathbb{R}^d$ represents the trained weights in the neural network, $\boldsymbol{\xi}_t$ represents random data, and $\mathbf{m}_t$ is the variance-reduced (VR) first-order momentum. The stochastic gradient difference term $\nabla f(\mathbf{x}_t, \boldsymbol{\xi}_t) - \nabla f(\mathbf{x}_{t-1}, \boldsymbol{\xi}_t)$ cancels out common noise brought by $\boldsymbol{\xi}_t$, while pushing the gradient estimation from the estimator of $\nabla F(\mathbf{x}_{t-1})$ to the estimator of $\nabla F(\mathbf{x}_t)$. However, $\mathbf{m}_t$ needs to be reset periodically to a full gradient (or a large batch stochastic gradient) $\nabla F(\mathbf{x}_t)$, which we refer to as an anchoring step, analogous to the anchor point in SVRG.

Subsequently, Cutkosky & Orabona (2019) introduced Stochastic Recursive Momentum (STORM), a variant of standard momentum with an additional term, achieving the same convergence rate as SPIDER while eliminating the need for periodic anchoring:

$$\mathbf{m}_t = \beta_1 \mathbf{m}_{t-1} + (1 - \beta_1) \nabla f(\mathbf{x}_t, \boldsymbol{\xi}_t)$$
$$+ \beta_1 \big( \nabla f(\mathbf{x}_t, \boldsymbol{\xi}_t) - \nabla f(\mathbf{x}_{t-1}, \boldsymbol{\xi}_t) \big) \tag{2.4}$$

where $\beta_1 > 0$ is momentum parameter, and $\beta_1(\nabla f(\mathbf{x}_t, \boldsymbol{\xi}_t) - \nabla f(\mathbf{x}_{t-1}, \boldsymbol{\xi}_t))$ is the additional term that has variance reduction effect. Note that if $\mathbf{x}_t \approx \mathbf{x}_{t-1}$, STORM becomes approximately the standard momentum. Alternatively, (2.4) can be rewritten as an exponential moving average (EMA) of the first order momentum from previous step/iteration and the stochastic gradient with a *gradient correction* term:

$$\mathbf{m}_t = \beta_1 \mathbf{m}_{t-1} + (1 - \beta_1) \Big[ \nabla f(\mathbf{x}_t, \boldsymbol{\xi}_t)$$
$$+ \underbrace{\frac{\beta_1}{1 - \beta_1} \big( \nabla f(\mathbf{x}_t, \boldsymbol{\xi}_t) - \nabla f(\mathbf{x}_{t-1}, \boldsymbol{\xi}_t) \big)}_{\text{gradient correction}} \Big]. \tag{2.5}$$

Theoretically, when assuming access to an unbiased stochastic first-order oracle to the objective function $F(\mathbf{x})$, STORM achieves the nearly optimal gradient complexity of $\mathcal{O}(\varepsilon^{-3})$ for non-convex and smooth optimization problems (Arjevani et al., 2023).

# 3 Method

In this section, we introduce MARS (**M**ake v**A**riance **R**eduction **S**hine), a family of preconditioned optimization algorithms that perform variance reduction in gradient estimation.

## 3.1 MARS Framework

We first introduce our framework for a preconditioned, variance-reduced stochastic optimization, which unifies both first-order (e.g., AdamW, Lion) and second-order (e.g., Shampoo) adaptive gradient methods.

**Preconditioned Variance Reduction.** Variance reduction methods achieve faster convergence than SGD, yet identifying optimal learning rates remains a practical challenge. Particularly, different parameters often exhibit varying curvatures, requiring tailored learning rates for each. One approach to addressing this issue is to use the Hessian matrix to precondition gradient updates, integrating curvature information into the updates. The idea stems from minimizing the second-order Taylor expansion at $\mathbf{x}_t$:

$$F(\mathbf{x}_{t+1}) \approx F(\mathbf{x}_t) + \nabla F(\mathbf{x}_t)(\mathbf{x}_{t+1} - \mathbf{x}_t)$$
$$+ \frac{1}{2}(\mathbf{x}_{t+1} - \mathbf{x}_t)^{\top} \nabla^2 F(\mathbf{x}_t)(\mathbf{x}_{t+1} - \mathbf{x}_t), \tag{3.1}$$

resulting in the update formula $\mathbf{x}_{t+1} = \mathbf{x}_t - \mathbf{H}_t^{-1} \nabla F(\mathbf{x}_t)$, where $\mathbf{H}_t := \nabla^2 F(\mathbf{x}_t) \in \mathbb{R}^{d \times d}$ is the Hessian matrix. In our paper, we encapsulate the preconditioned gradient $\mathbf{H}_t^{-1} \nabla F(\mathbf{x}_t)$ update within a more generalized framework of Online Mirror Descent (OMD) as in Gupta et al. (2018), leading to the following update rules:

$$\mathbf{x}_{t+1} = \arg \min_{\mathbf{x} \in \mathbb{R}^d} \left\{ \eta_t \langle \mathbf{m}_t, \mathbf{x} \rangle + \frac{1}{2} \|\mathbf{x} - \mathbf{x}_t\|_{\mathbf{H}_t}^2 \right\}, \tag{3.2}$$

where $\eta_t > 0$ can be viewed as a base learning rate. Combining (3.2) with the STORM momentum, we obtain the following preconditioned variance-reduced update:

$$\mathbf{m}_t = \beta_1 \mathbf{m}_{t-1} + (1 - \beta_1) \Big[ \nabla f(\mathbf{x}_t, \boldsymbol{\xi}_t)$$
$$+ \frac{\beta_1}{1 - \beta_1} \big( \nabla f(\mathbf{x}_t, \boldsymbol{\xi}_t) - \nabla f(\mathbf{x}_{t-1}, \boldsymbol{\xi}_t) \big) \Big], \tag{3.3}$$

$$\mathbf{x}_{t+1} = \arg \min_{\mathbf{x} \in \mathbb{R}^d} \left\{ \eta_t \langle \mathbf{m}_t, \mathbf{x} \rangle + \frac{1}{2} \|\mathbf{x} - \mathbf{x}_t\|_{\mathbf{H}_t}^2 \right\}. \tag{3.4}$$

**Remark 3.1.** SuperAdam (Huang et al., 2021) also incorporates the STORM into the design of adaptive gradient methods. However, their precondition matrix can be viewed as a special case of our general framework. SuperAdam's design focuses on diagonal precondition matrix and draws heavily from the design used in Adam (Kingma & Ba, 2015), AdaGrad-Norm (Ward et al., 2020), and AdaBelief (Zhuang et al., 2020). Furthermore, their preconditioner matrix is designed following Adam's structure but does not account

for the revised definition of variance-reduced momentum, resulting in a significant mismatch between the first-order and second-order momentum. We will further clarify these differences when discussing specific instances of our framework.

**Algorithm Design.** In practice, alongside our preconditioned variance-reduced update (3.3), we introduce a scaling parameter $\gamma_t$ to control the scale of gradient correction in variance reduction. We also introduce a new gradient estimator $\mathbf{c}_t$, which is the combination of stochastic gradient and the scaled gradient correction term:

$$\mathbf{c}_t = \nabla f(\mathbf{x}_t, \boldsymbol{\xi}_t) + \underbrace{\gamma_t \frac{\beta_1}{1-\beta_1} \left( \nabla f(\mathbf{x}_t, \boldsymbol{\xi}_t) - \nabla f(\mathbf{x}_{t-1}, \boldsymbol{\xi}_t) \right)}_{\text{scaled gradient correction}}.$$

When $\gamma_t = 1$, the above reduces to the second term of (3.3). On the other hand, when $\gamma_t = 0$, it reduces to the stochastic gradient. Thus, $\mathbf{c}_t$ can be seen a gradient estimator with adjustable variance control.

Following standard techniques in deep learning practice, we also perform gradient clipping on $\mathbf{c}_t$, which is calculated by:

$$\widetilde{\mathbf{c}}_t = \text{Clip}(\mathbf{c}_t, 1) = \begin{cases} \frac{\mathbf{c}_t}{\|\mathbf{c}_t\|_2} & \text{if } \|\mathbf{c}_t\|_2 > 1, \\ \mathbf{c}_t & \text{otherwise.} \end{cases} \quad (3.5)$$

We note that the Second-order Clipped Stochastic Optimization (Sophia) algorithm (Liu et al., 2023) also incorporates clipping in their algorithm design. However, their approach does clipping upon the preconditioned gradient with clipping-by-value, while our method applies clipping to the intermediate gradient estimate using the more standard technique of clipping-by-norm. After the gradient clipping, the VR momentum $\mathbf{m}_t$ can be calculated as the EMA of $\widetilde{\mathbf{c}}_t$. The resulting MARS algorithm is summarized in Algorithm 1.

---

**Algorithm 1** MARS

1: **input:** $\mathbf{x}_0, \beta_1, \{\gamma_t\}, \{\eta_t\}$
2: Set $\mathbf{m}_0 \leftarrow \mathbf{0}$ and $\mathbf{x}_1 \leftarrow \mathbf{x}_0$
3: **for** $t = 1$, **to** $n$ **do**
4:     Sample $\boldsymbol{\xi}_t$ and let $\mathbf{c}_t = \nabla f(\mathbf{x}_t, \boldsymbol{\xi}_t) + \gamma_t \frac{\beta_1}{1-\beta_1} \left( \nabla f(\mathbf{x}_t, \boldsymbol{\xi}_t) - \nabla f(\mathbf{x}_{t-1}, \boldsymbol{\xi}_t) \right)$
5:     if $\|\mathbf{c}_t\|_2 > 1$, then $\widetilde{\mathbf{c}}_t = \frac{\mathbf{c}_t}{\|\mathbf{c}_t\|_2}$ else $\widetilde{\mathbf{c}}_t = \mathbf{c}_t$
6:     $\mathbf{m}_t = \beta_1 \mathbf{m}_{t-1} + (1-\beta_1)\widetilde{\mathbf{c}}_t$
7:     $\mathbf{x}_{t+1} = \arg\min_{\mathbf{x}} \left\{ \eta_t \langle \mathbf{m}_t, \mathbf{x} \rangle + \frac{1}{2}\|\mathbf{x} - \mathbf{x}_t\|_{\mathbf{H}_t}^2 \right\}$
8: **end for**

---

**Why $\gamma_t$ improves convergence** A similar idea of adjusting the strength of variance reduction has been proposed by Yin et al. (2023) in the context of SVRG. This approach originates from a classical line of work on control variates (Asmussen & Glynn, 2007; Lavenberg et al., 1977). In the standard control variates setting, one considers the

estimator:

$$\mathbb{E}\left[ X - \mathbb{E}[X] - \gamma(Y - \mathbb{E}[Y]) \right]^2$$
$$= \text{Var}(X) - 2\gamma\,\mathbb{E}[(X - \mathbb{E}[X])(Y - \mathbb{E}[Y])] + \gamma^2\,\text{Var}(Y),$$

which admits an optimal choice of $\gamma$ that minimizes the variance:

$$\arg\min_{\gamma} \mathbb{E}\left[ X - \mathbb{E}[X] - \gamma(Y - \mathbb{E}[Y]) \right]^2$$
$$= \frac{\mathbb{E}[(X - \mathbb{E}[X])(Y - \mathbb{E}[Y])]}{\text{Var}(Y)}.$$

In the context of STORM updates used in our work, the update rule includes both stochastic gradients and recursive momentum terms. Let us define:

$$X = (1 - \beta)\nabla f(\mathbf{x}_{t+1}, \boldsymbol{\xi}_{t+1})$$
$$Y = \beta\left[ \nabla f(\mathbf{x}_{t+1}, \boldsymbol{\xi}_{t+1}) - \nabla f(\mathbf{x}_t, \boldsymbol{\xi}_{t+1}) \right]$$
$$Z_t = \mathbf{m}_t - \nabla F(\mathbf{x}_t)$$

and let $U := X - \mathbb{E}[X] + Z_t$. The updated $Z_{t+1}$ at step $t+1$ is of the form $U + (\gamma Y - \mathbb{E}[Y])$, whose squared expectation we aim to minimize.

The optimal choice of $\gamma$ in this setting is:

$$\gamma^* = 1 - \frac{\mathbb{E}[UY] + \text{Var}(Y)}{\mathbb{E}[Y^2]}.$$

With this choice, we obtain a variance reduction:

$$\mathbb{E}\left[ U + (\gamma^* Y - \mathbb{E}[Y]) \right]^2 = \mathbb{E}\left[ U + (Y - \mathbb{E}[Y]) \right]^2$$
$$- \frac{(\mathbb{E}[UY] + \text{Var}(Y))^2}{\mathbb{E}[Y^2]},$$

which is strictly smaller than the variance under any non-optimal choice of $\gamma$. Thus, dynamically tuning $\gamma_t$ improves the variance of the gradient estimator, leading to better convergence behavior. A full convergence analysis is provided in Appendix B.2.

We provide the convergence analysis of Algorithm 1 in Theorem B.5 in Appendix B.2. We prove that under standard assumptions, MARS achieves a superior convergence rate of $\mathcal{O}(T^{-1/3})$, outperforming the $\mathcal{O}(T^{-1/4})$ rate attainable by AdamW.

**Full Matrix Approximation.** In practice, calculating the Hessian matrix is computationally expensive or even intractable due to the complexity of second-order differentiation and the significant memory cost of storing $\mathbf{H}_t$, especially when the parameters in a neural network constitute a high-dimensional matrix. Many existing algorithms employ various approximations of the Hessian. For instance, K-FAC (Martens & Grosse, 2015) and Shampoo (Gupta et al., 2018) approximate the Gauss-Newton component of the Hessian (also known as the Fisher information matrix),

using a layerwise Kronecker product approximation (Morwani et al., 2024). Additionally, Sophia (Liu et al., 2023) suggests using Hutchinson's estimator or the Gauss-Newton-Barlett estimator for approximating the Hessian. We take various designs of the preconditioning matrix into account and broaden the definition of $\mathbf{H}_t$ in (3.4) to encompass various specifically designed preconditioning matrix in the rest of the paper.

**Diagonal Matrix Approximation.** Even when using approximated Hessian matrices, second-order algorithms mentioned above remain more computationally intensive compared to first-order gradient updates. Thus, another line of research focuses on approximating the Hessian matrix through diagonal matrices, as seen in optimization algorithms like AdaGrad (Duchi et al., 2011), RMSProp (Tieleman, 2012), AdaDelta (Zeiler, 2012), Adam (Kingma & Ba, 2015) and AdamW (Loshchilov & Hutter, 2019), etc. This approach to diagonal preconditioning effectively transforms the updates into a first-order method, assigning adaptive learning rates to each gradient coordinate. For example, in AdaGrad (Duchi et al., 2011), the preconditioned matrix is defined by:

$$[\mathbf{H}_t]_{ii} = \sqrt{\sum_{\tau=0}^{t} \left[\nabla f(\mathbf{x}_\tau, \boldsymbol{\xi}_\tau)\right]_i^2}.$$

On the other hand, Adam can be seen as using a diagonal $\mathbf{H}_t$, where each diagonal element is the EMA of $[\nabla f(\mathbf{x}_t, \boldsymbol{\xi}_t)]_i^2$:

$$[\mathbf{H}_t]_{ii} = \beta[\mathbf{H}_{t-1}]_{ii} + (1-\beta)[\nabla f(\mathbf{x}_t, \boldsymbol{\xi}_t)]_i^2. \quad (3.6)$$

Therefore, the update simplifies to elementwise adaptive gradient update, i.e., $[\mathbf{x}_{t+1}]_i = [\mathbf{x}_t]_i - \eta[\mathbf{m}_t]_i/[\mathbf{H}_t]_{ii}$. Our unified framework accommodates both types of preconditioning: full Hessian approximation and diagonal Hessian approximation. Different definitions of $\mathbf{H}_t$ give rise to different algorithms.

Notably, full-matrix approximations of the Hessian are potentially more powerful than diagonal approximations, as they can capture statistical correlations between the gradients of different parameters. Geometrically, full-matrix approximations allow both scaling and rotation of gradients, whereas diagonal matrices are limited to scaling alone.

### 3.2 Instantiation of MARS

In previous subsection, we introduced our preconditioned variance reduction framework in Algorithm 1 and discussed various approaches for approximating the Hessian matrix. In this subsection, we introduce practical designs of MARS under different choices of $\mathbf{H}_t$. While here we only present three instantiations: MARS-AdamW, MARS-Lion, and MARS-Shampoo, we believe there are many other instances of MARS can be derived similarly.

### 3.2.1 MARS-ADAMW

The first instance of MARS is built up on the idea of Adam/AdamW (Loshchilov & Hutter, 2019). To automatically adjust the learning rate and accelerate convergence, Adam (Kingma & Ba, 2015) adopts the adaptive preconditioned gradient in (3.6) together with a bias correction and $\ell_2$ regularization. AdamW (Loshchilov & Hutter, 2019) further changes the $\ell_2$ regularization to a decoupled weight decay. Overall, the full AdamW updates can be summarized as follows:

$$\mathbf{m}_t = \beta_1 \mathbf{m}_{t-1} + (1-\beta_1)\nabla f(\mathbf{x}_t, \boldsymbol{\xi}_t), \quad (3.7)$$

$$\mathbf{v}_t = \beta_2 \mathbf{v}_{t-1} + (1-\beta_2)\big(\nabla f(\mathbf{x}_t, \boldsymbol{\xi}_t)\big)^2, \quad (3.8)$$

$$\widehat{\mathbf{m}}_t = \frac{\mathbf{m}_t}{1-\beta_1^t}, \quad \widehat{\mathbf{v}}_t = \frac{\mathbf{v}_t}{1-\beta_2^t}, \quad (3.9)$$

$$\mathbf{x}_{t+1} = \mathbf{x}_t - \eta_t\left(\frac{\widehat{\mathbf{m}}_t}{\sqrt{\widehat{\mathbf{v}}_t}+\epsilon} + \lambda\mathbf{x}_t\right). \quad (3.10)$$

We see that except for the small $\epsilon$ introduced for computational stability, and the decoupled weight decay $\lambda\mathbf{x}_t$, AdamW can be seen as a step of mirror descent update (3.2) with $\mathbf{m}_t$ defined in (3.7), $\mathbf{v}_t$ defined in (3.8), and $\mathbf{H}_t$ defined by

$$\mathbf{H}_t := \sqrt{\text{diag}\big(\mathbf{v}_t\big)} \cdot \frac{1-\beta_1^t}{\sqrt{1-\beta_2^t}}. \quad (3.11)$$

In MARS-AdamW, we implement the preconditioned variance-reduced update as in (3.4), and utilize the same definitions for $\mathbf{H}_t$, $\epsilon$, and weight decay as those specified in AdamW. For $\mathbf{v}_t$, different from the EMA of squared gradients $\mathbf{m}_t^2$ in AdamW, we redefine it to fit our variance-reduced stochastic gradient. Specifically, we denote the summation of the stochastic gradient and the scaled gradient correction term by $\mathbf{c}_t$ and define $\mathbf{v}_t$ as the EMA of $\mathbf{c}_t^2$ as follows:

$$\mathbf{c}_t := \nabla f(\mathbf{x}_t, \boldsymbol{\xi}_t)$$
$$+ \gamma_t \frac{\beta_1}{1-\beta_1}\big(\nabla f(\mathbf{x}_t, \boldsymbol{\xi}_t) - \nabla f(\mathbf{x}_{t-1}, \boldsymbol{\xi}_t)\big), \quad (3.12)$$

$$\mathbf{m}_t = \beta_1 \mathbf{m}_{t-1} + (1-\beta_1)\mathbf{c}_t, \quad (3.13)$$

$$\mathbf{v}_t = \beta_2 \mathbf{v}_{t-1} + (1-\beta_2)\mathbf{c}_t^2. \quad (3.14)$$

Here, $\gamma_t$ is a scaling parameter that controls the strength of gradient correction. When $\gamma_t = 0$, the algorithm reduces to AdamW. Conversely, when $\gamma_t = 1$, (3.13) aligns with the STORM momentum. Combining (3.12), (3.13), (3.14) together with (3.11) and the mirror descent update (3.2), we derive the MARS-AdamW algorithm in Algorithm 2. In practice, $\gamma_t$ is often set between 0 and 1. Moreover, we employ gradient clipping-by-norm to $\mathbf{c}_t$ at Line 5, following the standard gradient clipping technique performed in neural network training. We provide a convergence analysis of Algorithm 2 in Theorem B.6 in Appendix B.2.

**Remark 3.2.** Compared with SuperAdam (Huang et al., 2021), one key difference is that our algorithm defines the second-order momentum $\mathbf{v}_t$ as the exponential moving average of the square norm of $\mathbf{c}_t$ rather than the square norm of the stochastic gradient. This new definition of second-order momentum is crucial for accommodating the right scale of updates on a coordinate-wise basis. Moreover, as we mentioned in Algorithm 1, we introduce a scaling parameter $\gamma_t$ and implement gradient clipping on $\mathbf{c}_t$. In Section 4, we will demonstrate empirically that the changes contribute to effective performance in large language model training. Finally, our algorithm utilizes bias correction and weight decay while SuperAdam does not.

**Remark 3.3.** Careful readers might have noticed that in each iteration of our algorithm, we need to calculate the stochastic gradient twice for different data batches $\boldsymbol{\xi}_{t-1}$ and $\boldsymbol{\xi}_t$ with the same parameters. In order to overcome this problem, we propose to use $\big(\nabla f(\mathbf{x}_t, \boldsymbol{\xi}_t) - \nabla f(\mathbf{x}_{t-1}, \boldsymbol{\xi}_t)\big)$ to approximate $\big(\nabla f(\mathbf{x}_t, \boldsymbol{\xi}_t) - \nabla f(\mathbf{x}_{t-1}, \boldsymbol{\xi}_t)\big)$ in (3.12) and $\mathbf{c}_t$ will be approximated by:

$$\mathbf{c}_t \approx \nabla f(\mathbf{x}_t, \boldsymbol{\xi}_t) + \gamma_t \frac{\beta_1}{1-\beta_1}\big(\nabla f(\mathbf{x}_t, \boldsymbol{\xi}_t) - \nabla f(\mathbf{x}_{t-1}, \boldsymbol{\xi}_{t-1})\big).$$

To avoid confusion, we refer to the approximate version as MARS-approx. While MARS and MARS-approx differ in their updates and may theoretically exhibit distinct convergence guarantees, our experiments show that MARS provides only marginal improvements over MARS-approx in practice. Thus, we recommend using MARS-approx for practical applications.

**Connection between MARS-AdamW and Adan.** Adan (Xie et al., 2024) is another adaptive gradient method improved upon Adam with reformulated Nesterov's accelerated SGD (See Lemma 1 in Xie et al. (2024) for more details). The Adan algorithm takes the following momentum updates:

$$\mathbf{y}_t = \beta_1 \mathbf{y}_{t-1} + (1-\beta_1)\nabla f(\mathbf{x}_t, \boldsymbol{\xi}_t),$$
$$\mathbf{z}_t = \beta_2 \mathbf{z}_{t-1} + (1-\beta_2)\big(\nabla f(\mathbf{x}_t, \boldsymbol{\xi}_t) - \nabla f(\mathbf{x}_{t-1}, \boldsymbol{\xi}_{t-1})\big),$$
$$\mathbf{m}_t := \mathbf{y}_t + \beta_2 \mathbf{z}_t.$$

When $\beta_2 = \beta_1$, this reduces to

$$\mathbf{m}_t = \beta_1 \mathbf{m}_{t-1} + (1-\beta_1)\big[\nabla f(\mathbf{x}_t, \boldsymbol{\xi}_t) \\ + \beta_1\big(\nabla f(\mathbf{x}_t, \boldsymbol{\xi}_t) - \nabla f(\mathbf{x}_{t-1}, \boldsymbol{\xi}_{t-1})\big)\big],$$

which is a special case of MARS-approx's momentum with $\gamma_t = 1 - \beta_1$. It is worth noting that although motivated by the Nesterov's momentum, Adan's momentum updates cannot recover Nesterov's momentum unless $\beta_1 = \beta_2$.

### 3.2.2 MARS-LION

Using symbolic program search, Chen et al. (2023) introduced a simpler algorithm Lion compared to AdamW,

---

**Algorithm 2** MARS-AdamW

1: **input:** $\mathbf{x}_0, \lambda, \beta_1, \beta_2, \{\gamma_t\}, \{\eta_t\}$
2: Set $\mathbf{m}_0 \leftarrow \mathbf{0}, \mathbf{v}_0 \leftarrow \mathbf{0}$ and $\mathbf{x}_1 \leftarrow \mathbf{x}_0$
3: **for** $t = 1$, **to** $n$ **do**
4:     Sample $\boldsymbol{\xi}_t$ and let $\mathbf{c}_t = \nabla f(\mathbf{x}_t, \boldsymbol{\xi}_t) + \gamma_t \frac{\beta_1}{1-\beta_1}\big(\nabla f(\mathbf{x}_t, \boldsymbol{\xi}_t) - \nabla f(\mathbf{x}_{t-1}, \boldsymbol{\xi}_t)\big)$
5:     if $\|\mathbf{c}_t\|_2 > 1$, then $\widetilde{\mathbf{c}}_t = \frac{\mathbf{c}_t}{\|\mathbf{c}_t\|_2}$ else $\widetilde{\mathbf{c}}_t = \mathbf{c}_t$
6:     $\mathbf{m}_t = \beta_1 \mathbf{m}_{t-1} + (1-\beta_1)\widetilde{\mathbf{c}}_t$
7:     $\mathbf{v}_t = \beta_2 \mathbf{v}_{t-1} + (1-\beta_2)\widetilde{\mathbf{c}}_t^2$
8:     $\widehat{\mathbf{m}}_t = \frac{\mathbf{m}_t}{1-\beta_1^t}, \widehat{\mathbf{v}}_t = \frac{\mathbf{v}_t}{1-\beta_2^t}$
9:     $\mathbf{x}_{t+1} = \mathbf{x}_t - \eta_t\Big(\frac{\widehat{\mathbf{m}}_t}{\sqrt{\widehat{\mathbf{v}}_t}+\epsilon} + \lambda\mathbf{x}_t\Big)$
10: **end for**

---

which employs a sign operation to maintain uniform magnitude across all parameters. The updates for Lion are illustrated as follows:

$$\mathbf{m}_t = \beta_2 \mathbf{u}_{t-1} + (1-\beta_2)\nabla f(\mathbf{x}_t, \boldsymbol{\xi}_t), \quad (3.15)$$
$$\mathbf{u}_t = \beta_1 \mathbf{u}_{t-1} + (1-\beta_1)\nabla f(\mathbf{x}_t, \boldsymbol{\xi}_t), \quad (3.16)$$
$$\mathbf{x}_{t+1} = \mathbf{x}_t - \eta_t\big(\text{sign}(\mathbf{m}_t) + \lambda\mathbf{x}_t\big). \quad (3.17)$$

Instead of employing an EMA of gradient norms as in (3.8) and (3.11) of AdamW, the sign preconditioning mechanism in Lion utilizes

$$\mathbf{H}_t := \sqrt{\text{diag}(\mathbf{m}_t^2)}. \quad (3.18)$$

Following the same definition of $\mathbf{H}_t$ as in (3.18), we present MARS-Lion in Algorithm 3.

---

**Algorithm 3** MARS-Lion

1: **input:** $\mathbf{x}_0, \lambda, \beta_1, \{\gamma_t\}, \{\eta_t\}$
2: Set $\mathbf{m}_0 \leftarrow \mathbf{0}$ and $\mathbf{x}_1 \leftarrow \mathbf{x}_0$
3: **for** $t = 1$, **to** $n$ **do**
4:     Sample $\boldsymbol{\xi}_t$ and let $\mathbf{c}_t = \nabla f(\mathbf{x}_t, \boldsymbol{\xi}_t) + \gamma_t \frac{\beta_1}{1-\beta_1}\big(\nabla f(\mathbf{x}_t, \boldsymbol{\xi}_t) - \nabla f(\mathbf{x}_{t-1}, \boldsymbol{\xi}_t)\big)$
5:     if $\|\mathbf{c}_t\|_2 > 1$, then $\widetilde{\mathbf{c}}_t = \frac{\mathbf{c}_t}{\|\mathbf{c}_t\|_2}$ else $\widetilde{\mathbf{c}}_t = \mathbf{c}_t$
6:     $\mathbf{m}_t = \beta_1 \mathbf{m}_{t-1} + (1-\beta_1)\widetilde{\mathbf{c}}_t$
7:     $\mathbf{x}_{t+1} = \mathbf{x}_t - \eta_t\big(\text{sign}(\mathbf{m}_t) + \lambda\mathbf{x}_t\big)$
8: **end for**

---

Setting $\mathbf{m}_t = \nabla f(\mathbf{x}_t, \boldsymbol{\xi}_t), a_1 = \beta_1, a_2 = 1 - \beta_1, b_1 = \beta_2, b_2 = 1 - \beta_2$ in Lemma D.1, and shifting the index of $\mathbf{u}_t$ by taking $\mathbf{u}_t = \mathbf{u}_{t-1}, \mathbf{u}_{t-1} = \mathbf{u}_{t-2}$ in Lemma D.1, we can show that Lion momentum updates in (3.15) and (3.16) are equivalent to the following single momentum update:

$$\mathbf{m}_t = \beta_1 \mathbf{m}_{t-1} + (1-\beta_1)\Big[\nabla f(\mathbf{x}_t, \boldsymbol{\xi}_t) \quad (3.19)$$
$$+ \frac{\beta_1(1-\beta_2)}{1-\beta_1}\big(\nabla f(\mathbf{x}_t, \boldsymbol{\xi}_t) - \nabla f(\mathbf{x}_{t-1}, \boldsymbol{\xi}_{t-1})\big)\Big].$$
$$(3.20)$$

On the other hand, setting $\gamma_t = 1 - \beta_2$ in the core updates

of MARS (3.12) and (3.13), we obtain the $\mathbf{m}_t$ update:

$$\mathbf{m}_t = \beta_1 \mathbf{m}_{t-1} + (1-\beta_1)\Big[\nabla f(\mathbf{x}_t, \boldsymbol{\xi}_t) \qquad (3.21)$$

$$+ \frac{\beta_1(1-\beta_2)}{1-\beta_1}\big(\nabla f(\mathbf{x}_t, \boldsymbol{\xi}_t) - \nabla f(\mathbf{x}_{t-1}, \boldsymbol{\xi}_t)\big)\Big]. \qquad (3.22)$$

The only difference between (3.26) and (3.28) lies in the stochasticity used, specifically, $\boldsymbol{\xi}_t$ versus $\boldsymbol{\xi}_{t-1}$ when calculating $\nabla f(\boldsymbol{x}_{t-1}, \cdot)$. Therefore, ignoring the gradient clipping at Line 3, we can see Lion as a special case of MARS-Lion when $\gamma_t = 1 - \beta_2$, and using approximate gradient calculation on $\nabla f(\mathbf{x}_{t-1}, \boldsymbol{\xi}_t)$. In practice, we observe little difference between using $f(\mathbf{x}_{t-1}, \boldsymbol{\xi}_{t-1})$ derived from the STORM momentum and its approximation $f(\mathbf{x}_{t-1}, \boldsymbol{\xi}_t)$.
**Connection between MARS-Lion and Lion.** Lion turns out to be a special case of MARS-Lion. The momentum updates in Lion can be seen as an approximate implementation of our updates. To facilitate this claim, we present a lemma that follows directly from straightforward arithmetic calculations.

**Lemma 3.4.** For any sequence $\{\mathbf{g}_t \in \mathbb{R}^d\}_{t=0,1,\ldots}$, consider the following updates of $\mathbf{m}_t$ for any constant factors $a_1, a_2, b_1,$ and $b_2$:

$$\mathbf{u}_t = a_1 \mathbf{u}_{t-1} + a_2 \mathbf{g}_t, \qquad (3.23)$$
$$\mathbf{m}_t = b_1 \mathbf{u}_t + b_2 \mathbf{g}_t. \qquad (3.24)$$

The updates are equivalent to

$$\mathbf{m}_t = a_1 \mathbf{m}_{t-1} + (b_1 a_2 - a_1 b_2 + b_2)\mathbf{g}_t + a_1 b_2(\mathbf{g}_t - \mathbf{g}_{t-1}).$$

Setting $\mathbf{g}_t = \nabla f(\mathbf{x}_t, \boldsymbol{\xi}_t)$, $a_1 = \beta_1$, $a_2 = 1 - \beta_1$, $b_1 = \beta_2$, $b_2 = 1 - \beta_2$ in Lemma D.1, and shifting the index of $\mathbf{u}_t$ by taking $\mathbf{u}_t = \mathbf{u}_{t-1}$, $\mathbf{u}_{t-1} = \mathbf{u}_{t-2}$ in Lemma D.1, we can show that Lion momentum updates in (3.15) and (3.16) are equivalent to the following single momentum update:

$$\mathbf{m}_t = \beta_1 \mathbf{m}_{t-1} + (1-\beta_1)\Big[\nabla f(\mathbf{x}_t, \boldsymbol{\xi}_t) \qquad (3.25)$$

$$+ \frac{\beta_1(1-\beta_2)}{1-\beta_1}\big(\nabla f(\mathbf{x}_t, \boldsymbol{\xi}_t) - \nabla f(\mathbf{x}_{t-1}, \boldsymbol{\xi}_{t-1})\big)\Big]. \qquad (3.26)$$

On the other hand, setting $\gamma_t = 1 - \beta_2$ in the core updates of MARS (3.12) and (3.13), we obtain the $\mathbf{m}_t$ update:

$$\mathbf{m}_t = \beta_1 \mathbf{m}_{t-1} + (1-\beta_1)\Big[\nabla f(\mathbf{x}_t, \boldsymbol{\xi}_t) \qquad (3.27)$$

$$+ \frac{\beta_1(1-\beta_2)}{1-\beta_1}\big(\nabla f(\mathbf{x}_t, \boldsymbol{\xi}_t) - \nabla f(\mathbf{x}_{t-1}, \boldsymbol{\xi}_t)\big)\Big]. \qquad (3.28)$$

The only difference between (3.26) and (3.28) lies in the stochasticity used, specifically, $\boldsymbol{\xi}_t$ versus $\boldsymbol{\xi}_{t-1}$ when calculating $\nabla f(\boldsymbol{x}_{t-1}, \cdot)$. Therefore, ignoring the gradient

clipping at Line 3, we can see Lion as a special case of MARS-Lion when $\gamma_t = 1 - \beta_2$, and using approximate gradient calculation on $\nabla f(\mathbf{x}_{t-1}, \boldsymbol{\xi}_t)$. In practice, we observe little difference between using $f(\mathbf{x}_{t-1}, \boldsymbol{\xi}_{t-1})$ derived from the STORM momentum and its approximation $f(\mathbf{x}_{t-1}, \boldsymbol{\xi}_t)$.

### 3.2.3 MARS-SHAMPOO

Shampoo (Gupta et al., 2018) introduces a preconditioning approach that operates on the eigenspace of matrices. Given the gradient matrix $\mathbf{m}_t := \nabla f_t(\mathbf{x}_t, \boldsymbol{\xi}_t) \in \mathbb{R}^{m \times n}$, the update rules of Shampoo are displayed as follows:

$$\mathbf{L}_t = \mathbf{L}_{t-1} + \mathbf{m}_t \mathbf{m}_t^\top,$$
$$\mathbf{R}_t = \mathbf{R}_{t-1} + \mathbf{m}_t^\top \mathbf{m}_t,$$
$$\mathbf{x}_{t+1} = \mathbf{x}_t - \eta_t \mathbf{L}_t^{-1/4} \mathbf{m}_t \mathbf{R}_t^{-1/4}, \qquad (3.29)$$

where $\mathbf{x}_t \in \mathbb{R}^{m \times n}$ (slightly abusing notation) represents the corresponding weight matrix. It has been shown that the two-sided preconditioning in (3.29) is equivalent to preconditioning on the flattened vector $\mathbf{m}_t := \text{vec}(\mathbf{m}_t)$ with a Kronecker product (Gupta et al., 2018; Morwani et al., 2024)

$$\mathbf{H}_t := \Big(\sum_{\tau=1}^t \mathbf{G}_\tau \mathbf{G}_\tau^\top\Big)^{1/4} \otimes \Big(\sum_{\tau=1}^t \mathbf{G}_\tau^\top \mathbf{G}_\tau\Big)^{1/4}.$$

In practice, an exponential moving average (EMA) is often used in place of the direct summation. The update rule in (3.29) can be simplified to $\mathbf{x}_{t+1} = \mathbf{x}_t - \eta_t(\mathbf{m}_t \mathbf{m}_t^\top)^{-1/4}\mathbf{m}_t(\mathbf{m}_t^\top \mathbf{m}_t)^{-1/4}$. This is equivalent to performing preconditioning on the eigenspace of $\mathbf{m}_t$:

$$\mathbf{U}_t, \boldsymbol{\Sigma}_t, \mathbf{V}_t = \text{SVD}(\mathbf{m}_t),$$
$$\mathbf{x}_{t+1} = \mathbf{x}_t - \eta_t \mathbf{U}_t \mathbf{V}_t^\top. \qquad (3.30)$$

Therefore, we borrow the eigenspace preconditioning from Shampoo, and design our algorithm to precondition on any matrix-shaped update as in (3.30). In particular, we present our algorithm in Algorithm 4.

---
**Algorithm 4** MARS-Shampoo

1: **input:** $\mathbf{x}_0, \lambda, \beta_1, \{\gamma_t\}, \{\eta_t\}$
2: Set $\mathbf{m}_0 \leftarrow \mathbf{0}$ and $\mathbf{x}_1 \leftarrow \mathbf{x}_0$
3: **for** $t = 1,$ **to** $n$ **do**
4:     sample $\boldsymbol{\xi}_t$ and let $\mathbf{c}_t = \nabla f(\mathbf{x}_t, \boldsymbol{\xi}_t) + \gamma_t(\frac{\beta_1}{1-\beta_1})\big(\nabla f(\mathbf{x}_t, \boldsymbol{\xi}_t) - \nabla f(\mathbf{x}_{t-1}, \boldsymbol{\xi}_t)\big)$
5:     $\mathbf{m}_t = \beta_1 \mathbf{m}_{t-1} + (1-\beta_1)\mathbf{c}_t$
6:     $\mathbf{U}_t, \boldsymbol{\Sigma}_t, \mathbf{V}_t = \text{SVD}(\mathbf{m}_t)$
7:     $\mathbf{x}_{t+1} = \mathbf{x}_t - \eta_t(\mathbf{U}_t \mathbf{V}_t^\top + \lambda \mathbf{x}_t)$
8: **end for**
---

To reduce the time complexity of SVD decomposition, Bernstein & Newhouse (2024) summarized four different approaches for computing or approximating (3.30) including SVD, sketching (Martinsson & Tropp, 2020), Newton

iteration (Lakić, 1998; Higham, 2008; Anil et al., 2020), and Newton-Schulz iteration (Schulz, 1933; Higham, 2008). Our algorithm design accommodates any of these SVD solvers to best fit specific computational needs.

**Connection between MARS-Shampoo and Muon.** Muon (Jordan et al., 2024) is a recently proposed algorithm that utilizes the Newton-Schulz iteration (Higham, 2008; Schulz, 1933) to solve the SVD problem. It has demonstrated superior performance in terms of convergence speed when compared with AdamW and Shampoo in training large language models. The update rules of Muon are demonstrated as follows:

$$\mathbf{u}_t = \mu \mathbf{u}_{t-1} + \nabla f(\mathbf{x}_t, \boldsymbol{\xi}_t), \qquad (3.31)$$

$$\mathbf{m}_t = \mu \mathbf{u}_t + \nabla f(\mathbf{x}_t, \boldsymbol{\xi}_t), \qquad (3.32)$$

$$\mathbf{O}_t = \text{NewtonSchulz}\left(\mathbf{m}_t\right),$$

$$\mathbf{x}_{t+1} = \mathbf{x}_t - \eta_t(\mathbf{O}_t + \lambda \mathbf{x}_t).$$

Applying Lemma D.1 to (3.31) and (3.32), with $\mathbf{m}_t = \nabla f(\mathbf{x}_t, \boldsymbol{\xi}_t)$, $a_1 = \mu$, $a_2 = 1$, $b_1 = \mu$, $b_2 = 1$, we obtain an equivalent single update of momentum:

$$\mathbf{m}_t = \mu \mathbf{m}_{t-1} + \nabla f(\mathbf{x}_t, \boldsymbol{\xi}_t)$$
$$+ \mu\big(\nabla f(\mathbf{x}_t, \boldsymbol{\xi}_t) - \nabla f(\mathbf{x}_{t-1}, \boldsymbol{\xi}_{t-1})\big). \quad (3.33)$$

On the other hand, taking $\beta_1 = \mu$, $\gamma_t = 1 - \mu = 1 - \beta_1$ in MARS, (3.12) and (3.13) reduces to

$$\mathbf{m}_t = \mu \mathbf{m}_{t-1} + (1 - \mu)\nabla f(\mathbf{x}_t, \boldsymbol{\xi}_t)$$
$$+ \mu(1 - \mu)\big(\nabla f(\mathbf{x}_t, \boldsymbol{\xi}_t) - \nabla f(\mathbf{x}_{t-1}, \boldsymbol{\xi}_t)\big).$$

By dividing both sides of the above equation by $1 - \mu$, we obtain

$$\frac{\mathbf{m}_t}{1 - \mu} = \mu \cdot \frac{\mathbf{m}_{t-1}}{1 - \mu} + \nabla f(\mathbf{x}_t, \boldsymbol{\xi}_t)$$
$$+ \mu\big(\nabla f(\mathbf{x}_t, \boldsymbol{\xi}_t) - \nabla f(\mathbf{x}_{t-1}, \boldsymbol{\xi}_t)\big). \quad (3.34)$$

In can be seen that (3.34) is a rescaled version of (3.33), except that the stochastic gradients $\nabla f(\mathbf{x}_t, \boldsymbol{\xi}_t)$ and $\nabla f(\mathbf{x}_{t-1}, \boldsymbol{\xi}_t)$ are taken both at $\boldsymbol{\xi}_t$.

## 4  Experiments

In this section, we evaluate the performances of two instantiations of our algorithm, MARS-AdamW and MARS-Lion[2], in comparison with AdamW (Loshchilov & Hutter, 2019), the predominant algorithm for training large language models, Lion (Chen et al., 2023) and Muon (Jordan et al., 2024) on GPT-2 model series. More experiment results and abalation study, including the computer vision experiments, the effect of different learning rate schedulers, as well as sensitivity to $\gamma$ and batch size, are postponed to Section E.

---

[2]For the sake of training efficiency, we use MARS-approx for the experiments as the default configuration, except for Appendices E.2 and E.4. Discussion of the difference in performance between MARS-exact and MARS-approx is in Appendix E.2.

### 4.1  Experimental Setup

All our experiments are done based on the nanoGPT (Karpathy, 2022) implementation of the GPT-2 (Radford et al., 2019) architecture, and on the OpenWebText (Gokaslan et al., 2019) dataset. The training and validation sets contain approximately 9 billion and $4.4$ million tokens, respectively, all preprocessed using the GPT-2 tokenizer. We conduct experiments on three scales of GPT-2 models: small (125M parameters), medium (355M parameters), and large (770M parameters). Per the nanoGPT configurations, we disabled biases, applied GeLU activations, and set the Dropout rate (Srivastava et al., 2014) to $0.0$. We utilized 16 NVIDIA A100 GPUs for training the small models. For the medium and large models, training was conducted on 32 NVIDIA A100 GPUs and 32 NVIDIA H100 GPUs, respectively. Other hyper-parameters of training are listed in Appendix F.

### 4.2  Results

In Figure 1 and Figures 2–3 (in the Appendix), we demonstrate the training and validation losses as a function of training tokens and wall-clock time for various model sizes[3]. Across the small, medium, and large GPT-2 models, MARS consistently surpasses both the AdamW and Muon baselines in training and validation losses. The performance gap becomes more pronounced with increasing model size. Notably, MARS exhibits both rapid initial decay and sustained superiority throughout the training process. Further, we explore the performance of additional learning rate choices in Appendix E. Notably, the best validation losses of MARS-AdamW and MARS-Lion achieved in our GPT-2 large experiments are 2.511 and 2.534. For comparison, the best validation losses are 2.568, 2.565 and 2.606 for AdamW, Lion and Muon. These results demonstrate that our reported performance is highly competitive with state-of-the-art optimizers.

In Figure 1(c), as well as Figures 2(c) and 3(c) in the Appendix, we compare the wall-clock time of different algorithms. We observe that MARS-AdamW and MARS-Lion have a slightly higher per-iteration cost compared to AdamW but is much faster than Muon. Additionally, they consistently demonstrate lower validation losses than both AdamW, Muon and Lion within equivalent training durations.

We also evaluate 0-shot and 5-shot performances of our optimizer on common benchmarks including ARC (Yadav et al., 2019), BoolQ (Clark et al., 2019), HellaSwag (Zellers et al., 2019), OBQA (Mihaylov et al., 2018), PIQA (Bisk et al., 2020), WinoGrande (Sakaguchi et al., 2020) and MMLU (Hendrycks et al., 2021), with the `lm-evaluation-harness` codebase (Gao et al., 2024). We only list the 5-shot performances for large mod-

---

[3]The training loss curves are smoothed using Exponential Moving Average.

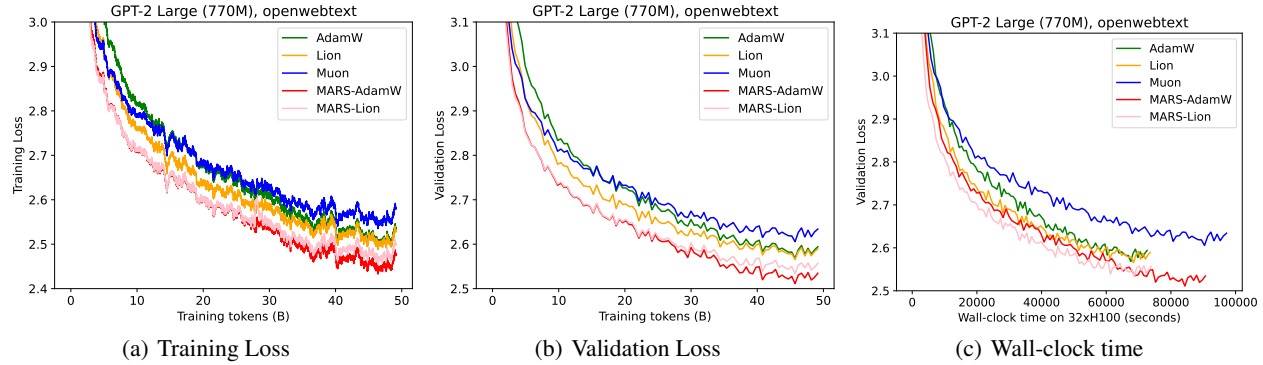

(a) Training Loss          (b) Validation Loss          (c) Wall-clock time

*Figure 1.* The training and validation loss curves, plotted against both training tokens and wall-clock time on GPT-2 large model (770M).

*Table 1.* The evaluation results of large models pre-trained using the OpenWebText dataset (5-shot with lm-evaluation-harness). The best scores in each column are **bolded**. Abbreviations: HellaSwag = HellaSwag, WG = WinoGrande.

| Method | ARC-E | ARC-C | BoolQ | HellaSwag | OBQA | PIQA | WG | MMLU | SciQ | Avg. |
|---|---|---|---|---|---|---|---|---|---|---|
| AdamW | 52.95 | 28.67 | 56.33 | 42.55 | 29.40 | 67.68 | 52.01 | 25.27 | 82.90 | 48.64 |
| Lion | 52.53 | 26.88 | 52.42 | 43.41 | 29.80 | 67.63 | 54.46 | 24.70 | **85.70** | 48.61 |
| Muon | 49.58 | 26.88 | 55.78 | 40.42 | 30.20 | 66.65 | 52.64 | 24.58 | 79.10 | 47.31 |
| MARS-AdamW | 54.04 | 26.28 | **62.78** | **45.66** | **31.60** | 68.12 | 52.49 | 25.93 | 84.50 | **50.15** |
| MARS-Lion | **54.25** | **28.92** | 56.36 | 44.00 | 29.20 | **69.10** | **54.93** | **25.98** | 85.50 | 49.80 |

els in Table 1, and leave other results in the Appendix E.1. The models pre-trained with MARS-AdamW and MARS-Lion outperform those pre-trained with AdamW, Muon and Lion optimizers, validating an enhanced downstream performance within the same number of pre-training steps.

## 5  Conclusion

In this work, we introduce MARS, a unified framework for adaptive gradient methods that integrates variance reduction techniques to improve the training of large models. Our approach combines the adaptive learning rate introduced by preconditioning with the faster convergence enabled by variance reduction. Within our framework, we have developed three optimization algorithms based on the ideas of AdamW, Lion, and Shampoo. Through extensive empirical experiments on GPT-2 pre-training tasks, we demonstrate that MARS consistently outperforms baseline algorithms in terms of both token efficiency and wall-clock time. Our results establish a generic framework for combining adaptive gradient methods with variance reduction techniques, contributing to the advancement of optimizers in large model training.

## Impact Statement

This paper presents work whose goal is to advance the field of optimization theory in deep learning. We believe that our work contributes meaningfully to the field, specifically on advancing the efficiency in the pre-training stage of deep learning models, especially Large Language Models. By involvement of variance reduction in adaptive learning methods, our method can greatly lower the cost for pre-training language models on limited training corpora and more resource-constrained devices as well as in broader settings, opening new avenues for their application in various downstream tasks. Improvement in efficiency typically correlates with reduced energy consumption, potentially decreasing the environmental footprint of LLM pre-training. This advancement underscores the potential of optimization method development in deep learning field in both technological and societal contexts.

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

# Appendix

# A    Related Work

In this section, we provide a review of additional related works, including some previously mentioned, to help readers gain a deeper understanding of the history and development of adaptive gradient methods and variance reduction techniques.

**Adaptive Gradient Methods.** RProp (Riedmiller & Braun, 1993) is probably one of the earliest adaptive gradient methods by dynamically adjusting the learning rate. AdaGrad (Duchi et al., 2011; McMahan & Streeter, 2010) adjusts the learning rate based on the geometry of the training data observed during earlier iterations. To tackle with the issue of diminishing gradient in AdaGrad (Carlson et al., 2015a;b), Tieleman (2012) introduced RMSProp by incorporating the idea of exponential moving average. A significant advancement came with Adam (Kingma & Ba, 2015), which integrated RMSProp with Nesterov's momentum (Nesterov, 1983; 2013) achieving superior performance and becoming a prevalent optimizer in deep neural network training. Later, Loshchilov & Hutter (2019) proposed to decouple weight decay from gradient calculations in Adam and introduced AdamW, an optimization algorithm having become the predominant optimization algorithm in contemporary deep learning applications. To fix the convergence issue of Adam, Reddi et al. (2019b) introduced the AMSGrad optimizer, which maintains a running maximum of past second-order momentum terms to achieve non-increasing step sizes. Subsequently, Chen et al. (2018) unified AMSGrad and SGD within the Padam framework by introducing a partial adaptive parameter to control the degree of adaptiveness. Notably, AdamW and its variations have been widely used in the training of popular large language models, including OPT (Zhang et al., 2022a), Llama 3 (Dubey et al., 2024), and DeepSeek-V2 (Liu et al., 2024).

**Variance Reduction Methods.** SAG(Roux et al., 2012) and SDCA(Shalev-Shwartz & Zhang, 2013) were among the first attempts to apply variance reduction techniques to accelerate the convergence of SGD. Subsequently, simpler algorithms like SVRG(Johnson & Zhang, 2013) and SAGA(Defazio et al., 2014) were introduced, achieving the same improved convergence rates. SARAH (Nguyen et al., 2017a) further simplified these approaches by employing biased recursive gradient estimation, which reduces storage requirements while achieving the complexity bounds for convex optimization problems. And some researchers have also attempted to apply preconditioning into variance reduction in the convex setting (Frangella et al., 2024; Derezinski). For non-convex optimization, besides SVRG (Allen-Zhu & Yuan, 2016; Reddi et al., 2016) and SARAH (Nguyen et al., 2017b), SPIDER (Fang et al., 2018) integrates Normalized Gradient Descent (Nesterov, 2013; Hazan et al., 2015) with recursive estimation of gradients, while SNVRG (Zhou et al., 2020) introduces multiple reference points for semi-stochastic gradient calculation for improved variance reduction and convergence rate. SpiderBoost (Wang et al., 2019) refines SPIDER by enabling the use of a significantly larger constant step size while preserving the same near-optimal oracle complexity. Subsequently, STORM (Cutkosky & Orabona, 2019) was proposed to further simplifies the SPIDER and SNVRG algorithms through the use of stochastic recursive momentum. This was later improved into a parameter-free variant, namely STORM+(Levy et al., 2021).

**Variance Reduction for Adaptive Gradient Methods.** Few works have explored the application of variance reduction techniques to adaptive gradient methods. To the best of our knowledge, the only exceptions are $\text{Adam}^+$, SuperAdam and AdaSPIDER. $\text{Adam}^+$ (Liu et al., 2020) attempts to reduce the variance of first-order moment into Adam by estimating the gradient only at extrapolated points. SuperAdam (Huang et al., 2021) and VRAdam (Li, 2024) integrates variance reduction with AdamW to achieve improved convergence rates. And AdaSPIDER (Kavis et al., 2022) introduced adaptive step size in SPIDER algorithm. However, these variance-reduced adaptive gradient methods have primarily been validated on basic computer vision tasks, such as MNIST (Schölkopf & Smola, 2002) and CIFAR-10 (Krizhevsky et al., 2009), and simple natural language modeling tasks, like SWB-300 (Saon et al., 2017), using straightforward architectures such as LeNet (LeCun et al., 1998), ResNet-32 (He et al., 2016), 2-layer LSTMs (Graves & Graves, 2012), and 2-layer Transformers (Vaswani, 2017). As a result, a significant gap remains in the successful application of variance reduction techniques to adaptive gradient methods, particularly in the rapidly evolving domain of large language models.

# B    Theoretical Analysis

## B.1    Connection to Nesterov's Acceleration

Many optimization algorithms exhibit similarities with Nesterov's acceleration and can be considered adaptations of Nesterov's acceleration with varying parameterization schedules, as discussed in works such as Defazio et al. (2024) and Xie et al. (2024). In this section, we compare and contrast Nesterov's momentum and STORM momentum used in our paper. Specifically, Nesterov's accelerated gradient descent can be equivalently written as (Xie et al., 2024):

$$\mathbf{m}_t = \beta_1 \mathbf{m}_{t-1} + \left[ \nabla f(\mathbf{x}_t, \boldsymbol{\xi}_t) + \beta_1 (\nabla f(\mathbf{x}_t, \boldsymbol{\xi}_t) - \nabla f(\mathbf{x}_{t-1}, \boldsymbol{\xi}_{t-1})) \right],$$

while the STORM momentum is

$$\mathbf{m}_t = \beta_1 \mathbf{m}_{t-1} + (1 - \beta_1)\nabla f(\mathbf{x}_t, \boldsymbol{\xi}_t) + \beta_1\big(\nabla f(\mathbf{x}_t, \boldsymbol{\xi}_t) - \nabla f(\mathbf{x}_{t-1}, \boldsymbol{\xi}_t)\big)$$

The most significant difference lies in the noise handling schemes. In Nesterov's acceleration, $\nabla f(\mathbf{x}_t, \boldsymbol{\xi}_t)$ is subtracted by $\nabla f(\mathbf{x}_{t-1}, \boldsymbol{\xi}_{t-1})$ to determine a direction of improvement. In contrast, STORM variance reduction subtracts $\nabla f(\mathbf{x}_{t-1}, \boldsymbol{\xi}_t)$ from $\nabla f(\mathbf{x}_t, \boldsymbol{\xi}_t)$ to cancel out the noise introduced by $\boldsymbol{\xi}_t$. Furthermore, in our theoretical analysis (Section B.2), we prove that variance-reduced variants of AdamW achieve an improved convergence rate of $\mathcal{O}(T^{-1/3})$. Empirically, we also show that a variance reduced noise schedule performs better than its approximate counterpart.

## B.2 Convergence of MARS

Although Kingma & Ba (2015) did convergence analysis for Adam, Reddi et al. (2019a) pointed out that they made some mistakes in the proof, and they also proved that in some special cases, Adam does not converge. However, there are some attempts to prove the convergence of Adam and AdamW in special circumstances (Zhang et al., 2022b; Li et al., 2024; Zhou et al., 2024). Our algorithm, MARS, is also based on AdamW. In addition, it involves the property of variance reduction. We prove that MARS can converge with a better convergence rate with careful selection of hyperparameters.

To help analyze the convergence of the algorithm, we make the following assumptions:

**Assumption B.1** (Bounded Variance). We assume that the variance of gradient estimator is bounded by $\sigma^2$. i.e., for any noise $\boldsymbol{\xi}$, parameter $\boldsymbol{x}$, and $\nabla F(\boldsymbol{x}) = \mathbb{E}[\nabla f(\boldsymbol{x}, \boldsymbol{\xi})]$, there exists a positive $\sigma$ such that:

$$\mathbb{E}\big[\|\nabla f(\mathbf{x}, \boldsymbol{\xi}) - \nabla F(\mathbf{x})\|_2^2\big] \leq \sigma^2. \tag{B.1}$$

**Assumption B.2** ($L$-Smoothness). We assume that for arbitrary $\boldsymbol{\xi}$, $f(\boldsymbol{x}, \boldsymbol{\xi})$ is $L$-smooth:

$$\|\nabla f(\mathbf{x}, \boldsymbol{\xi}) - \nabla f(\mathbf{y}, \boldsymbol{\xi})\|_2 \leq L\|\mathbf{x} - \mathbf{y}\|_2, \ \forall \mathbf{x}, \mathbf{y}. \tag{B.2}$$

**Assumption B.3** ($H$ Lower Bounded). We assume that there is a constant $\rho > 0$ such that for all $\mathbf{H}_t, t > 0$, $\mathbf{H}_t \succ \rho \boldsymbol{I}$.

**Remark B.4.** Note that this is implicitly satisfied by the instantiations of MARS since we add a small $\epsilon$ to semi-positive definite $\mathbf{H}_t$ for computational stability.

We proposed Theorem B.5 for our main Algorithm 1 and Theorem B.6 for MARS-AdamW (Algorithm 2, where an additional weight decay is involved). We note that for theoretical analysis, it is necessary to consider time-varying parameters $\beta_{1,t}$ and $\beta_{2,t}$. However, in practice, these parameters are typically set as constants.

**Theorem B.5.** In Algorithm 1, under Assumptions B.1, B.2 and B.3, when choosing $\eta_t = (s + t)^{-1/3}, s \geq 8L^3/\rho^3$. Suppose $c \geq 32L^2\rho^{-2} + 1$, $\beta_{1,t+1} = 1 - c\eta_t^2$ and $\beta_{2,t+1} = 1 - \eta_t^6$, then $\forall T \geq s$, it holds that

$$\frac{1}{T}\sum_{t=1}^{T}\mathbb{E}\|\nabla F(\mathbf{x}_t) - \mathbf{m}_t\|_2^2 \leq \Big(2\rho G + \frac{\rho c^2 \sigma^2}{4L^2} \cdot \log(s + T)\Big) \cdot \frac{1}{T^{2/3}} - \frac{\rho^2 \sum_{t=1}^{T} M_{t+1}}{8L^2 T^{1/3}},$$

$$\frac{1}{T}\sum_{t=1}^{T}\frac{1}{\eta_t} \cdot \mathbb{E}\|\mathbf{x}_{t+1} - \mathbf{x}_t\|_2^2 \leq \Big(\frac{16G}{3\rho} + \frac{2c^2 \sigma^2}{3L^2} \cdot \log(s + T)\Big) \cdot \frac{1}{T^{2/3}} - \frac{\sum_{t=1}^{T} M_{t+1}}{6L^2 T^{1/3}},$$

where $G = F(\mathbf{x}_1) - \min_{\mathbf{x}} F(\mathbf{x}) + \frac{\rho s^{1/3}\sigma^2}{16L^2}$ and $M_{t+1}$ is defined in (C.2).

**Theorem B.6.** In Algorithm 2, under Assumptions B.1, B.2 and B.3, when choosing $\eta_t = (s + t)^{-1/3}, s \geq \max(8L^3/\rho^3, 64\lambda^3)$. Suppose $||\mathbf{x}_t||_2 \leq D$, $c \geq 32L^2\rho^{-2} + 1$, $\beta_{1,t+1} = 1 - c\eta_t^2$ and $\beta_{2,t+1} = 1 - \eta_t^6$, then $\forall T \geq s$, it holds that

$$\frac{1}{T}\sum_{t=1}^{T}\mathbb{E}\|\nabla F(\mathbf{x}_t) - \mathbf{m}_t\|_2^2 \leq \Big(2\rho(G + \lambda D^2 \log(s + T)) + \frac{\rho c^2 \sigma^2}{4L^2} \cdot \log(s + T)\Big) \cdot \frac{1}{T^{2/3}} - \frac{\rho^2 \sum_{t=1}^{T} M_{t+1}}{16L^2 T^{1/3}},$$

$$\frac{1}{T}\sum_{t=1}^{T}\frac{1}{\eta_t} \cdot \mathbb{E}\|\mathbf{x}_{t+1} - \mathbf{x}_t\|_2^2 \leq \Big(\frac{16(G + \lambda D^2 \log(s + T))}{\rho} + \frac{2c^2 \sigma^2}{L^2} \cdot \log(s + T)\Big) \cdot \frac{1}{T^{2/3}} - \frac{\sum_{t=1}^{T} M_{t+1}}{L^2 T^{1/3}}.$$

where $G = F(\mathbf{x}_1) - \min_{\mathbf{x}} F(\mathbf{x}) + \frac{\lambda}{2}D^2(1 + \epsilon) + \frac{\rho s^{1/3}\sigma^2}{16L^2}$ and $M_{t+1}$ is defined in (C.2).

The theorems above guarantee the convergence rate of $O(\log(T)/T^{1/3})$. We remark that even though it seems that the involvement of weight decay may results in slower convergence, it performs better in practice, à la Loshchilov & Hutter (2019). Finally, the term $M_{t+1}$ is always greater or equal to $0$, and when $\gamma_{t+1} = 1$, we would have $M_{t+1} = 0$. The dependency of $M_{t+1}$ on $\gamma_{t+1}$ is shown in (C.2) and Lemma C.2. This proves that the convergence is faster if we allow a flexible $\gamma_t$ schedule.

# C  Proof of Theorems

First, we introduce auxiliary lemmas necessary for proving the theorems.

**Lemma C.1.** In Algorithm 2, for any $0 \leq \beta_{2,t} \leq 1$ and $\forall t \geq 1$, the following inequality holds:

$$\|\sqrt{\mathbf{v}_t} - \sqrt{\mathbf{v}_{t+1}}\|_\infty \leq \sqrt{2(1 - \beta_{2,t})}. \tag{C.1}$$

The proof is in Section D.3.

**Lemma C.2.** In Algorithm 1. Under Assumption B.1 and B.2, if $1 \geq \beta_{1,t+1} \geq 0, \forall t$, under approximate choice of $\gamma_{t+1}$ as in (D.12), we have

$$\mathbb{E}\|\nabla F(\mathbf{x}_{t+1}) - \mathbf{m}_{t+1}\|_2^2 \leq \beta_{1,t+1}^2 \mathbb{E}\|\nabla F(\boldsymbol{x}_t) - \mathbf{m}_t\|_2^2 + 2\beta_{1,t+1}^2 L^2 \mathbb{E}\|\mathbf{x}_{t+1} - \mathbf{x}_t\|_2^2 + 2(1 - \beta_{1,t+1})^2 \sigma^2 - M_{t+1}$$

where

$$M_{t+1} := \mathbb{E}\|\nabla f(\mathbf{x}_{t+1}, \boldsymbol{\xi}_{t+1}) - \nabla f(\mathbf{x}_t, \boldsymbol{\xi}_{t+1})\|_2^2 \left( A_{t+1}^2 - \left( \beta_{1,t+1}(1 - \gamma_{t+1}) - A_{t+1} \right)^2 \right), \tag{C.2}$$

$$A_{t+1} := \frac{G_{t+1} + \beta_{1,t+1} \text{tr}\left( \text{Var}\left( \nabla f(\mathbf{x}_{t+1}, \boldsymbol{\xi}_{t+1}) - \nabla f(\mathbf{x}_t, \boldsymbol{\xi}_{t+1}) \right) \right)}{\mathbb{E}\|\nabla f(\mathbf{x}_{t+1}, \boldsymbol{\xi}_{t+1}) - \nabla f(\mathbf{x}_t, \boldsymbol{\xi}_{t+1})\|_2^2}$$

and

$$G_{t+1} := (1 - \beta_{1,t+1}) \mathbb{E}\left\langle \nabla f(\mathbf{x}_{t+1}, \boldsymbol{\xi}_{t+1}) - \nabla f(\mathbf{x}_t, \boldsymbol{\xi}_{t+1}), \nabla f(\mathbf{x}_{t+1}, \boldsymbol{\xi}_{t+1}) - \nabla F(\mathbf{x}_{t+1}) \right\rangle$$
$$+ \beta_{1,t+1} \mathbb{E}\left\langle \nabla F(\mathbf{x}_t) - \nabla F(\mathbf{x}_t), F(\mathbf{x}_t) - \mathbf{m}_t \right\rangle.$$

The proof is in Section D.4.

**Lemma C.3.** In Algorithm 1. With Assuptions B.2 and B.3 and $\eta_t \leq \rho \cdot (2L)^{-1}, \forall t \geq 1$, it holds that

$$F(\mathbf{x}_{t+1}) \leq F(\mathbf{x}_t) - \frac{\rho}{2\eta_t} \cdot \|\mathbf{x}_t - \mathbf{x}_{t+1}\|_2^2 + \frac{\eta_t}{\rho} \cdot \|\nabla F(\mathbf{x}_t) - \mathbf{m}_t\|_2^2.$$

The proof of Lemma C.3 is in Section D.5.

**Lemma C.4.** In Algorithm 2. With Assuptions B.2 and B.3 and $\eta_t \leq \min\{(4\lambda)^{-1}, \rho \cdot (2L)^{-1}\}, \forall t \geq 1$, it holds that

$$F(\mathbf{x}_{t+1}) + \frac{\lambda}{2} \cdot \mathbf{x}_{t+1}^\top \mathbf{H}_{t+1} \mathbf{x}_{t+1} \leq F(\mathbf{x}_t) + \frac{\lambda}{2} \cdot \mathbf{x}_t^\top \mathbf{H}_t \mathbf{x}_t - \frac{\rho}{4\eta_t} \cdot \|\mathbf{x}_t - \mathbf{x}_{t+1}\|_2^2$$
$$+ \frac{\eta_t}{\rho} \cdot \|\nabla F(\mathbf{x}_t) - \mathbf{m}_t\|_2^2 + \frac{\lambda}{2}\sqrt{2(1 - \beta_{2,t})}D^2.$$

The proof of Lemma C.4 is in Section D.5.

**Lemma C.5.** Let $\eta_t = (s + t)^{-1/3}, s \geq 1, \forall t \geq 0$. Then $\eta_t^{-1} - \eta_{t-1}^{-1} \leq \eta_t, \forall t \geq 1$.

### C.1 Proof of Theorem B.5

*Proof of Theorem B.5.* First, we define the Lyapunov function as

$$\Phi_t = \mathbb{E}\Big[F(\mathbf{x}_t) + \frac{\rho}{16L^2\eta_{t-1}} \cdot \|\nabla F(\mathbf{x}_t) - \mathbf{m}_t\|_2^2\Big], \quad \forall t \geq 1.$$

Then we calculate the difference between two consecutive Lyapunov functions as:

$$\Phi_{t+1} - \Phi_t = \underbrace{\mathbb{E}[F(\mathbf{x}_{t+1}) - F(\mathbf{x}_t)]}_{I_1} + \underbrace{\mathbb{E}\Big[\frac{\rho}{16L^2\eta_t} \cdot \|\nabla F(\mathbf{x}_{t+1}) - \mathbf{m}_{t+1}\|_2^2 - \frac{\rho}{16L^2\eta_{t-1}} \cdot \|\nabla F(\mathbf{x}_t) - \mathbf{m}_t\|_2^2\Big]}_{I_2}. \quad (C.3)$$

For $I_1$, we use Lemma C.3 to obtain

$$I_1 \leq \mathbb{E}\Big[-\frac{\rho}{2\eta_t} \cdot \|\mathbf{x}_t - \mathbf{x}_{t+1}\|_2^2 + \frac{\eta_t}{\rho} \cdot \|\nabla F(\mathbf{x}_t) - \mathbf{m}_t\|_2^2\Big]. \quad (C.4)$$

For $I_2$, we use Lemma C.2 to obtain

$$I_2 = \mathbb{E}\Big[\frac{\rho}{16L^2\eta_t} \cdot \|\nabla F(\mathbf{x}_{t+1}) - \mathbf{m}_{t+1}\|_2^2 - \frac{\rho}{16L^2\eta_{t-1}} \cdot \|\nabla F(\mathbf{x}_t) - \mathbf{m}_t\|_2^2\Big]$$

$$\leq \frac{\rho}{16L^2} \cdot \Big(\frac{\beta_{1,t+1}^2}{\eta_t} - \frac{1}{\eta_{t-1}}\Big)\mathbb{E}\|\nabla F(\mathbf{x}_t) - \mathbf{m}_t\|_2^2 + \frac{\rho\beta_{1,t+1}^2}{8\eta_t} \cdot \mathbb{E}\|\mathbf{x}_{t+1} - \mathbf{x}_t\|_2^2 + \frac{\rho(1-\beta_{1,t+1})^2\sigma^2}{8L^2\eta_t} - \frac{\rho}{16L^2\eta_t}M_{t+1}$$

$$\leq \frac{\rho}{16L^2} \cdot \Big(\frac{\beta_{1,t+1}^2}{\eta_t} - \frac{1}{\eta_{t-1}}\Big)\mathbb{E}\|\nabla F(\mathbf{x}_t) - \mathbf{m}_t\|_2^2 + \frac{\rho}{8\eta_t} \cdot \mathbb{E}\|\mathbf{x}_{t+1} - \mathbf{x}_t\|_2^2 + \frac{\rho c^2\eta_t^3\sigma^2}{8L^2} - \frac{\rho}{16L^2\eta_t}M_{t+1}, \quad (C.5)$$

where the last inequality follows from the definition that $\beta_{1,t+1} = 1 - c\eta_t^2$. Further, for the first term on the right hand side, we have

$$\frac{\rho}{16L^2} \cdot \Big(\frac{\beta_{1,t+1}^2}{\eta_t} - \frac{1}{\eta_{t-1}}\Big) \leq \frac{\rho}{16L^2} \cdot \Big(\frac{\beta_{1,t+1}}{\eta_t} - \frac{1}{\eta_{t-1}}\Big) = \frac{\rho}{16L^2} \cdot \Big(\frac{1-c\eta_t^2}{\eta_t} - \frac{1}{\eta_{t-1}}\Big) = \frac{\rho}{16L^2} \cdot \Big(\frac{1}{\eta_t} - \frac{1}{\eta_{t-1}} - c\eta_t\Big).$$

From Lemma C.5, we know that $\frac{1}{\eta_t} - \frac{1}{\eta_{t-1}} < \eta_t$. Choosing $c$ such that $c \geq 32L^2\rho^{-2} + 1$, we obtain

$$\frac{\rho}{16L^2} \cdot \Big(\frac{\beta_{1,t+1}^2}{\eta_t} - \frac{1}{\eta_{t-1}}\Big) \leq \frac{\rho}{16L^2} \cdot (\eta_t - c\eta_t) \leq -2\eta_t\rho^{-1}. \quad (C.6)$$

Bringing (C.6) into (C.5), we arrive at the upper bound for $I_2$:

$$I_2 \leq -\frac{2\eta_t}{\rho}\mathbb{E}\|\nabla F(\mathbf{x}_t) - \mathbf{m}_t\|_2^2 + \frac{\rho}{8\eta_t} \cdot \mathbb{E}\|\mathbf{x}_{t+1} - \mathbf{x}_t\|_2^2 + \frac{\rho c^2\eta_t^3\sigma^2}{8L^2} - \frac{\rho}{16L^2\eta_t}M_{t+1}. \quad (C.7)$$

Now combining (C.3), (C.4) and (C.7), we derive

$$\Phi_{t+1} - \Phi_t \leq -\frac{\eta_t}{\rho}\mathbb{E}\|\nabla F(\mathbf{x}_t) - \mathbf{m}_t\|_2^2 - \frac{3\rho}{8\eta_t} \cdot \mathbb{E}\|\mathbf{x}_{t+1} - \mathbf{x}_t\|_2^2 + \frac{\rho c^2\eta_t^3\sigma^2}{8L^2} - \frac{\rho}{16L^2\eta_t}M_{t+1}.$$

Taking a telescoping sum for $t = 1, \cdots, T$ gives

$$\sum_{t=1}^{T}\Big(\frac{\eta_t}{\rho}\mathbb{E}\|\nabla F(\mathbf{x}_t) - \mathbf{m}_t\|_2^2 + \frac{3\rho}{8\eta_t} \cdot \mathbb{E}\|\mathbf{x}_{t+1} - \mathbf{x}_t\|_2^2\Big) \leq \Phi_1 - \Phi_{T+1} + \frac{\rho c^2\sigma^2}{8L^2}\sum_{t=1}^{T}\frac{1}{s+t} - \sum_{t=1}^{T}\frac{\rho}{16L^2\eta_t}M_{t+1}$$

$$\leq \Phi_1 - \Phi_{T+1} + \frac{\rho c^2\sigma^2}{8L^2} \cdot \log(s+T) - \sum_{t=1}^{T}\frac{\rho}{16L^2\eta_t}M_{t+1}.$$

By the definition of $\Phi_t$, we have $\Phi_{T+1} \geq F(\mathbf{x}_{T+1}) \geq \min_{\mathbf{x}} F(\mathbf{x})$. And for $\Phi_1$,

$$\Phi_1 = \mathbb{E}\Big[F(\mathbf{x}_1) + \frac{\rho s^{1/3}}{16L^2} \cdot \|\nabla F(\mathbf{x}_1) - \mathbf{m}_1\|_2^2\Big] = F(\mathbf{x}_1) + \frac{\rho s^{1/3}}{16L^2} \cdot \mathbb{E}[\|\nabla F(\mathbf{x}_1) - \nabla f(\mathbf{x}_1, \boldsymbol{\xi}_1)\|_2^2] \leq F(\mathbf{x}_1) + \frac{\rho s^{1/3}\sigma^2}{16L^2}.$$

Consequently, defining $G = F(\mathbf{x}_1) - \min_{\mathbf{x}} F(\mathbf{x}) + \frac{\rho s^{1/3} \sigma^2}{16L^2}$, the following inequality holds:

$$\frac{1}{T}\sum_{t=1}^{T}\left(\frac{\eta_t}{\rho}\mathbb{E}\|\nabla F(\mathbf{x}_t) - \mathbf{m}_t\|_2^2 + \frac{3\rho}{8\eta_t}\cdot\mathbb{E}\|\mathbf{x}_{t+1} - \mathbf{x}_t\|_2^2\right) \le \frac{G}{T} + \frac{\rho c^2 \sigma^2}{8L^2 T}\cdot\log(s+T) - \frac{1}{T}\sum_{t=1}^{T}\frac{\rho}{16L^2\eta_t}M_{t+1}. \quad \text{(C.8)}$$

Dealing with the two terms on the left hand side separately, we have

$$\frac{1}{T}\sum_{t=1}^{T}\mathbb{E}\|\nabla F(\mathbf{x}_t) - \mathbf{m}_t\|_2^2 \le \frac{\rho G}{T\eta_T} + \frac{\rho^2 c^2 \sigma^2}{8L^2 T\eta_T}\cdot\log(s+T) - \frac{1}{T}\sum_{t=1}^{T}\frac{\rho^2}{16L^2\eta_t^2}M_{t+1}$$

$$\le \left(\rho G + \frac{\rho^2 c^2 \sigma^2}{8L^2}\cdot\log(s+T)\right)\cdot\frac{(s+T)^{1/3}}{T} - \frac{1}{T}\sum_{t=1}^{T}\frac{\rho^2}{16L^2\eta_t^2}M_{t+1}$$

$$\le \left(2\rho G + \frac{\rho c^2 \sigma^2}{4L^2}\cdot\log(s+T)\right)\cdot\frac{1}{T^{2/3}} - \frac{\rho^2\sum_{t=1}^{T}M_{t+1}}{8L^2 T^{1/3}},$$

where the last inequality holds when $T \ge s$. Similarly, for the second term in (C.8), we also have

$$\frac{1}{T}\sum_{t=1}^{T}\frac{3\rho}{8\eta_t}\cdot\mathbb{E}\|\mathbf{x}_{t+1} - \mathbf{x}_t\|_2^2 \le \frac{G}{T} + \frac{\rho c^2 \sigma^2}{8L^2 T}\cdot\log(s+T) - \frac{\rho\sum_{t=1}^{T}M_{t+1}}{16L^2 T^{2/3}},$$

which implies that when $T \ge s$, due to the definition of $\eta_t$,

$$\frac{1}{T}\sum_{t=1}^{T}\frac{1}{\eta_t^2}\cdot\mathbb{E}\|\mathbf{x}_{t+1} - \mathbf{x}_t\|_2^2 \le \frac{8G}{3\rho T\eta_T} + \frac{c^2 \sigma^2}{3L^2 T\eta_T}\cdot\log(s+T) - \frac{\sum_{t=1}^{T}M_{t+1}}{6L^2 T^{2/3}}$$

$$\le \frac{16G}{3\rho T^{2/3}} + \frac{2c^2 \sigma^2}{3L^2 T^{2/3}}\cdot\log(s+T) - \frac{\sum_{t=1}^{T}M_{t+1}}{6L^2 T^{1/3}}.$$

That concludes our proof. $\qquad\square$

## C.2  Proof of Theorem B.6

*Proof of Theorem B.6.*  First, we define the Lyapunov function as

$$\Phi_t = \mathbb{E}\left[F(\mathbf{x}_t) + \frac{\lambda}{2}\cdot\mathbf{x}_t^\top\mathbf{H}_t\mathbf{x}_t + \frac{\rho}{16L^2\eta_{t-1}}\cdot\|\nabla F(\mathbf{x}_t) - \mathbf{m}_t\|_2^2\right], \quad \forall t \ge 1.$$

Then we calculate the difference between two consecutive Lyapunov functions as:

$$\Phi_{t+1} - \Phi_t = \underbrace{\mathbb{E}\left[F(\mathbf{x}_{t+1}) + \frac{\lambda}{2}\cdot\mathbf{x}_{t+1}^\top\mathbf{H}_{t+1}\mathbf{x}_{t+1} - F(\mathbf{x}_t) - \frac{\lambda}{2}\cdot\mathbf{x}_t^\top\mathbf{H}_t\mathbf{x}_t\right]}_{I_1}$$

$$+ \underbrace{\mathbb{E}\left[\frac{\rho}{16L^2\eta_t}\cdot\|\nabla F(\mathbf{x}_{t+1}) - \mathbf{m}_{t+1}\|_2^2 - \frac{\rho}{16L^2\eta_{t-1}}\cdot\|\nabla F(\mathbf{x}_t) - \mathbf{m}_t\|_2^2\right]}_{I_2}. \quad \text{(C.9)}$$

For $I_1$, we use Lemma C.4 to obtain

$$I_1 \le \mathbb{E}\left[-\frac{\rho}{4\eta_t}\cdot\|\mathbf{x}_t - \mathbf{x}_{t+1}\|_2^2 + \frac{\eta_t}{\rho}\cdot\|\nabla F(\mathbf{x}_t) - \mathbf{m}_t\|_2^2 + \frac{\lambda D^2\sqrt{2(1 - \beta_{2,t})}}{2}\right]. \quad \text{(C.10)}$$

For $I_2$, we use Lemma C.2 to obtain

$$I_2 = \mathbb{E}\Big[\frac{\rho}{16L^2\eta_t}\cdot\|\nabla F(\mathbf{x}_{t+1})-\mathbf{m}_{t+1}\|_2^2 - \frac{\rho}{16L^2\eta_{t-1}}\cdot\|\nabla F(\mathbf{x}_t)-\mathbf{m}_t\|_2^2\Big]$$

$$\leq \frac{\rho}{16L^2}\cdot\Big(\frac{\beta_{1,t+1}^2}{\eta_t}-\frac{1}{\eta_{t-1}}\Big)\mathbb{E}\|\nabla F(\mathbf{x}_t)-\mathbf{m}_t\|_2^2 + \frac{\rho\beta_{1,t+1}^2}{8\eta_t}\cdot\mathbb{E}\|\mathbf{x}_{t+1}-\mathbf{x}_t\|_2^2 + \frac{\rho(1-\beta_{1,t+1})^2\sigma^2}{8L^2\eta_t} - \frac{\rho}{16L^2\eta_t}M_{t+1}$$

$$\leq \frac{\rho}{16L^2}\cdot\Big(\frac{\beta_{1,t+1}^2}{\eta_t}-\frac{1}{\eta_{t-1}}\Big)\mathbb{E}\|\nabla F(\mathbf{x}_t)-\mathbf{m}_t\|_2^2 + \frac{\rho}{8\eta_t}\cdot\mathbb{E}\|\mathbf{x}_{t+1}-\mathbf{x}_t\|_2^2 + \frac{\rho c^2\eta_t^3\sigma^2}{8L^2} - \frac{\rho}{16L^2\eta_t}M_{t+1}, \tag{C.11}$$

where the last inequality follows from the definition that $\beta_{1,t+1}=1-c\eta_t^2$. Further, for the first term on the right hand side, we have

$$\frac{\rho}{16L^2}\cdot\Big(\frac{\beta_{1,t+1}^2}{\eta_t}-\frac{1}{\eta_{t-1}}\Big) \leq \frac{\rho}{16L^2}\cdot\Big(\frac{\beta_{1,t+1}}{\eta_t}-\frac{1}{\eta_{t-1}}\Big) = \frac{\rho}{16L^2}\cdot\Big(\frac{1-c\eta_t^2}{\eta_t}-\frac{1}{\eta_{t-1}}\Big) = \frac{\rho}{16L^2}\cdot\Big(\frac{1}{\eta_t}-\frac{1}{\eta_{t-1}}-c\eta_t\Big).$$

From Lemma C.5, we know that $\frac{1}{\eta_t}-\frac{1}{\eta_{t-1}} < \eta_t$. Choosing $c$ such that $c \geq 32L^2\rho^{-2}+1$, we obtain

$$\frac{\rho}{16L^2}\cdot\Big(\frac{\beta_{1,t+1}^2}{\eta_t}-\frac{1}{\eta_{t-1}}\Big) \leq \frac{\rho}{16L^2}\cdot(\eta_t-c\eta_t) \leq -2\eta_t\rho^{-1}. \tag{C.12}$$

Bringing (C.12) into (C.11), we arrive at the upper bound for $I_2$:

$$I_2 \leq -\frac{2\eta_t}{\rho}\mathbb{E}\|\nabla F(\mathbf{x}_t)-\mathbf{m}_t\|_2^2 + \frac{\rho}{8\eta_t}\cdot\mathbb{E}\|\mathbf{x}_{t+1}-\mathbf{x}_t\|_2^2 + \frac{\rho c^2\eta_t^3\sigma^2}{8L^2} - \frac{\rho}{16L^2\eta_t}M_{t+1}. \tag{C.13}$$

Now combining Formulas (C.9), (C.10) and (C.13), we obtain

$$\Phi_{t+1} - \Phi_t \leq -\frac{\eta_t}{\rho}\mathbb{E}\|\nabla F(\mathbf{x}_t)-\mathbf{m}_t\|_2^2 - \frac{\rho}{8\eta_t}\cdot\mathbb{E}\|\mathbf{x}_{t+1}-\mathbf{x}_t\|_2^2 + \frac{\rho c^2\eta_t^3\sigma^2}{8L^2} + \frac{\lambda D^2\sqrt{2(1-\beta_{2,t})}}{2} - \frac{\rho}{16L^2\eta_t}M_{t+1}.$$

Taking a telescoping sum for $t=1,\cdots,T$ gives

$$\sum_{t=1}^{T}\Big(\frac{\eta_t}{\rho}\mathbb{E}\|\nabla F(\mathbf{x}_t)-\mathbf{m}_t\|_2^2 + \frac{\rho}{8\eta_t}\cdot\mathbb{E}\|\mathbf{x}_{t+1}-\mathbf{x}_t\|_2^2\Big)$$

$$\leq \Phi_1 - \Phi_{T+1} + \frac{\rho c^2\sigma^2}{8L^2}\sum_{t=1}^{T}\frac{1}{s+t} + \sum_{t=1}^{T}\frac{\lambda D^2\sqrt{2(1-\beta_{2,t})}}{2} - \sum_{t=1}^{T}\frac{\rho}{16L^2\eta_t}M_{t+1}$$

$$\leq \Phi_1 - \Phi_{T+1} + \frac{\rho c^2\sigma^2}{8L^2}\cdot\log(s+T) + \lambda D^2\log(s+T) - \sum_{t=1}^{T}\frac{\rho}{16L^2\eta_t}M_{t+1},$$

where the last inequality follows by taking $\beta_{2,t}=1-\eta_t^6$. By the definition of $\Phi_t$, we have $\Phi_{T+1} \geq F(\mathbf{x}_{t+1}) \geq \min_{\mathbf{x}} F(\mathbf{x})$. And for $\Phi_1$, according to the fact following (D.7) that $\|\mathbf{H}_{t+1}\|_2 = \Big\|\text{diag}(\sqrt{\mathbf{v}_{t+1}}+\epsilon)\Big\|_2 \leq 1+\epsilon$, we obtain

$$\Phi_1 = \mathbb{E}\Big[F(\mathbf{x}_1) + \frac{\lambda}{2}\cdot\mathbf{x}_1^\top\mathbf{H}_1\mathbf{x}_1 + \frac{\rho s^{1/3}}{16L^2}\cdot\|\nabla F(\mathbf{x}_1)-\mathbf{m}_1\|_2^2\Big]$$

$$\leq F(\mathbf{x}_1) + \frac{\lambda}{2}D^2(1+\epsilon) + \frac{\rho s^{1/3}}{16L^2}\cdot\mathbb{E}[\|\nabla F(\mathbf{x}_1)-\nabla f(\mathbf{x}_1,\boldsymbol{\xi}_1)\|_2^2]$$

$$\leq F(\mathbf{x}_1) + \frac{\lambda}{2}D^2(1+\epsilon) + \frac{\rho s^{1/3}\sigma^2}{16L^2}.$$

Consequently, defining $G = F(\mathbf{x}_1) - \min_{\mathbf{x}} F(\mathbf{x}) + \frac{\lambda}{2}D^2(1+\epsilon) + \frac{\rho s^{1/3}\sigma^2}{16L^2}$, the following inequality holds:

$$\frac{1}{T}\sum_{t=1}^{T}\Big(\frac{\eta_t}{\rho}\mathbb{E}\|\nabla F(\mathbf{x}_t)-\mathbf{m}_t\|_2^2 + \frac{\rho}{8\eta_t}\cdot\mathbb{E}\|\mathbf{x}_{t+1}-\mathbf{x}_t\|_2^2\Big) \tag{C.14}$$

$$\leq \frac{G}{T} + \frac{\rho c^2\sigma^2}{8L^2 T}\cdot\log(s+T) + \frac{\lambda D^2\log(s+T)}{T} - \frac{1}{T}\sum_{t=1}^{T}\frac{\rho}{16L^2\eta_t}M_{t+1}. \tag{C.15}$$

Dealing with the two terms on the left hand side separately, we have

$$\frac{1}{T}\sum_{t=1}^{T}\mathbb{E}\|\nabla F(\mathbf{x}_t) - \mathbf{m}_t\|_2^2 \leq \frac{\rho(G + \lambda D^2 \log(s+T))}{T\eta_T} + \frac{\rho^2 c^2 \sigma^2}{8L^2 T\eta_T} \cdot \log(s+T) - \frac{\rho^2 \sum_{t=1}^{T} M_{t+1}}{16L^2 T^{1/3}}$$

$$\leq \left(\rho(G + \lambda D^2 \log(s+T)) + \frac{\rho^2 c^2 \sigma^2}{8L^2} \cdot \log(s+T)\right) \cdot \frac{(s+T)^{1/3}}{T} - \frac{\rho^2 \sum_{t=1}^{T} M_{t+1}}{16L^2 T^{1/3}}$$

$$\leq \left(2\rho(G + \lambda D^2 \log(s+T)) + \frac{\rho c^2 \sigma^2}{4L^2} \cdot \log(s+T)\right) \cdot \frac{1}{T^{2/3}} - \frac{\rho^2 \sum_{t=1}^{T} M_{t+1}}{16L^2 T^{1/3}},$$

where the last inequality holds when $T \geq s$. Similarly, for the second term, we also have

$$\frac{1}{T}\sum_{t=1}^{T}\frac{\rho}{8\eta_t} \cdot \mathbb{E}\|\mathbf{x}_{t+1} - \mathbf{x}_t\|_2^2 \leq \frac{G + \lambda D^2 \log(s+T)}{T} + \frac{\rho c^2 \sigma^2}{8L^2 T} \cdot \log(s+T) - \frac{\rho \sum_{t=1}^{T} M_{t+1}}{16L^2 T\eta_t},$$

which implies that when $T \geq s$, due to the definition of $\eta_t$,

$$\frac{1}{T}\sum_{t=1}^{T}\frac{1}{\eta_t^2} \cdot \mathbb{E}\|\mathbf{x}_{t+1} - \mathbf{x}_t\|_2^2 \leq \frac{8(G + \lambda D^2 \log(s+T))}{\rho T\eta_T} + \frac{c^2 \sigma^2}{L^2 T\eta_T} \cdot \log(s+T) - \frac{\sum_{t=1}^{T} M_{t+1}}{2L^2 T\eta_t^2}$$

$$\leq \frac{16(G + \lambda D^2 \log(s+T))}{\rho T^{2/3}} + \frac{2c^2 \sigma^2}{L^2 T^{2/3}} \cdot \log(s+T) - \frac{\sum_{t=1}^{T} M_{t+1}}{L^2 T^{1/3}}.$$

That finishes the proof. $\square$

# D  Proof of Auxiliary Lemmas

## D.1  Lemma D.1 and Proof

**Lemma D.1.** For any sequence $\{\mathbf{m}_t \in \mathbb{R}^d\}_{t=0,1,\ldots}$, consider the following updates of $\mathbf{m}_t$ for any constant factors $a_1, a_2, b_1$, and $b_2$:

$$\mathbf{u}_t = a_1 \mathbf{u}_{t-1} + a_2 \mathbf{g}_t, \tag{D.1}$$

$$\mathbf{m}_t = b_1 \mathbf{u}_t + b_2 \mathbf{g}_t. \tag{D.2}$$

The updates are equivalent to

$$\mathbf{m}_t = a_1 \mathbf{m}_{t-1} + (b_1 a_2 - a_1 b_2 + b_2)\mathbf{g}_t + a_1 b_2 (\mathbf{g}_t - \mathbf{g}_{t-1}).$$

*Proof of Lemma D.1.* Substituting (D.1) into (D.2), we obtain

$$\mathbf{m}_t = b_1 \left(a_1 \mathbf{u}_{t-1} + a_2 \mathbf{g}_t\right) + b_2 \mathbf{g}_t = a_1 b_1 \mathbf{u}_{t-1} + (b_1 a_2 + b_2)\mathbf{g}_t. \tag{D.3}$$

On the other hand, shifting the index of (D.2) by 1, it holds that

$$\mathbf{m}_{t-1} = b_1 \mathbf{u}_{t-1} + b_2 \mathbf{g}_{t-1}. \tag{D.4}$$

Combining (D.3) and (D.4), we obtain the iterative update of $\mathbf{m}_t$ from its previous value $\mathbf{m}_{t-1}$ as:

$$\mathbf{m}_t = a_1 \mathbf{m}_{t-1} - a_1 b_2 \mathbf{g}_{t-1} + (b_1 a_2 + b_2)\mathbf{g}_t$$

$$= a_1 \mathbf{m}_{t-1} + (b_1 a_2 - a_1 b_2 + b_2)\mathbf{g}_t + a_1 b_2 (\mathbf{g}_t - \mathbf{g}_{t-1}).$$

This completes the proof. $\square$

### D.2 Lemma D.2 and Proof

**Lemma D.2.** In Algorithm 2, assume there is a constant $D > 0$ such that $\|\mathbf{x}_t\|_2 \leq D$ for all $t > 0$. Given that $0 \leq \beta_{2,t} \leq 1$ for all $t > 0$, the following inequality holds:

$$\langle \mathbf{m}_t, \mathbf{x}_t - \mathbf{x}_{t+1} \rangle \geq \frac{\rho(1 - \eta_t\lambda)}{\eta_t}\|\mathbf{x}_t - \mathbf{x}_{t+1}\|_2^2 + \frac{\lambda}{2}\big[\mathbf{x}_t^\top\mathbf{H}_{t+1}\mathbf{x}_t - \mathbf{x}_{t+1}^\top\mathbf{H}_t\mathbf{x}_{t+1} - \sqrt{2(1 - \beta_{2,t})}D^2\big]. \quad \text{(D.5)}$$

*Proof of Lemma D.2.* By definition of $\mathbf{m}_t$ and the update rule of $\mathbf{x}_{t+1}$ in Algorithm 2, we have

$$\langle \mathbf{m}_t, \mathbf{x}_t - \mathbf{x}_{t+1} \rangle = \Big\langle \frac{1}{\eta_t} \cdot \mathbf{H}_t[(1 - \eta_t\lambda)\mathbf{x}_t - \mathbf{x}_{t+1}], \mathbf{x}_t - \mathbf{x}_{t+1} \Big\rangle$$

$$\geq \frac{\rho(1 - \eta_t\lambda)}{\eta_t}\|\mathbf{x}_t - \mathbf{x}_{t+1}\|_2^2 - \lambda\langle \mathbf{H}_t\mathbf{x}_{t+1}, \mathbf{x}_t - \mathbf{x}_{t+1} \rangle, \quad \text{(D.6)}$$

where the inequality follows from Assumption B.3. By convexity of $h_1(\mathbf{x}) := \frac{1}{2}\mathbf{x}^\top\mathbf{H}_t\mathbf{x}$, we obtain

$$\langle \mathbf{H}_t\mathbf{x}_{t+1}, \mathbf{x}_t - \mathbf{x}_{t+1} \rangle \leq \frac{1}{2}\mathbf{x}_t^\top\mathbf{H}_t\mathbf{x}_t - \frac{1}{2}\mathbf{x}_{t+1}^\top\mathbf{H}_t\mathbf{x}_{t+1}.$$

Therefore, (D.6) becomes

$$\langle \mathbf{m}_t, \mathbf{x}_t - \mathbf{x}_{t+1} \rangle \geq \frac{\rho(1 - \eta_t\lambda)}{\eta_t}\|\mathbf{x}_t - \mathbf{x}_{t+1}\|_2^2 - \frac{\lambda}{2}\mathbf{x}_t^\top\mathbf{H}_t\mathbf{x}_t + \frac{\lambda}{2}\mathbf{x}_{t+1}^\top\mathbf{H}_t\mathbf{x}_{t+1}$$

$$= \frac{\rho(1 - \eta_t\lambda)}{\eta_t}\|\mathbf{x}_t - \mathbf{x}_{t+1}\|_2^2 - \frac{\lambda}{2}\mathbf{x}_t^\top\mathbf{H}_t\mathbf{x}_t + \frac{\lambda}{2}\mathbf{x}_{t+1}^\top\mathbf{H}_{t+1}\mathbf{x}_{t+1} + \frac{\lambda}{2}\mathbf{x}_{t+1}^\top\big(\mathbf{H}_t - \mathbf{H}_{t+1}\big)\mathbf{x}_{t+1}.$$

We recall that $\mathbf{H}_t = \text{diag}(\sqrt{\widehat{\mathbf{v}}_t} + \epsilon)$ in Algorithm 2. Combining this with Lemma C.1, we derive

$$\mathbf{x}_{t+1}^\top(\mathbf{H}_t - \mathbf{H}_{t+1})\mathbf{x}_{t+1} = \mathbf{x}_{t+1}^\top\text{diag}(\sqrt{\mathbf{v}_t} - \sqrt{\mathbf{v}_{t+1}})\mathbf{x}_{t+1}$$

$$\geq -\sqrt{2(1 - \beta_{2,t})}D^2.$$

Overall, we conclude

$$\langle \mathbf{m}_t, \mathbf{x}_t - \mathbf{x}_{t+1} \rangle \geq \frac{\rho(1 - \eta_t\lambda)}{\eta_t}\|\mathbf{x}_t - \mathbf{x}_{t+1}\|_2^2 + \frac{\lambda}{2}\big[\mathbf{x}_t^\top\mathbf{H}_{t+1}\mathbf{x}_t - \mathbf{x}_{t+1}^\top\mathbf{H}_t\mathbf{x}_{t+1} - \sqrt{2(1 - \beta_{2,t})}D^2\big].$$

$\square$

### D.3 Proof of Lemma C.1

*Proof of Lemma C.1.* According to Algorithm 2, $\widetilde{\mathbf{c}}_t$ is the clipped $\mathbf{c}_t$ with the norm $\|\widetilde{\mathbf{c}}_t\|_2 \leq 1$. Therefore, we can bound $\mathbf{v}_t$ by:

$$\|\mathbf{v}_t\|_2 = \Big\| \sum_{k=1}^{t}(1 - \beta_{2,k})\widetilde{\mathbf{c}}_k^2 \prod_{j=k+1}^{t}\beta_{2,j} + \prod_{j=1}^{t}\beta_{2,j}\mathbf{v}_0 \Big\|_2 \leq \sum_{k=1}^{t}(1 - \beta_{2,k})\prod_{j=k+1}^{t}\beta_{2,j} = \Big(1 - \prod_{k=1}^{t}\beta_{2,k}\Big) \leq 1, \quad \text{(D.7)}$$

where the first inequality is due to $\mathbf{v}_0 = \mathbf{0}$, and the second inequality holds since $0 \leq \beta_{2,k} \leq 1$. We note that when $k = t$, we treat $\prod_{j=t+1}^{t}\beta_{2,j}$ as 1. Similarly, since $\mathbf{m}_0 = \mathbf{0}$ and $0 \leq \beta_{1,t} \leq 1$, we have an upper bound of $\mathbf{m}_t$ as:

$$\|\mathbf{m}_t\|_2 = \Big\| \sum_{k=1}^{t}(1 - \beta_{1,k})\widetilde{\mathbf{c}}_k \prod_{j=k+1}^{t}\beta_{1,j} + \prod_{j=1}^{t}\beta_{1,j}\mathbf{m}_0 \Big\|_2 \leq \sum_{k=1}^{t}(1 - \beta_{1,k})\prod_{j=k+1}^{t}\beta_{1,j} = \Big(1 - \prod_{k=1}^{t}\beta_{1,k}\Big) \leq 1. \quad \text{(D.8)}$$

Therefore, according to the $\mathbf{v}_{t+1}$ update in Algorithm 2, we have

$$\|\mathbf{v}_{t+1} - \mathbf{v}_t\|_\infty = \|(1 - \beta_{2,t})(\widetilde{\mathbf{c}}_{t+1}^2 - \mathbf{v}_t)\|_\infty$$

$$\leq (1 - \beta_{2,t})(\|\widetilde{\mathbf{c}}_{t+1}\|_\infty + \|\mathbf{v}_t\|_\infty)$$

$$\leq (1 - \beta_{2,t})(\|\widetilde{\mathbf{c}}_{t+1}\|_2 + \|\mathbf{v}_t\|_2)$$

$$\leq 2(1 - \beta_{2,t}),$$

where the first inequality is due to triangle inequality and the second inequality derives from that $\|\mathbf{x}\|_\infty \le \|\mathbf{x}\|_2$. Since $|\sqrt{x} - \sqrt{y}| \le \sqrt{|x - y|}, \forall x, y \ge 0$, it holds that

$$\|\sqrt{\mathbf{v}_t} - \sqrt{\mathbf{v}_{t+1}}\|_\infty \le \sqrt{\|\mathbf{v}_{t+1} - \mathbf{v}_t\|_\infty} \le \sqrt{2(1 - \beta_{2,t})}.$$

$\square$

### D.4 Proof of Lemma C.2

*Proof of Lemma C.2.* By the definition of $\mathbf{m}_t$ in Algorithm 1,

$$\mathbf{m}_{t+1} = \beta_{1,t+1}\mathbf{m}_t + (1 - \beta_{1,t+1})\left(\nabla f(\mathbf{x}_{t+1}, \boldsymbol{\xi}_{t+1}) + \gamma_{t+1}\frac{\beta_{1,t+1}}{1 - \beta_{1,t+1}}\left(\nabla f(\mathbf{x}_{t+1}, \boldsymbol{\xi}_{t+1}) - \nabla f(\mathbf{x}_t, \boldsymbol{\xi}_{t+1})\right)\right)$$

$$= (1 - \beta_{1,t+1})\nabla f(\mathbf{x}_{t+1}, \boldsymbol{\xi}_{t+1}) + \beta_{1,t+1}\left(\mathbf{m}_t + \gamma_{t+1}\left(\nabla f(\mathbf{x}_{t+1}, \boldsymbol{\xi}_{t+1}) - \nabla f(\mathbf{x}_t, \boldsymbol{\xi}_{t+1})\right)\right).$$

Subtracting both sides by $\nabla F(\mathbf{x}_{t+1})$, we obtain

$$\mathbf{m}_{t+1} - \nabla F(\mathbf{x}_{t+1})$$

$$= (1 - \beta_{1,t+1})\nabla f(\mathbf{x}_{t+1}, \boldsymbol{\xi}_{t+1}) + \beta_{1,t+1}\left(\mathbf{m}_t + \gamma_{t+1}\left(\nabla f(\mathbf{x}_{t+1}, \boldsymbol{\xi}_{t+1}) - \nabla f(\mathbf{x}_t, \boldsymbol{\xi}_{t+1})\right)\right) - \nabla F(\mathbf{x}_{t+1})$$

$$= \beta_{1,t+1}\left(\mathbf{m}_t - \nabla F(\mathbf{x}_t)\right) + \beta_{1,t+1}\nabla F(\mathbf{x}_t) - \nabla F(\mathbf{x}_{t+1})$$

$$\quad + (1 - \beta_{1,t+1})\nabla f(\mathbf{x}_{t+1}, \boldsymbol{\xi}_{t+1}) + \gamma_{t+1}\beta_{1,t+1}\left(\nabla f(\mathbf{x}_{t+1}, \boldsymbol{\xi}_{t+1}) - \nabla f(\mathbf{x}_t, \boldsymbol{\xi}_{t+1})\right)$$

$$= \beta_{1,t+1}\left(\mathbf{m}_t - \nabla F(\mathbf{x}_t)\right) + (1 - \beta_{1,t+1})\left(\nabla f(\mathbf{x}_{t+1}, \boldsymbol{\xi}_{t+1}) - \nabla F(\mathbf{x}_{t+1})\right)$$

$$\quad + \beta_{1,t+1}\left(\nabla F(\mathbf{x}_t) - \nabla F(\mathbf{x}_{t+1})\right) + \gamma_{t+1}\beta_{1,t+1}\left(\nabla f(\mathbf{x}_{t+1}, \boldsymbol{\xi}_{t+1}) - \nabla f(\mathbf{x}_t, \boldsymbol{\xi}_{t+1})\right).$$

Rearranging the terms, we get

$$\mathbf{m}_{t+1} - \nabla F(\mathbf{x}_{t+1}) = (1 - \beta_{1,t+1})\left(\nabla f(\mathbf{x}_{t+1}, \boldsymbol{\xi}_{t+1}) - \nabla F(\mathbf{x}_{t+1})\right) + \beta_{1,t+1}\left(\mathbf{m}_t - \nabla F(\mathbf{x}_t)\right)$$

$$\quad + \beta_{1,t+1}\left(\nabla F(\mathbf{x}_t) - \nabla F(\mathbf{x}_{t+1}) + \gamma_{t+1}\left(\nabla f(\mathbf{x}_{t+1}, \boldsymbol{\xi}_{t+1}) - \nabla f(\mathbf{x}_t, \boldsymbol{\xi}_{t+1})\right)\right).$$

With a shorthand of notations, we write $\varepsilon_t := \mathbf{m}_t - \nabla F(\mathbf{x}_t)$, and $\boldsymbol{\Delta}_t := \nabla f(\mathbf{x}_{t+1}, \boldsymbol{\xi}_{t+1}) - \nabla f(\mathbf{x}_t, \boldsymbol{\xi}_{t+1})$. The above becomes

$$\varepsilon_{t+1} = (1 - \beta_{1,t+1})\left(\nabla f(\mathbf{x}_{t+1}, \boldsymbol{\xi}_{t+1}) - \nabla F(\mathbf{x}_{t+1})\right) + \beta_{1,t+1}\varepsilon_t + \beta_{1,t+1}\left(\gamma_{t+1}\boldsymbol{\Delta}_t - \mathbb{E}\boldsymbol{\Delta}_t\right), \quad (D.9)$$

where the expectation in the last term is taken over the randomness in $\boldsymbol{\xi}_{t+1}$. Taking squared norm over both sides of (D.9) and then take expectation over $\boldsymbol{\xi}_{t+1}$, we have

$$\mathbb{E}\|\varepsilon_{t+1}\|_2^2 = \mathbb{E}\|(1 - \beta_{1,t+1})\left(\nabla f(\mathbf{x}_{t+1}, \boldsymbol{\xi}_{t+1}) - \nabla F(\mathbf{x}_{t+1})\right) + \beta_{1,t+1}\varepsilon_t + \beta_{1,t+1}\left(\gamma_{t+1}\boldsymbol{\Delta}_t - \mathbb{E}\boldsymbol{\Delta}_t\right)\|_2^2$$

$$= \mathbb{E}\|(1 - \beta_{1,t+1})\left(\nabla f(\mathbf{x}_{t+1}, \boldsymbol{\xi}_{t+1}) - \nabla F(\mathbf{x}_{t+1})\right) + \beta_{1,t+1}\varepsilon_t + \beta_{1,t+1}\left(\boldsymbol{\Delta}_t - \mathbb{E}\boldsymbol{\Delta}_t + (\gamma_{t+1} - 1)\boldsymbol{\Delta}_t\right)\|_2^2$$

$$= \underbrace{\mathbb{E}\|(1 - \beta_{1,t+1})\left(\nabla f(\mathbf{x}_{t+1}, \boldsymbol{\xi}_{t+1}) - \nabla F(\mathbf{x}_{t+1})\right) + \beta_{1,t+1}\varepsilon_t + \beta_{1,t+1}\left(\boldsymbol{\Delta}_t - \mathbb{E}\boldsymbol{\Delta}_t\right)\|_2^2}_{I}$$

$$+ \underbrace{\beta_{1,t+1}^2(\gamma_{t+1} - 1)^2\mathbb{E}\|\boldsymbol{\Delta}_t\|_2^2}_{II}$$

$$- \underbrace{2(1 - \gamma_{t+1})\beta_{1,t+1}\,\mathbb{E}\left\langle \boldsymbol{\Delta}_t, (1 - \beta_{1,t+1})\left(\nabla f(\mathbf{x}_{t+1}, \boldsymbol{\xi}_{t+1}) - \nabla F(\mathbf{x}_{t+1})\right) + \beta_{1,t+1}\varepsilon_t + \beta_{1,t+1}\left(\boldsymbol{\Delta}_t - \mathbb{E}\boldsymbol{\Delta}_t\right)\right\rangle}_{III}.$$

$$(D.10)$$

For $I$, we observe that $\boldsymbol{\varepsilon}_t$ is independent of $\boldsymbol{\xi}_{t+1}$ and the expectations of $\nabla f(\mathbf{x}_{t+1}, \boldsymbol{\xi}_{t+1}) - \nabla F(\mathbf{x}_{t+1})$ and $\boldsymbol{\Delta}_t - \mathbb{E}\boldsymbol{\Delta}_t$ are all 0. Therefore

$$
\begin{aligned}
I &= \mathbb{E}\|(1 - \beta_{1,t+1})\big(\nabla f(\mathbf{x}_{t+1}, \boldsymbol{\xi}_{t+1}) - \nabla F(\mathbf{x}_{t+1})\big) + \beta_{1,t+1}\big(\boldsymbol{\Delta}_t - \mathbb{E}\boldsymbol{\Delta}_t\big)\|_2^2 + \beta_{1,t+1}^2 \mathbb{E}\|\boldsymbol{\varepsilon}_t\|_2^2 \\
&\le 2(1 - \beta_{1,t+1})^2 \mathbb{E}\|\nabla f(\mathbf{x}_{t+1}, \boldsymbol{\xi}_{t+1}) - \nabla F(\mathbf{x}_{t+1})\|_2^2 + 2\beta_{1,t+1}^2 \mathbb{E}\|\boldsymbol{\Delta}_t - \mathbb{E}\boldsymbol{\Delta}_t\|_2^2 + \beta_{1,t+1}^2 \mathbb{E}\|\boldsymbol{\varepsilon}_t\|_2^2.
\end{aligned} \tag{D.11}
$$

Minimizing $II - 2(1 - \gamma_{t+1})\beta_{1,t+1} III$ over $\gamma_{t+1}$ in (D.10), we know that when

$$
1 - \gamma_{t+1} = \frac{\mathbb{E}\Big\langle \boldsymbol{\Delta}_t, (1 - \beta_{1,t+1})\big(\nabla f(\mathbf{x}_{t+1}, \boldsymbol{\xi}_{t+1}) - \nabla F(\mathbf{x}_{t+1})\big) + \beta_{1,t+1}\boldsymbol{\varepsilon}_t + \beta_{1,t+1}\big(\boldsymbol{\Delta}_t - \mathbb{E}\boldsymbol{\Delta}_t\big)\Big\rangle}{\beta_{1,t+1}\mathbb{E}\|\boldsymbol{\Delta}_t\|_2^2},
$$

$II - 2(1 - \gamma_{t+1})\beta_{1,t+1} III$ reaches optimality at

$$
II - 2(1 - \gamma_{t+1})\beta_{1,t+1} III = -\frac{\Big(\mathbb{E}\Big\langle \boldsymbol{\Delta}_t, (1 - \beta_{1,t+1})\big(\nabla f(\mathbf{x}_{t+1}, \boldsymbol{\xi}_{t+1}) - \nabla F(\mathbf{x}_{t+1})\big) + \beta_{1,t+1}\boldsymbol{\varepsilon}_t + \beta_{1,t+1}\big(\boldsymbol{\Delta}_t - \mathbb{E}\boldsymbol{\Delta}_t\big)\Big\rangle\Big)^2}{\mathbb{E}\|\boldsymbol{\Delta}_t\|_2^2}.
$$

Using $G_t$ to represent

$$
G_{t+1} := (1 - \beta_{1,t+1})\mathbb{E}\Big\langle \boldsymbol{\Delta}_t, \nabla f(\mathbf{x}_{t+1}, \boldsymbol{\xi}_{t+1}) - \nabla F(\mathbf{x}_{t+1})\Big\rangle + \beta_{1,t+1}\mathbb{E}\Big\langle \boldsymbol{\Delta}_t, \boldsymbol{\varepsilon}_t\Big\rangle,
$$

and defining

$$
A_{t+1} := \frac{\Big(G_{t+1} + \beta_{1,t+1}\big(\mathbb{E}\|\boldsymbol{\Delta}_t\|_2^2 - \|\mathbb{E}\boldsymbol{\Delta}_t\|_2^2\big)\Big)}{\mathbb{E}\|\boldsymbol{\Delta}_t\|_2^2}.
$$

We have

$$
II - 2(1 - \gamma_{t+1})\beta_{1,t+1} III = \mathbb{E}\|\boldsymbol{\Delta}_t\|_2^2 \bigg(\beta_{1,t+1}(1 - \gamma_{t+1}) - A_{t+1}\bigg)^2 - \mathbb{E}\|\boldsymbol{\Delta}_t\|_2^2 A_{t+1}^2
$$

and when

$$
\gamma_{t+1} = 1 - \frac{\Big(G_{t+1} + \beta_{1,t+1}\big(\mathbb{E}\|\boldsymbol{\Delta}_t\|_2^2 - \|\mathbb{E}\boldsymbol{\Delta}_t\|_2^2\big)\Big)}{\beta_{1,t+1}\mathbb{E}\|\boldsymbol{\Delta}_t\|_2^2} = \frac{\Big(\|\mathbb{E}\boldsymbol{\Delta}_t\|_2^2\big) - G_{t+1}\Big)}{\beta_{1,t+1}\mathbb{E}\|\boldsymbol{\Delta}_t\|_2^2}. \tag{D.12}
$$

$$
II - 2(1 - \gamma_{t+1})\beta_{1,t+1} III = -\mathbb{E}\|\boldsymbol{\Delta}_t\|_2^2 A_{t+1}^2.
$$

Defining

$$
M_{t+1} := \mathbb{E}\|\boldsymbol{\Delta}_t\|_2^2 A_{t+1}^2 - \mathbb{E}\|\boldsymbol{\Delta}_t\|_2^2 \bigg(\beta_{1,t+1}(1 - \gamma_{t+1}) - A_{t+1}\bigg)^2
$$

we conclude that

$$
\begin{aligned}
&\min_{\{\gamma_i\}_{i=1,\dots,t}} \mathbb{E}\|\boldsymbol{\varepsilon}_{t+1}\|_2^2 \\
&\le \mathbb{E}\|I\|_2^2 - M_{t+1} \\
&\le 2(1 - \beta_{1,t+1})^2 \mathbb{E}\|\nabla f(\mathbf{x}_{t+1}, \boldsymbol{\xi}_{t+1}) - \nabla F(\mathbf{x}_{t+1})\|_2^2 + 2\beta_{1,t+1}^2 \mathbb{E}\|\boldsymbol{\Delta}_t - \mathbb{E}\boldsymbol{\Delta}_t\|_2^2 + \beta_{1,t+1}^2 \mathbb{E}\|\boldsymbol{\varepsilon}_t\|_2^2 - M_{t+1} \\
&\le 2(1 - \beta_{1,t+1})^2 \sigma^2 + 2\beta_{1,t+1}^2 L^2 \|\mathbf{x}_{t+1} - \mathbf{x}_t\|_2^2 + \beta_{1,t+1}^2 \mathbb{E}\|\boldsymbol{\varepsilon}_t\|_2^2 - M_{t+1}.
\end{aligned}
$$

where in the last inequality we utilized the fact that $\mathbb{E}\|\boldsymbol{\Delta}_t - \mathbb{E}\boldsymbol{\Delta}_t\|_2^2 \le \mathbb{E}\|\boldsymbol{\Delta}_t\|_2^2$ and the $L$-smoothness of $f$. Rearranging the terms finishes the proof. $\qquad\square$

### D.5 Proof of Lemma C.3 and C.4

*Proof of Lemma C.3.* Given the L-smoothness of $F(\mathbf{x})$ in Assuption B.2, we have

$$F(\mathbf{x}_{t+1}) \leq F(\mathbf{x}_t) + \langle \nabla F(\mathbf{x}_t), \mathbf{x}_{t+1} - \mathbf{x}_t \rangle + \frac{L}{2} \cdot \|\mathbf{x}_{t+1} - \mathbf{x}_t\|_2^2$$

$$= F(\mathbf{x}_t) + \langle \mathbf{m}_t, \mathbf{x}_{t+1} - \mathbf{x}_t \rangle + \langle \nabla F(\mathbf{x}_t) - \mathbf{m}_t, \mathbf{x}_{t+1} - \mathbf{x}_t \rangle + \frac{L}{2} \cdot \|\mathbf{x}_{t+1} - \mathbf{x}_t\|_2^2. \qquad (\text{D.13})$$

By definition of $\mathbf{m}_t$ and the update rule of $\mathbf{x}_{t+1}$ in Algorithm 1, we have

$$\langle \mathbf{m}_t, \mathbf{x}_t - \mathbf{x}_{t+1} \rangle = \langle \frac{1}{\eta_t} \cdot \mathbf{H}_t[\mathbf{x}_t - \mathbf{x}_{t+1}], \mathbf{x}_t - \mathbf{x}_{t+1} \rangle$$

$$\geq \frac{\rho}{\eta_t} \|\mathbf{x}_t - \mathbf{x}_{t+1}\|_2^2. \qquad (\text{D.14})$$

Here the inequality follows from Assumption B.3. $\qquad \square$

*Proof of Lemma C.4.* Bringing (D.14) into (D.13), we obtain

$$F(\mathbf{x}_{t+1}) \leq F(\mathbf{x}_t) - \frac{\rho}{\eta_t} \|\mathbf{x}_t - \mathbf{x}_{t+1}\|_2^2 + \langle \nabla F(\mathbf{x}_t) - \mathbf{m}_t, \mathbf{x}_{t+1} - \mathbf{x}_t \rangle + \frac{L}{2} \cdot \|\mathbf{x}_{t+1} - \mathbf{x}_t\|_2^2$$

$$\leq F(\mathbf{x}_t) - \frac{\rho}{\eta_t} \|\mathbf{x}_t - \mathbf{x}_{t+1}\|_2^2 + \frac{\eta_t}{\rho} \|\nabla F(\mathbf{x}_t) - \mathbf{m}_t\|_2^2 + \frac{\rho}{4\eta_t} \|\mathbf{x}_{t+1} - \mathbf{x}_t\|_2^2 + \frac{L}{2} \cdot \|\mathbf{x}_{t+1} - \mathbf{x}_t\|_2^2$$

$$\leq F(\mathbf{x}_t) - \frac{\rho}{2\eta_t} \|\mathbf{x}_t - \mathbf{x}_{t+1}\|_2^2 + \frac{\eta_t}{\rho} \|\nabla F(\mathbf{x}_t) - \mathbf{m}_t\|_2^2,$$

The second inequality follows from applying both the Cauchy-Schwarz and Young's inequalities. Moreover, the final inequality results from selecting $\eta_t$ to satisfy $\eta_t < \frac{\rho}{2L}$. This completes our proof.

Bringing (D.5) in Lemma D.2 into (D.13), we have

$$F(\mathbf{x}_{t+1}) \leq F(\mathbf{x}_t) + \langle \mathbf{m}_t, \mathbf{x}_{t+1} - \mathbf{x}_t \rangle + \langle \nabla F(\mathbf{x}_t) - \mathbf{m}_t, \mathbf{x}_{t+1} - \mathbf{x}_t \rangle + \frac{L}{2} \cdot \|\mathbf{x}_{t+1} - \mathbf{x}_t\|_2^2$$

$$\leq F(\mathbf{x}_t) + \langle \nabla F(\mathbf{x}_t) - \mathbf{m}_t, \mathbf{x}_{t+1} - \mathbf{x}_t \rangle + \frac{L}{2} \cdot \|\mathbf{x}_{t+1} - \mathbf{x}_t\|_2^2$$

$$- \frac{\rho(1 - \eta_t \lambda)}{\eta_t} \|\mathbf{x}_t - \mathbf{x}_{t+1}\|_2^2 - \frac{\lambda}{2} \big[ \mathbf{x}_t^\top \mathbf{H}_{t+1} \mathbf{x}_t - \mathbf{x}_{t+1}^\top \mathbf{H}_t \mathbf{x}_{t+1} - \sqrt{2(1 - \beta_{2,t})} D^2 \big].$$

Taking $\eta_t < \min\{(4\lambda)^{-1}, \rho \cdot (2L)^{-1}\}$, we have

$$- \frac{\rho(1 - \eta_t \lambda)}{\eta_t} \|\mathbf{x}_t - \mathbf{x}_{t+1}\|_2^2 + \frac{L}{2} \cdot \|\mathbf{x}_{t+1} - \mathbf{x}_t\|_2^2 \leq \Big( -\frac{3\rho}{4\eta_t} + \frac{L}{2} \Big) \|\mathbf{x}_{t+1} - \mathbf{x}_t\|_2^2 \leq -\frac{\rho}{2\eta_t} \|\mathbf{x}_{t+1} - \mathbf{x}_t\|_2^2.$$

Therefore,

$$F(\mathbf{x}_{t+1}) \leq F(\mathbf{x}_t) + \langle \nabla F(\mathbf{x}_t) - \mathbf{m}_t, \mathbf{x}_{t+1} - \mathbf{x}_t \rangle - \frac{\rho}{2\eta_t} \|\mathbf{x}_t - \mathbf{x}_{t+1}\|_2^2$$

$$- \frac{\lambda}{2} \big[ \mathbf{x}_t^\top \mathbf{H}_{t+1} \mathbf{x}_t - \mathbf{x}_{t+1}^\top \mathbf{H}_t \mathbf{x}_{t+1} - \sqrt{2(1 - \beta_{2,t})} D^2 \big].$$

By Cauchy-Schwarz inequality and Young's inequality, we have

$$\langle \nabla F(\mathbf{x}_t) - \mathbf{m}_t, \mathbf{x}_{t+1} - \mathbf{x}_t \rangle \leq \|\nabla F(\mathbf{x}_t) - \mathbf{m}_t\|_2 \cdot \|\mathbf{x}_{t+1} - \mathbf{x}_t\|_2 \leq \frac{\eta_t}{\rho} \|\nabla F(\mathbf{x}_t) - \mathbf{m}_t\|_2^2 + \frac{\rho}{4\eta_t} \|\mathbf{x}_{t+1} - \mathbf{x}_t\|_2^2.$$

Therefore, we conclude that

$$F(\mathbf{x}_{t+1}) \leq F(\mathbf{x}_t) + \frac{\eta_t}{\rho} \|\nabla F(\mathbf{x}_t) - \mathbf{m}_t\|_2^2 + \frac{\rho}{4\eta_t} \|\mathbf{x}_{t+1} - \mathbf{x}_t\|_2^2 - \frac{\rho}{2\eta_t} \|\mathbf{x}_t - \mathbf{x}_{t+1}\|_2^2$$

$$- \frac{\lambda}{2} \big[ \mathbf{x}_t^\top \mathbf{H}_{t+1} \mathbf{x}_t - \mathbf{x}_{t+1}^\top \mathbf{H}_t \mathbf{x}_{t+1} - \sqrt{2(1 - \beta_{2,t})} D^2 \big].$$

$$= F(\mathbf{x}_t) + \frac{\eta_t}{\rho} \|\nabla F(\mathbf{x}_t) - \mathbf{m}_t\|_2^2 - \frac{\rho}{4\eta_t} \|\mathbf{x}_{t+1} - \mathbf{x}_t\|_2^2 - \frac{\lambda}{2} \big[ \mathbf{x}_t^\top \mathbf{H}_{t+1} \mathbf{x}_t - \mathbf{x}_{t+1}^\top \mathbf{H}_t \mathbf{x}_{t+1} \big] + \frac{\lambda}{2} \sqrt{2(1 - \beta_{2,t})} D^2.$$

Rearranging terms finishes the proof. $\qquad \square$

## D.6 Proof of Lemma C.5

*Proof of Lemma C.5.* By the definition of $\eta_t$, it holds that

$$\frac{1}{\eta_t} - \frac{1}{\eta_{t-1}} = (s+t)^{1/3} - (s+t-1)^{1/3} \leq \frac{1}{3(s+t-1)^{2/3}} \leq \frac{1}{(s+t)^{2/3}} = \eta_t^2 \leq \eta_t,$$

where the first inequality follows by the concavity of $h_2(\mathbf{x}) = x^{1/3}$, and the second inequality follows by $s \geq 1$, which implied that $s + t \geq 2$ and $27(s+t-1)^2 \geq (s+t)^2$. This finishes the proof. □

# E   Additional Experiment Results

## E.1   Supplementary Results for the Main Experiments

Here we display the supplementary results for the experiments in Section 4. The training and validation losses as well as wall-clock time curves for small and medium models are displayed in Figures 2 and 3. And the 0-shot and 5-shot evaluation results on different downstream tasks for small, medium and large models are listed in Tables 2,3,4 and Tables 5,6, respectively. It can be observed that different sizes of models trained with MARS-AdamW and MARS-Lion can achieve better performances than baseline optimization methods with respect to cross-entropy loss, time efficiency as well as downstream task performances.

*Table 2.* The evaluation results of small models pre-trained using the OpenWebText dataset (0-shot with lm-evaluation-harness). The best scores in each column are **bolded**. Abbreviations: HellaSwag = HellaSwag, WG = WinoGrande.

| METHOD | ARC-E | ARC-C | BOOLQ | HELLASWAG | OBQA | PIQA | WG | MMLU | SCIQ | AVG. |
|---|---|---|---|---|---|---|---|---|---|---|
| ADAMW | **41.37** | 22.27 | 55.02 | 31.73 | 27.80 | **63.00** | **52.01** | 22.97 | **67.50** | 42.63 |
| LION | 40.15 | 21.93 | **59.72** | 31.72 | 26.00 | 62.95 | 51.07 | 22.92 | 64.80 | 42.36 |
| MUON | 39.73 | 23.55 | 57.31 | 30.84 | 25.00 | 61.48 | 50.36 | 22.89 | 62.70 | 41.54 |
| MARS-ADAMW | 40.70 | 23.63 | 59.17 | **32.46** | 27.00 | 61.92 | 51.22 | **22.98** | 67.40 | **42.94** |
| MARS-LION | 40.78 | **23.72** | 51.74 | 31.59 | **29.20** | 62.68 | 51.30 | 22.94 | 65.50 | 42.16 |

*Table 3.* The evaluation results of medium models pre-trained using the OpenWebText dataset (0-shot with lm-evaluation-harness). The best scores in each column are **bolded**. Abbreviations: HellaSwag = HellaSwag, WG = WinoGrande.

| METHOD | ARC-E | ARC-C | BOOLQ | HELLASWAG | OBQA | PIQA | WG | MMLU | SCIQ | AVG. |
|---|---|---|---|---|---|---|---|---|---|---|
| ADAMW | 43.43 | 23.98 | 58.13 | 37.76 | 27.20 | 65.56 | 52.49 | 22.80 | 67.60 | 44.33 |
| LION | 44.11 | 25.43 | **60.06** | 37.64 | **31.40** | 66.05 | 53.20 | 22.97 | 69.50 | **45.60** |
| MUON | 43.01 | 24.57 | 58.93 | 35.85 | 30.60 | 64.85 | 51.54 | 22.89 | 66.70 | 44.33 |
| MARS-ADAMW | 43.94 | **25.85** | 54.50 | **39.88** | 30.60 | **66.87** | 52.01 | 22.97 | **72.10** | 45.41 |
| MARS-LION | **45.33** | 24.74 | 55.84 | 38.80 | 30.60 | 64.96 | **53.83** | **23.33** | 68.70 | 45.13 |

*Table 4.* The evaluation results of large models pre-trained using the OpenWebText dataset (0-shot with lm-evaluation-harness). The best scores in each column are **bolded**. Abbreviations: HellaSwag = HellaSwag, WG = WinoGrande.

| METHOD | ARC-E | ARC-C | BOOLQ | HELLASWAG | OBQA | PIQA | WG | MMLU | SCIQ | AVG. |
|---|---|---|---|---|---|---|---|---|---|---|
| ADAMW | 46.30 | 26.19 | 59.91 | 41.70 | 31.40 | 68.12 | 51.46 | 23.10 | 72.80 | 46.78 |
| LION | 47.73 | 26.45 | 57.09 | 42.43 | 30.20 | 68.01 | 54.38 | 23.41 | **74.00** | 47.08 |
| MUON | 45.45 | 26.37 | 59.69 | 40.28 | 31.00 | 67.08 | 52.41 | 23.26 | 66.70 | 45.80 |
| MARS-ADAMW | **48.11** | 25.77 | **62.26** | **44.64** | **32.60** | 68.06 | **56.04** | 23.98 | 73.00 | **48.27** |
| MARS-LION | 47.77 | **26.71** | 59.45 | 43.07 | 31.20 | **68.39** | 55.72 | **24.53** | 72.50 | 47.70 |

*Table 5.* The evaluation results of small models pre-trained using the OpenWebText dataset (5-shot with lm-evaluation-harness). The best scores in each column are **bolded**. Abbreviations: HellaSwag = HellaSwag, WG = WinoGrande.

| METHOD | ARC-E | ARC-C | BOOLQ | HELLASWAG | OBQA | PIQA | WG | MMLU | SCIQ | AVG. |
|---|---|---|---|---|---|---|---|---|---|---|
| ADAMW | 41.75 | 22.78 | 54.04 | 32.33 | **28.20** | 63.38 | 52.57 | **26.88** | 76.00 | 44.21 |
| LION | 42.21 | 22.70 | 55.41 | 31.82 | 24.80 | 62.40 | **53.04** | 24.63 | 74.80 | 43.53 |
| MUON | 41.50 | 23.46 | 48.78 | 30.48 | 24.60 | 61.26 | 52.01 | 24.63 | 67.20 | 41.55 |
| MARS-ADAMW | **45.24** | **24.66** | **56.97** | **32.76** | 25.60 | 62.40 | 50.43 | 25.78 | **76.70** | **44.51** |
| MARS-LION | 43.06 | 22.78 | 55.66 | 32.17 | 26.20 | 62.24 | 50.59 | 25.32 | 72.80 | 43.42 |

## E.2   MARS and MARS-approx.

We then conduct experiments to compare the performance of MARS and MARS-approx (MARS-AdamW instantiation) on GPT-2 small and medium models, the training and validation loss curves are shown in Figures 4 and 5. Models trained with MARS exhibit consistently better performance than those trained with MARS-approx. This suggests that: (a) The exact version, which employs the variance reduction formulation, is more fundamental than the approximate version. (b) The

*Table 6.* The evaluation results of medium models pre-trained using the OpenWebText dataset (5-shot with lm-evaluation-harness). The best scores in each column are **bolded**. Abbreviations: HellaSwag = HellaSwag, WG = WinoGrande.

| METHOD | ARC-E | ARC-C | BOOLQ | HELLASWAG | OBQA | PIQA | WG | MMLU | SCIQ | AVG. |
|---|---|---|---|---|---|---|---|---|---|---|
| ADAMW | 48.23 | 25.43 | 45.26 | 38.32 | 27.60 | 65.83 | 52.33 | 26.21 | 80.90 | 45.57 |
| LION | **49.16** | 24.49 | 58.32 | 38.09 | 30.00 | **66.05** | 51.22 | **26.43** | 81.20 | 47.22 |
| MUON | 47.56 | 24.49 | **58.56** | 36.10 | 29.20 | 65.13 | 52.72 | 25.15 | 73.10 | 45.78 |
| MARS-ADAMW | 48.99 | 25.60 | 52.11 | **40.02** | **30.80** | 65.56 | **54.30** | 25.49 | **83.50** | **47.37** |
| MARS-LION | 49.03 | **25.77** | 51.59 | 39.11 | 29.80 | 65.51 | 53.59 | 24.85 | 81.40 | 46.74 |

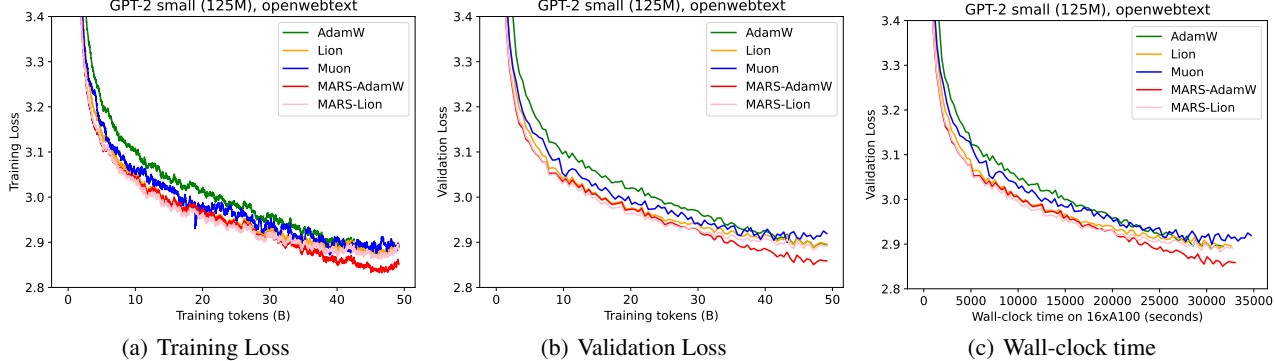

(a) Training Loss   (b) Validation Loss   (c) Wall-clock time

*Figure 2.* The training and validation loss curves, plotted against both training tokens and wall-clock time on GPT-2 small model (125M).

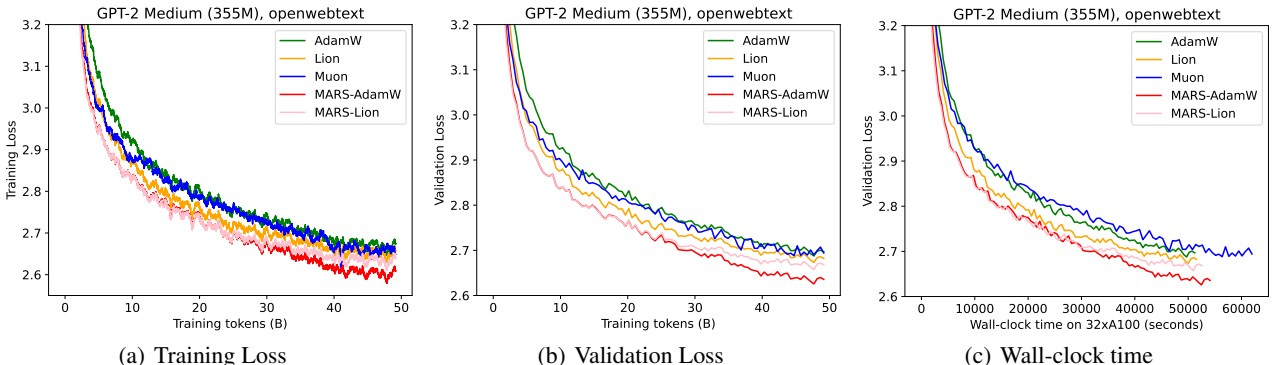

(a) Training Loss   (b) Validation Loss   (c) Wall-clock time

*Figure 3.* The training and validation loss curves, plotted against both training tokens and wall-clock time on GPT-2 medium model (355M).

approximate version serves as a practical alternative in scenarios where computational efficiency is a priority, as it incurs only minimal performance loss. However, in settings where maximizing validation accuracy is crucial, the exact version is recommended.

### E.3 Experiments on FineWeb-Edu 100B Dataset

FineWeb-Edu dataset (Lozhkov et al., 2024) is a high-quality dataset based on well-filtered educational web pages. To better investigate the efficiency of our algorithm, we also use FineWeb-Edu 100B, a subset of FineWeb-Edu with around 100B tokens to train GPT-2 small (125M) and XL (1.5B, with the same learning rates as GPT-2 large models) models with optimizers including AdamW, Muon and MARS-AdamW-approx. We leave around 0.1B tokens for validation and other tokens for training. The training and evaluation curves are shown in Figures 6 and 7. It can be seen that our algorithms can also achieve better performances even with different datasets. For a comprehensive investigation, we evaluate these models on metrics same as experiments in Section 4.2, and the results are shown in Tables 7 and 8. We also compare the results with the open-source GPT-2 models on Hugging Face (Radford et al., 2019) (denoted as "OpenAI-Comm." in the tables). Compared with Table 2, it can be observed that this dataset is actually better for the superior performances. However, models trained with our algorithm can also show advantages over baseline optimization approaches trained with such a high-quality dataset.

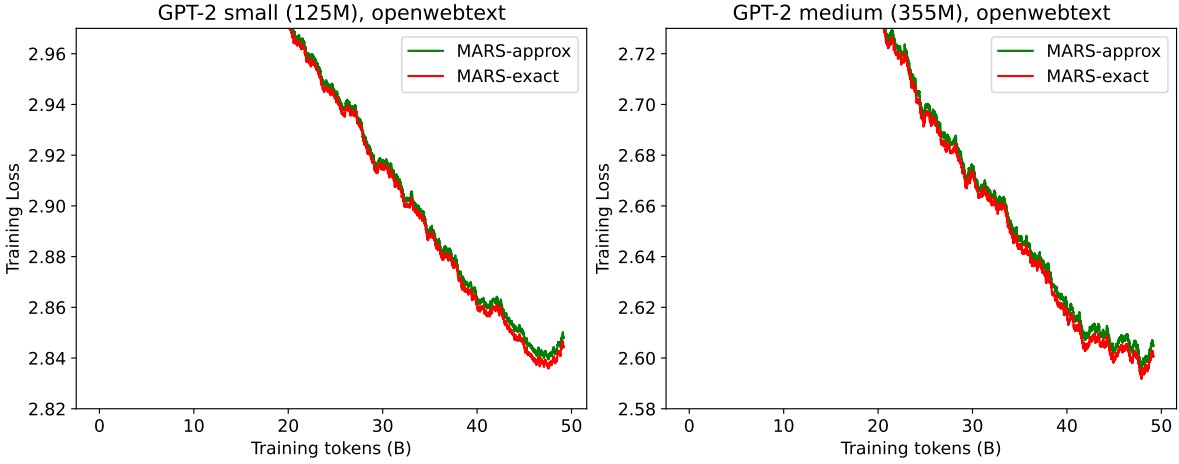

*Figure 4.* Training loss curves for MARS-AdamW and MARS-AdamW-approx on GPT-2 small (125M, left) and medium (355M, right), pretrained with OpenWebText dataset and plotted against training tokens.

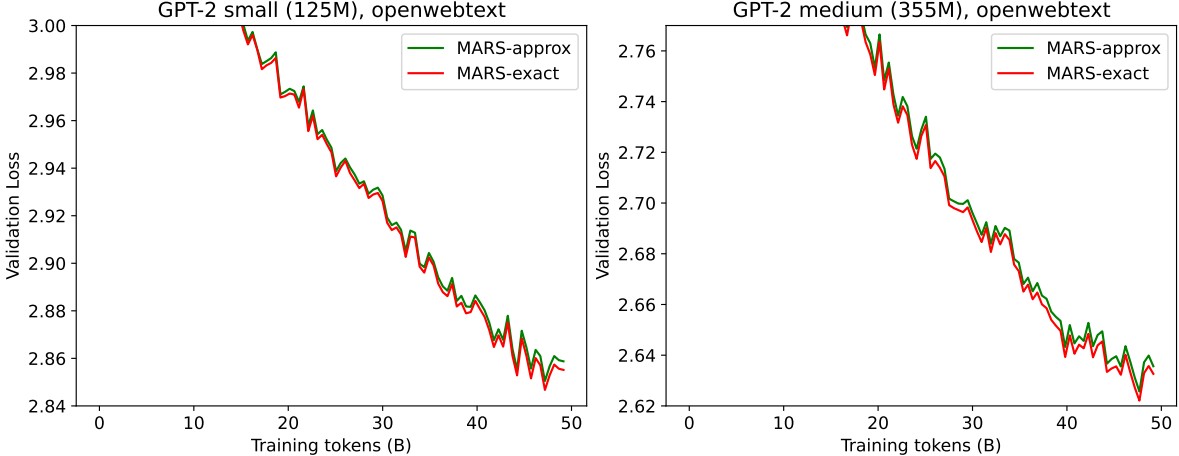

*Figure 5.* Validation loss curves for MARS-AdamW and MARS-AdamW-approx on GPT-2 small (125M, left) and medium (355M, right), pretrained with OpenWebText dataset.

*Table 7.* The evaluation results of small models pre-trained using the FineWeb-Edu 100B dataset (0-shot with lm-evaluation-harness). The best scores in each column are **bolded**. Abbreviations: HellaSwag = HellaSwag, WG = WinoGrande.

| METHOD | ARC-E | ARC-C | BOOLQ | HELLASWAG | OBQA | PIQA | WG | MMLU | SCIQ | AVG. |
|---|---|---|---|---|---|---|---|---|---|---|
| OPENAI-COMM. | 39.48 | 22.70 | 48.72 | 31.14 | 27.20 | 62.51 | **51.62** | 22.92 | 64.40 | 41.19 |
| ADAMW | 51.43 | 26.54 | 55.78 | 36.26 | 30.60 | 64.53 | 50.36 | **24.49** | **71.50** | 45.72 |
| MUON | 47.85 | **27.56** | **57.16** | 33.46 | 31.60 | 63.66 | 51.30 | 23.17 | 67.30 | 44.78 |
| MARS-ADAMW | **52.23** | 27.39 | 55.84 | **36.91** | **32.20** | **64.80** | 49.96 | 22.95 | 71.10 | **45.93** |

*Table 8.* The evaluation results of XL models pre-trained using the FineWeb-Edu 100B dataset (0-shot with lm-evaluation-harness). The best scores in each column are **bolded**. Abbreviations: HellaSwag = HellaSwag, WG = WinoGrande.

| METHOD | ARC-E | ARC-C | BOOLQ | HELLASWAG | OBQA | PIQA | WG | MMLU | SCIQ | AVG. |
|---|---|---|---|---|---|---|---|---|---|---|
| OPENAI-COMM. | 51.05 | 28.50 | 61.77 | 50.89 | 32.00 | 70.51 | **58.33** | 25.24 | 76.00 | 50.48 |
| ADAMW | **68.22** | 38.40 | 61.13 | 53.93 | 39.00 | 72.69 | 54.78 | **25.47** | 85.30 | 55.43 |
| MUON | 64.18 | 36.52 | 58.38 | 51.83 | 37.40 | 72.03 | 55.56 | 24.93 | 81.90 | 53.64 |
| MARS-ADAMW | 66.54 | **39.85** | **63.82** | **56.52** | **41.20** | **73.34** | 56.59 | 23.86 | **86.00** | **56.41** |

### E.4   Computer Vision Experiments

We also carry out experiments on the classification task in the field of computer vision. We conduct experiments with ResNet-18 model (He et al., 2016) on the CIFAR-10 and CIFAR-100 datasets (Krizhevsky et al., 2009) for AdamW, Lion,

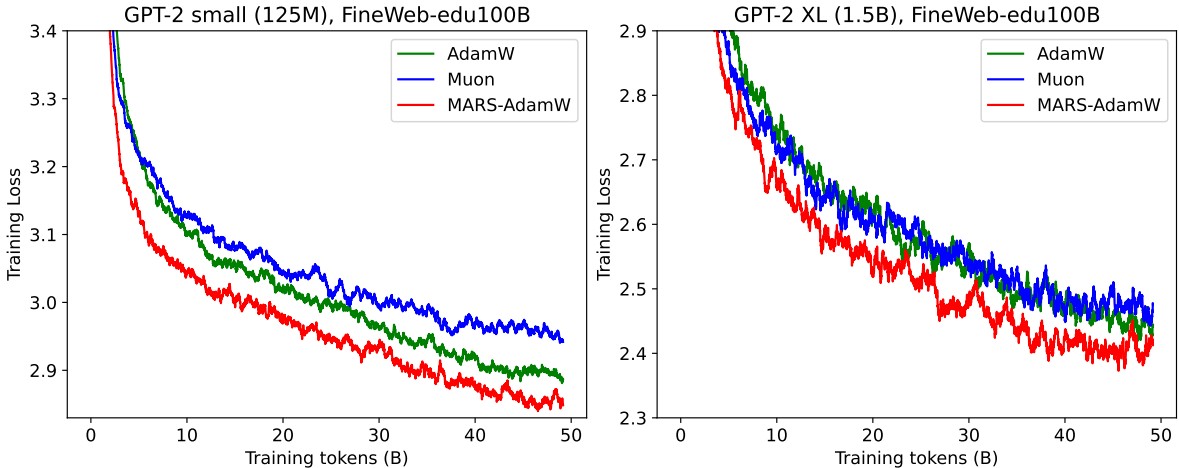

*Figure 6.* Training loss curves for AdamW, Muon and MARS-AdamW-approx on GPT-2 small (125M, left) and XL (1.5B, right), pretrained with FineWeb-edu 100B dataset and plotted against training tokens.

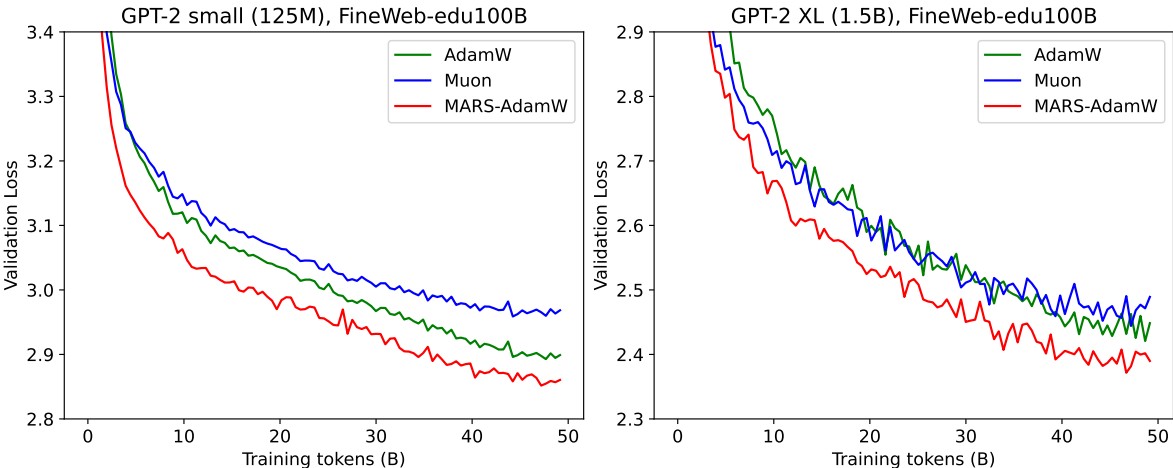

*Figure 7.* Validation loss curves for AdamW, Muon and MARS-AdamW-approx on GPT-2 small (125M, left) and XL (1.5B, right), pretrained with FineWeb-edu 100B dataset and plotted against training tokens.

Shampoo[1], Muon, and variants of MARS instantiation, following the setting in Chen et al. (2018).

We do grid search to explore the best hyper-parameters for each of these optimization methods. We search over $\{10^{-5}, ..., 10^0\}$ for the learning rate and $\{0, ..., 1.0\}$ for the weight decay. We set $\beta_1 = 0.9$ and search over $\{0.99, 0.999\}$ for $\beta_2$ for AdamW and Muon; fix $\beta_1 = 0.9$ and search $\beta_2$ over $\{0.99, 0.999\}$ for Lion; search over $\{0.9, 0.95\}$ for $\beta_1$ and $\{0.95, 0.99, 0.999\}$ for $\beta_2$ for Shampoo; and we fix the $(\beta_1, \beta_2) = (0.95, 0.99)$ and $\gamma = 0.025$ for MARS models. We train for 200 epochs with training batch size 128 on 1 NVIDIA A6000 GPU. And we also apply MultiStepLR scheduler so that the learning rate would decrease to 10% of the original rate at the 100th epoch and to 1% at the 150th epoch. We display the test loss and test accuracy for CIFAR-10 and CIFAR-100 datasets in Figures 8 and 9, respectively. The results show that our algorithm can achieve better validation loss after the decay of learning rate and better test accuracy within the final stage of training.

We also compare the performances among the baselines of AdamW, Lion and Shampoo without variance reduction, the approximate and the exact versions of MARS instantiations. The test loss and accuracy curves for CIFAR10 dataset are shown in Figures, and Figures, respectively. And the test loss and accuracy curves for CIFAR100 dataset are shown in Figures 10–11, and Figures 12–13, respectively. Moreover, we list the best test losses and accuracies in Table 9. It can be observed that the exact versions perform a little better than the approximate versions, but much better than the baseline approaches, showing the superiority of variance reduction in MARS.

---

[1]In practice, we use Distributed Shampoo (Shi et al., 2023) to facilitate the training of Shampoo.

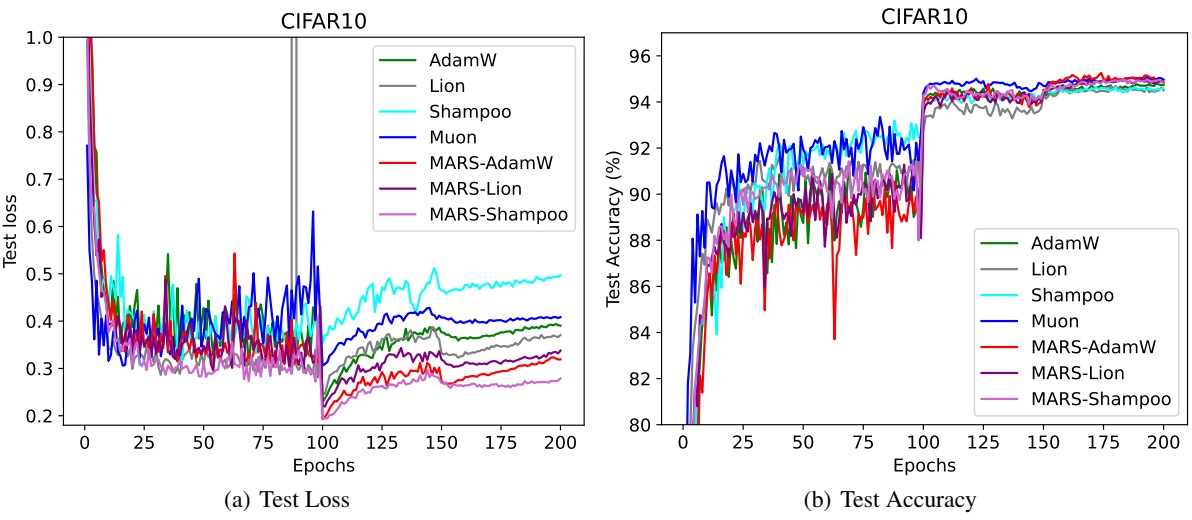

(a) Test Loss

(b) Test Accuracy

*Figure 8.* The test loss and test accuracy for different optimizers on CIFAR-10 dataset.

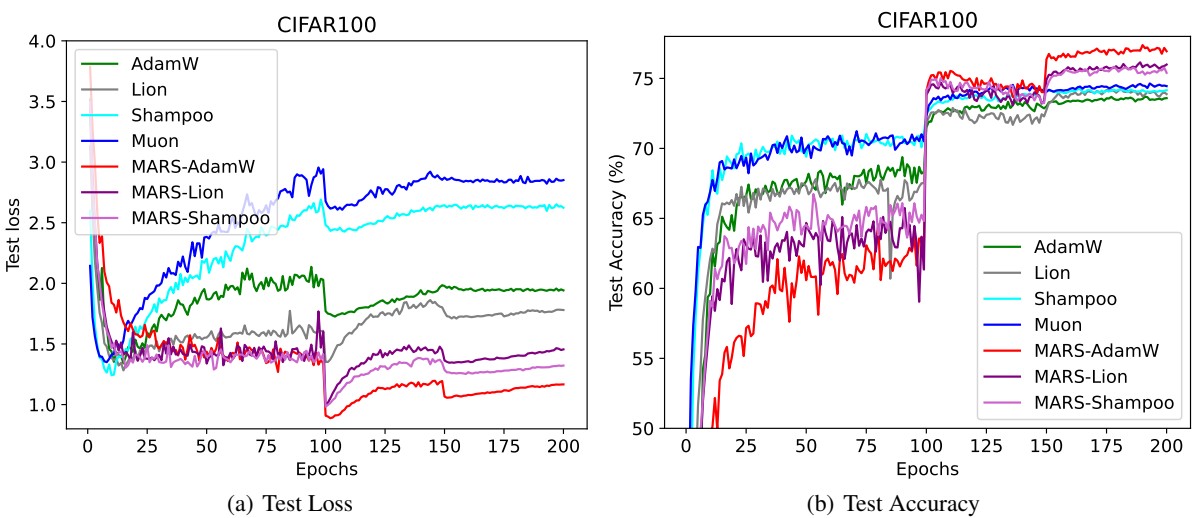

(a) Test Loss

(b) Test Accuracy

*Figure 9.* The test loss and test accuracy for different optimizers on CIFAR-100 dataset.

*Table 9.* The best test losses and accuracies of different optimizers on CIFAR10 and CIFAR100 datasets during the training epochs. The **bolded** are the best test loss or best test accuracy among the listed optimizers.

| Optimizer | Best Test Loss | | Best Test Accuracy | |
|---|---|---|---|---|
| | CIFAR10 | CIFAR100 | CIFAR10 | CIFAR100 |
| AdamW | 0.230 | 1.726 | 94.81 | 73.70 |
| Lion | 0.245 | 1.351 | 94.68 | 74.28 |
| Shampoo | 0.354 | 2.426 | 94.65 | 74.27 |
| Muon | 0.306 | 2.608 | 95.08 | 74.64 |
| MARS-AdamW-approx | 0.199 | 0.971 | **95.29** | 76.97 |
| MARS-AdamW | **0.193** | **0.888** | 95.26 | **77.38** |
| MARS-Lion-approx | 0.202 | 0.985 | 95.05 | 75.97 |
| MARS-Lion | 0.219 | 0.991 | 94.98 | 76.15 |
| MARS-Shampoo-approx | 0.202 | 1.256 | 94.92 | 74.80 |
| MARS-Shampoo | 0.194 | 0.982 | 94.98 | 75.83 |

## E.5 Sensitivity to $\gamma$.

To explore the impact of $\gamma_t$, we test various $\gamma$s on GPT-2 small model, including constant and linearly changing schedules. And we plot the training and validation curves in Figure 14. It can be observed that there are slight differences among

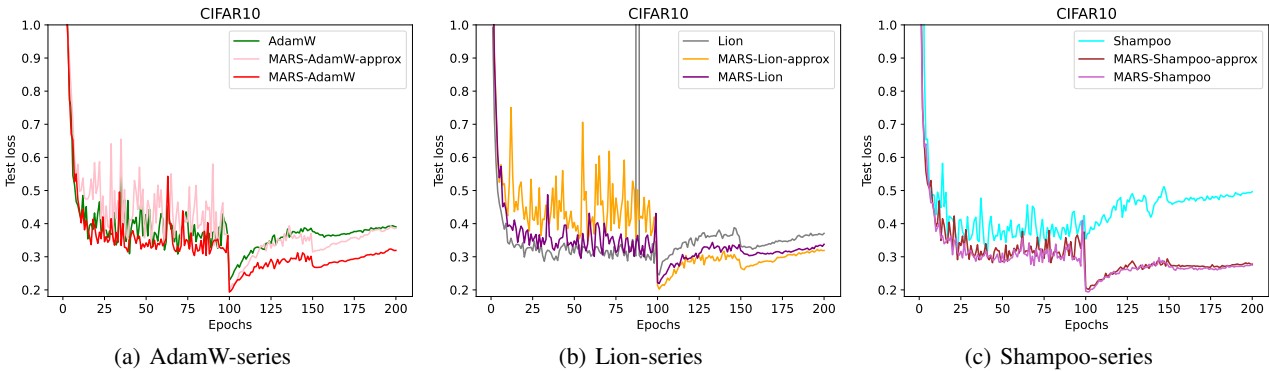

(a) AdamW-series      (b) Lion-series      (c) Shampoo-series

*Figure 10.* The test loss curves for the baselines of AdamW, Lion and Shampoo without variance reduction, the approximate and the exact versions of MARS instantiations on CIFAR-10 dataset.

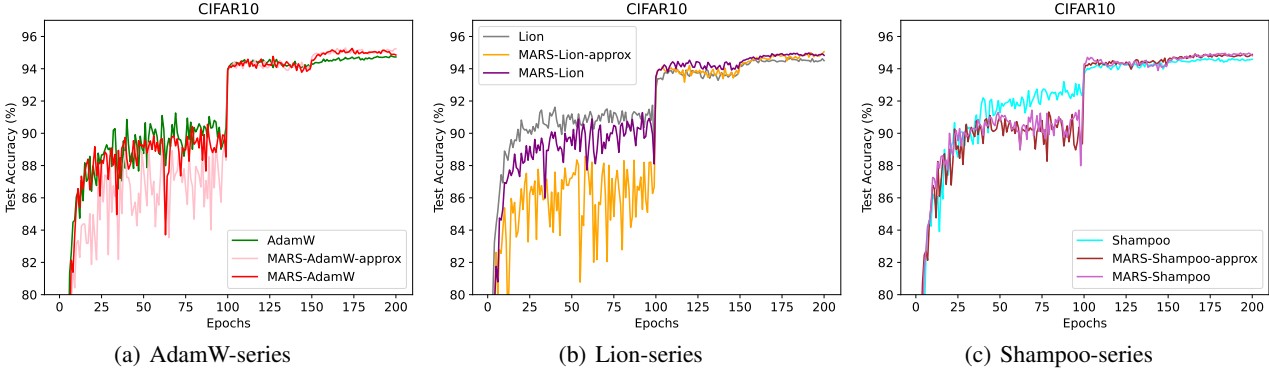

(a) AdamW-series      (b) Lion-series      (c) Shampoo-series

*Figure 11.* The test accuracy curves for the baselines of AdamW, Lion and Shampoo without variance reduction, the approximate and the exact versions of MARS instantiations on CIFAR-10 dataset.

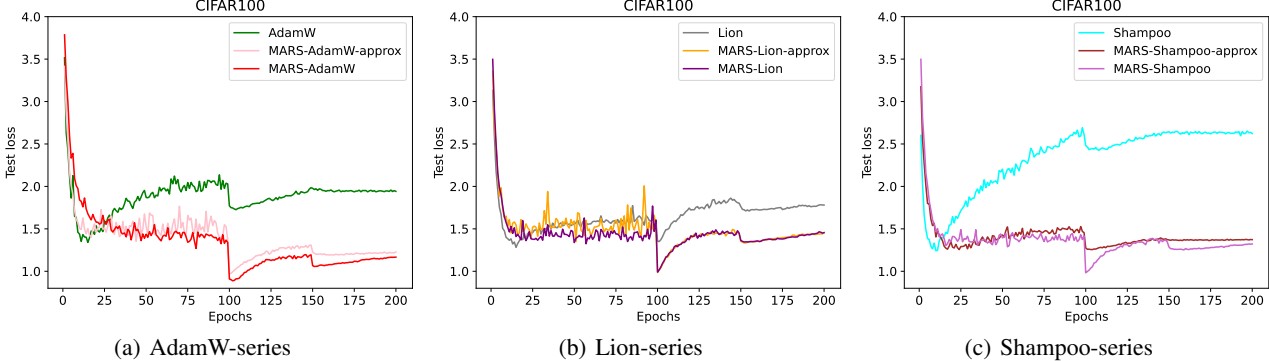

(a) AdamW-series      (b) Lion-series      (c) Shampoo-series

*Figure 12.* The test loss curves for the baselines of AdamW, Lion and Shampoo without variance reduction, the approximate and the exact versions of MARS instantiations on CIFAR-100 dataset.

different $\gamma$s where 0.025 is the best $\gamma$. Therefore, we used $\gamma = 0.025$ for other experiments in this paper.

### E.6 Different Learning Rate Scheduler

#### E.6.1 CONSTANT LR

To ensure a fair comparison by eliminating the influence of learning rate changes during training and to explore the potential for continuous training with MARS, we conduct supplementary experiments on GPT-2 small, medium, and large models using a constant learning rate for both AdamW and MARS-AdamW-approx. For each group of experiments, we compared between 2 different maximum learning rates. The training and validation curves are displayed in Figures 15 and 16. The results also indicate that our algorithm has superior performances over AdamW optimizer under a fair comparison of

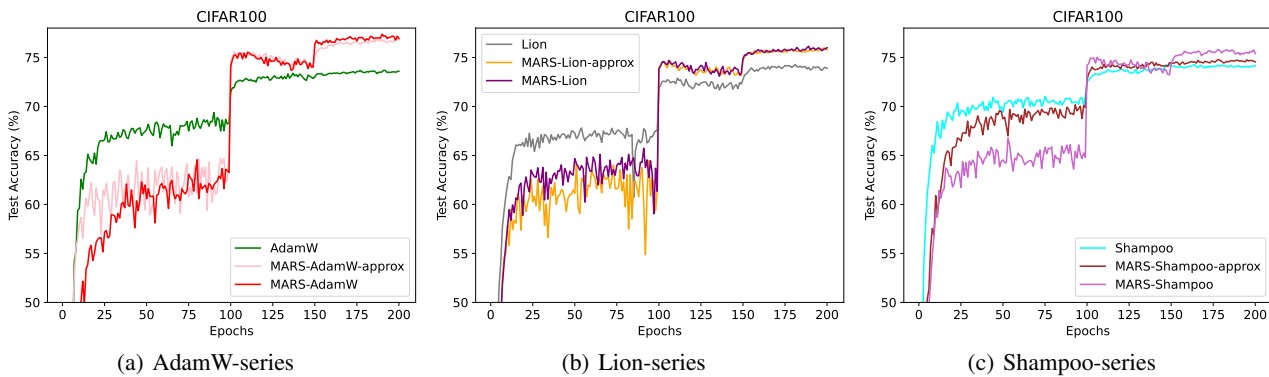

*Figure 13.* The test accuracy curves for the baselines of AdamW, Lion and Shampoo without variance reduction, the approximate and the exact versions of MARS instantiations on CIFAR-100 dataset.

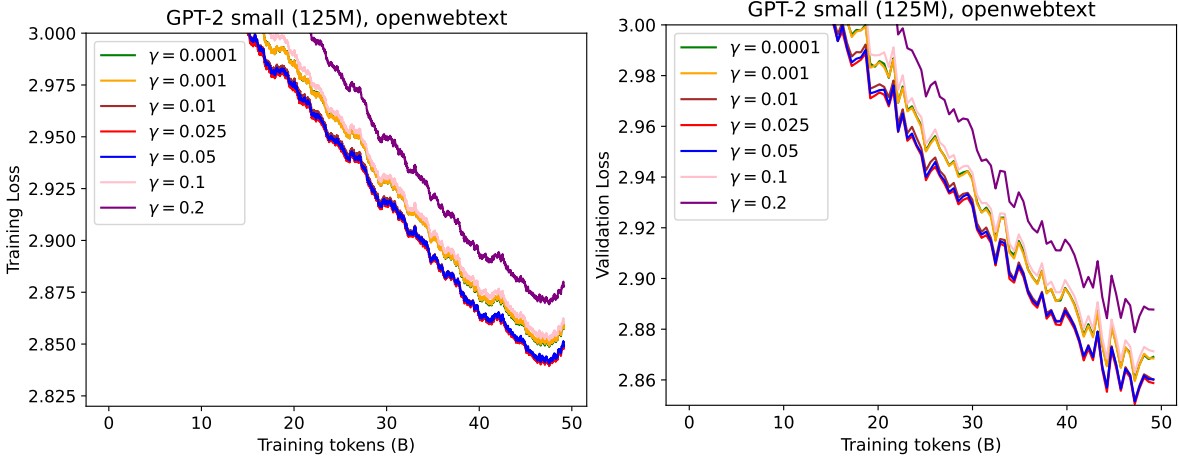

*Figure 14.* Training loss and validation loss curves for MARS and MARS-approx on GPT-2 small (125M) models trained with MARS-AdamW-approx with different $\gamma$s, pretrained with OpenWebText dataset and plotted against training tokens.

constant learning rates.

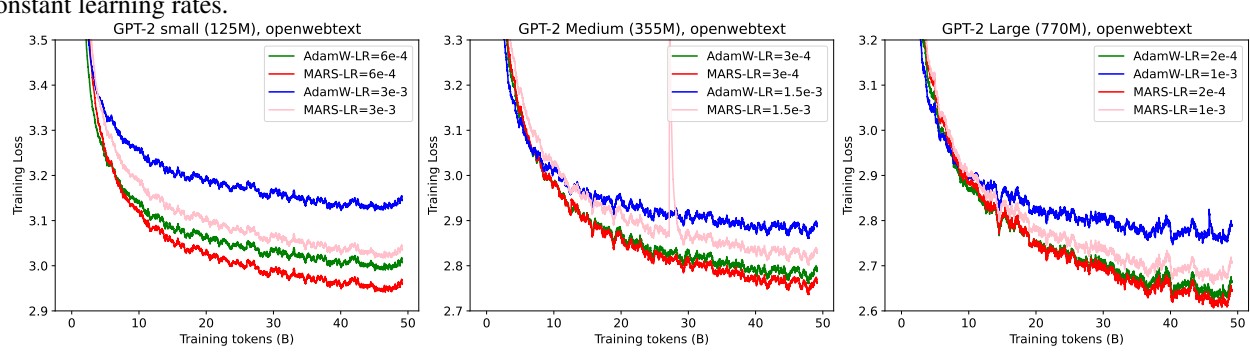

*Figure 15.* Training loss curves for MARS and MARS-approx on GPT-2 small (125M, left), medium (355M, middle) and large (770M, right) with constant learning rate, pretrained with OpenWebText dataset and plotted against training tokens.

### E.6.2 WSD Scheduler

Cosine learning rate scheduler is utilized in most of our experiments. Recently Hu et al. (2024) introduced a novel learning rate scheduler called Warmup-Stable-Decay (WSD) scheduler, which composed of 3 stages, including learning rate linear-warmup stage, constant learning rate stage, as well as learning rate decay stage. To test the flexibility and ability of continuous training of MARS trained with respect to learning rate schedulers, we implement experiments on GPT-2 small and medium models with AdamW and MARS-AdamW-approx scheduled with WSD for 10k, 20k, 50k and 100k steps. The

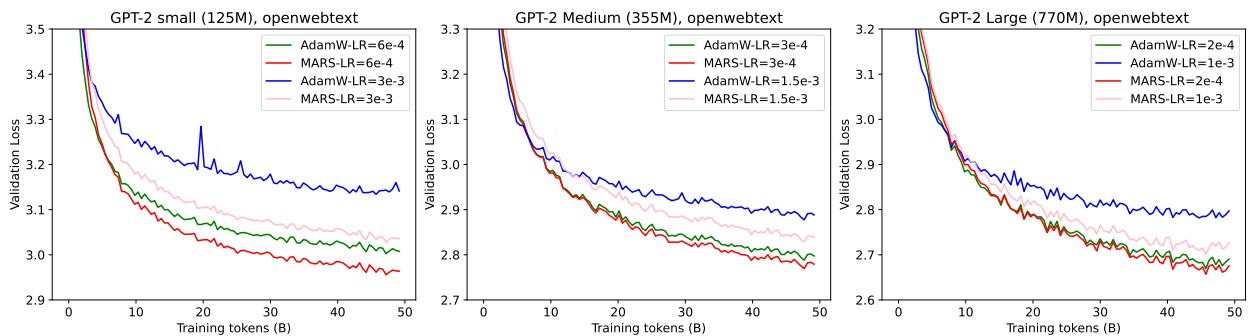

*Figure 16.* Validation loss curves for MARS and MARS-approx on GPT-2 small (125M, left), medium (355M, middle) and large (770M, right) with constatnt learning rate, pretrained with OpenWebText dataset and plotted against training tokens.

training and validation loss curves are shown in Figures 17 and 18. The curves indicate that MARS has a better potential for continuous training and exhibits an explicit edge over baseline algorithm.

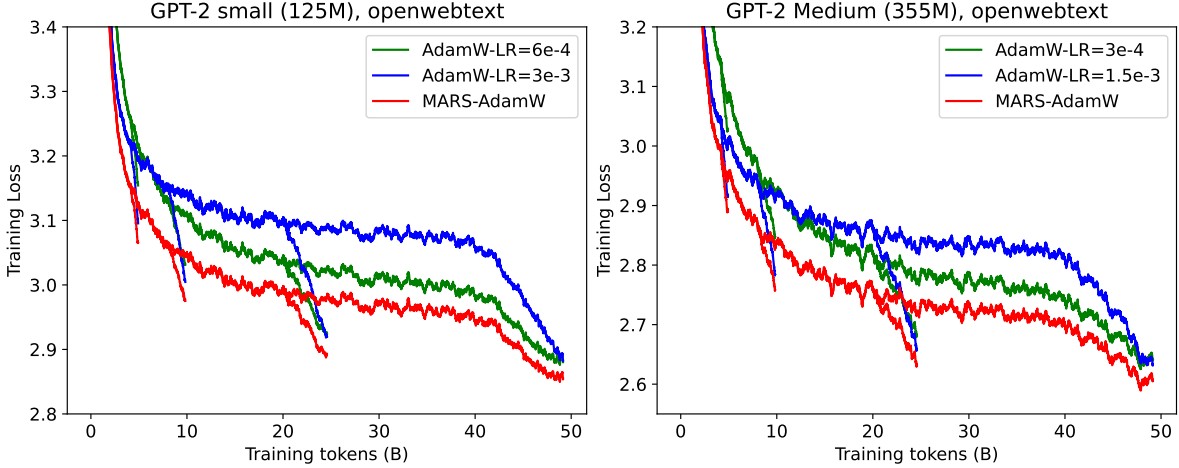

*Figure 17.* Training loss curves for AdamW (with different maximum learning rates, labeled with "LR") and MARS-AdamW-approx on GPT-2 small (125M, left) and medium (355M, right) with WSD Scheduler for 10k, 20k, 50k and 100k steps, pretrained with OpenWebText dataset and plotted against training tokens.

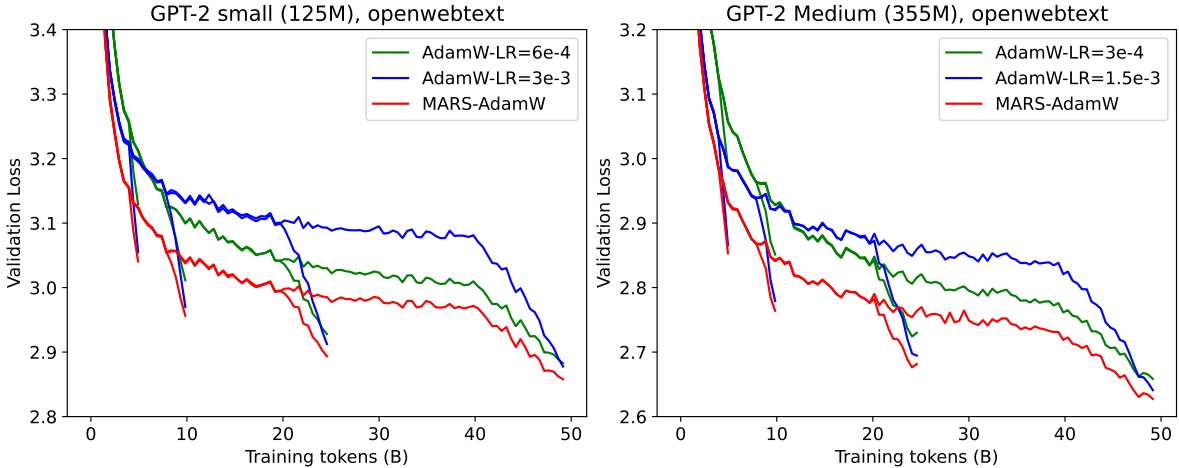

*Figure 18.* Validation loss curves for AdamW (with different maximum learning rates, labeled with "LR") and MARS-AdamW-approx on GPT-2 small (125M, left) and medium (355M, right) with WSD Scheduler for 10k, 20k, 50k and 100k steps, pretrained with OpenWebText dataset and plotted against training tokens.

### E.7 Sensitivity to Batch Size

We also investigate the sensitivity to batch size of our algorithm. We implement experiments with MARS-AdamW on GPT-2 small with batch size 240, 480 or 960, and compare them to AdamW. We fix other hyper-parameters the same as the main experiments and trained with 80,000 steps. The results are plotted in Figure 19. It can be seen that MARS-AdamW consistently outperforms AdamW, with the performance gap widening at smaller batch sizes—supporting the intuition that variance reduction is especially effective in high-variance (small-batch) settings.

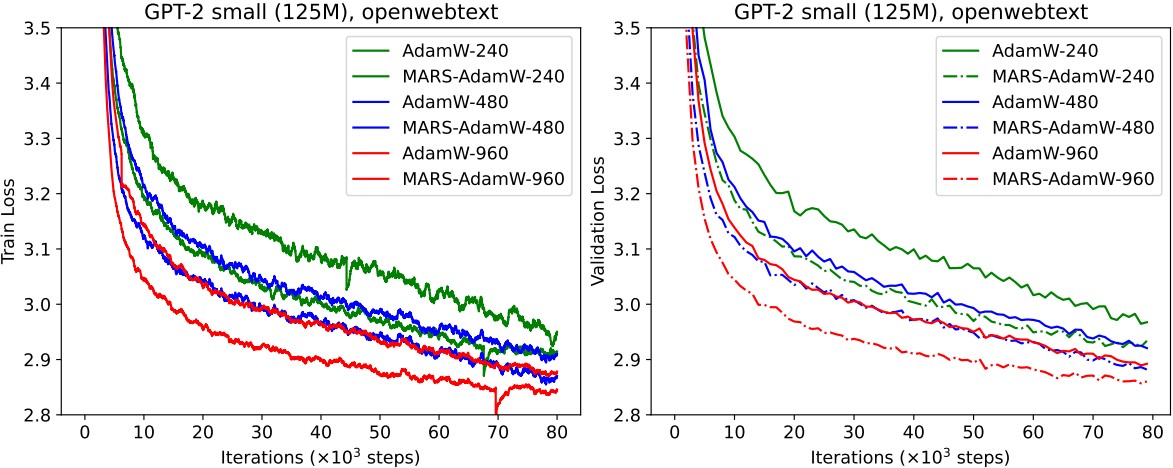

*Figure 19.* The training and validation loss curves for varying global batch sizes (240/480/960), plotted against iteration steps on GPT-2 small model (125M).

## F   Hyper-parameter Settings

For training parameters, we did a grid search over learning rates between $\{1e-4, 1.5e-4, 3e-4, 6e-4, 1e-3, 1.5e-3, 3e-3, 6e-3\}$, for weight decay coefficient, we did a grid search over $\{1e-1, 1e-2, 1e-3\}$. For AdamW baseline, although we utilized the golden standard learning rates in literature (a parameter search for AdamW have been done in Liu et al. (2023)), we also did a grid search on different parameters. Part of the results of different learning rates and learning rate schedules are also shown in Section E.6.1 and E.6.2, respectively. Table 10 summarizes the architectural hyperparameters for GPT-2 models with 125M (small), 355M (medium), 770M (large) and 1.5B (XL, only used in Appendix E.3) parameters. Table 11 lists the general hyperparameters used across all experiments, while Table 12 present the training hyperparameters for the small, medium, and large models, respectively.

*Table 10.* Architecture hyperparameters for GPT-2 series models (Radford et al., 2019).

| Model | #Param | #Layer | $n_{\text{head}}$ | $d_{\text{emb}}$ |
|---|---|---|---|---|
| GPT-2 small | 125M | 12 | 12 | 768 |
| GPT-2 medium | 355M | 24 | 16 | 1024 |
| GPT-2 large | 770M | 36 | 20 | 1280 |
| GPT-2 XL | 1.5B | 48 | 25 | 1600 |

*Table 11.* General hyper-parameters for the experiments.

| Hyper-parameter | Value |
|---|---|
| Steps | 100,000 |
| Batch size in total | 480 |
| Context length | 1024 |
| Gradient clipping threshold | 1.0 |
| Dropout | 0.0 |
| Learning rate schedule | Cosine |
| Warm-up steps | 2000 |
| Base seed | 5000 |

*Table 12.* Hyper-parameters for GPT-2 experiments. We use $\gamma_t \equiv 0.025$ for MARS and $\mu = 0.95$ for Muon.

| Hyper-parameter | GPT-2 Size | AdamW | Muon | MARS-AdamW |
|---|---|---|---|---|
| Max learning rate | small (125M) | 6e-4 | 2e-2 | 6e-3 |
| | medium (355M) | 3e-4 | 1e-2 | 3e-3 |
| | large (770M) | 2e-4 | 6.67e-3 | 2e-3 |
| Min learning rate | small (125M) | 3e-5 | 3e-5 | 3e-5 |
| | medium (355M) | 6e-5 | 6e-5 | 6e-5 |
| | large (770M) | 1e-5 | 1e-5 | 1e-5 |
| $(\beta_1, \beta_2)$ | small/medium/large | (0.9,0.95) | - | (0.95,0.99) |

