# OpenReview forum: "MARS: Unleashing the Power of Variance Reduction for Training Large Models"
_ICML.cc/2025/Conference — ICML 2025 poster_

### Official Review · Reviewer_ZHwA · 2025-03-12

**Overall Recommendation:** 4

**Summary:**

Training large deep neural networks is a challenging endeavor due to their massive scale and non-convexity.
This is true even more so for large language models where the current state-of-the-art method AdamW.
The paper propose incorporating a form of variance reduction, ala STORM, and combining it with a family of preconditioned adaptive gradient methods.
Notably, much work over the past decade has tried to incorporate variance reduction (which has exhibited great success in training convex ML models) with Deep Learning.

**Claims And Evidence:**

**Experimental**

- The experimental evaluation in the paper is very strong. The proposed MARS optimizer with AdamW or Lion both outperform tuned AdamW (the gold-standard for LLMs) on some standard large-language training tasks. The framework is also tested on some more classical tasks in the supplement on some benchmark computer visions tasks like CIFAR-10 and CIFAR-100, where the proposed optimizer also shows good performance.
- The supplement also contains comparisons across various hyperparameter values, so the paper is quite thorough.

**Theory**
- In Appendix B.2, the authors show under standard assumptions, that MARS AdamW converges to an $\epsilon$-approximate stationary point after $\tilde O(\epsilon^{-3})$ iterations, which matches the known lower bound (Arjevani et al. '23) in the literature up to log-factors.
Thus, in terms of theory, you can't really reasonably ask for a much better result without having to impose stricter assumptions.

**Overall**

- I think the results in the paper are compelling, and the proposed framework has promise to prove training of LLMs and large deep nets.
So I recommend acceptance.

**References**
Arjevani, Yossi, Yair Carmon, John C. Duchi, Dylan J. Foster, Nathan Srebro, and Blake Woodworth. "Lower bounds for non-convex stochastic optimization." Mathematical Programming 199, no. 1 (2023): 165-214.

**Essential References Not Discussed:**

The paper has done a thorough job reviewing variance reduction and adaptive gradient methods in the context of deep learning. All the major works are either mentioned in the main paper or in Appendix A of the supplement.

**Experimental Designs Or Analyses:**

The experiment design and the conclusions drawn seem fine to me. The paper uses standard benchmark models and does a thorough hyperparameter evaluation for their method and the competing methods.

**Methods And Evaluation Criteria:**

The datasets, models, and optimizers used in the paper for evaluation are all logical choices for the setting of the paper. AdamW is the most widely used optimizer in LLM training so it provides a natural baseline. Muon is a relatively recent optimizer, but has been shown to outperform AdamW on certain LLM tasks. So the baselines employed in the paper are sound.

**Other Comments Or Suggestions:**

**Typos**
-  First sentence in the abstract:

"Training deep neural networks—and more recently, large models demands efficient and scalable optimizers."

I'm assuming you meant to include language after large?

**Suggestions**

The essential related work is complete, but to give a more thorough picture of the optimization in ML literature, you should cite the following two recent papers:

- Derezinski, Michal. "Stochastic Variance-Reduced Newton: Accelerating Finite-Sum Minimization with Large Batches." In OPT 2023: Optimization for Machine Learning.
- Frangella, Zachary, Pratik Rathore, Shipu Zhao, and Madeleine Udell. "Promise: Preconditioned stochastic optimization methods by incorporating scalable curvature estimates." Journal of Machine Learning Research 25, no. 346 (2024): 1-57.

These papers combine variance reduction with preconditioning in the convex setting, and show it makes for a very strong combination, both in theory and in practice. So, I think these works are very much in alignment with the papers message.

**Other Strengths And Weaknesses:**

**Additional Strengths**

- The paper is well-written and very easy to follow.

**Weaknesses**

- Overall the paper, does not have any significant weaknesses in my view. It might be a little lacking in novelty in certain places (see my question in the box below), but in my view that takes a backseat to the fact that framework shows good practical performance and has good theoretical support.

**Questions For Authors:**

**On novelty**
- The main optimizer proposed here MARS-AdamW is very close to VRAdam, Algorithm 2 in Li et al. 2024, in that the variance reduction mechanisms are almost the same, except that MARS-AdamW has some scaling factors not present in VRAdam. Nevertheless, I was wondering if the authors could comment on the relation between the two algorithms?  Also, I'm very aware that although seemingly small, such things can have a big impact on practical performance. So I don't mind that much that its somewhat similar to existing approaches.

**Open source implementation**
- The algorithms seem quite effective. Do you plan on providing an open-source implementation if the paper is accepted? I think it could be useful for researchers working in this area.

**Relation To Broader Scientific Literature:**

The paper introduces an optimization framework based on variance-reduction that yields algorithms with the potential to make training of large models more efficient. This is of course a timely question given importance of LLMs and the fact they are so expensive to train.
I also find it interesting that the paper gives real practical evidence that variance reduction can be helpful in Deep Learning. Since Defazio and Bottou (2019), the conventional wisdom is that variance reduction just isn't useful in Deep Learning.
Based on the results of this paper, this no longer seems to be the case.
So, I think the paper's results are quite interesting relative to the existing literature.

**Theoretical Claims:**

The proofs I read in detail seem correct to me. I read  the full proofs for the main theorems, Theorems B.4 and B.5, and I skimmed the proofs of the supporting lemmas.

---

> ### Author Rebuttal · Authors · 2025-04-01
>
> Thank you for your careful evaluation of our work. We appreciate your recognition of our contributions and your constructive questions. We have carefully considered your concerns and we provide detailed responses to your questions below.
>
> ---
> **Q1**: First sentence in the abstract: Training deep neural networks—and more recently, large models demands efficient and scalable optimizers." I'm assuming you meant to include language after large?
>
> **A1**: Thank you for pointing this out. We intentionally omitted the word “language” to highlight the broader applicability of our approach beyond language models. As demonstrated in Appendix E.4, we include experiments on computer vision tasks to support the general applicability of the MARS optimizer. Nevertheless, we appreciate your feedback and will revise the sentence to avoid confusion.
>
> ---
>
> **Q2**: The essential related work is complete, but to give a more thorough picture of the optimization in ML literature, you should cite the following two recent papers.
>
> **A2**: We appreciate the suggestion and will incorporate these references into our related work section. [1] introduces SVRN, a method that combines variance reduction with second-order optimization. It achieves faster convergence rates for smooth, strongly convex problems while enabling large-batch parallelization, demonstrating advantages over first-order methods and subsampled Newton approaches. [2] proposes a suite of sketching-based preconditioned algorithms (including SVRG, SAGA, and Katyusha variants) for ill-conditioned convex problems. Their work introduces "quadratic regularity" to guarantee linear convergence even with infrequent preconditioner updates, outperforming tuned baselines on regression tasks.
>
> [1] Dereziński, M. (2022). Stochastic variance-reduced newton: Accelerating finite-sum minimization with large batches.
> [2] Frangella, Z., Rathore, P., Zhao, S., & Udell, M. (2024). Promise: Preconditioned stochastic optimization methods by incorporating scalable curvature estimates.
>
> ---
>
> **Q3**: On novelty: The main optimizer proposed here MARS-AdamW is very close to VRAdam, Algorithm 2 in Li et al. 2024, in that the variance reduction mechanisms are almost the same, except that MARS-AdamW has some scaling factors not present in VRAdam. Nevertheless, I was wondering if the authors could comment on the relation between the two algorithms? Also, I'm very aware that although seemingly small, such things can have a big impact on practical performance. So I don't mind that much that it's somewhat similar to existing approaches.
>
> **A3**: Thank you for bringing our attention to VRAdam. While MARS-AdamW and VRAdam share the high-level goal of integrating variance reduction into adaptive methods, there are several key differences.
>
> First, unlike VRAdam (and SuperAdam by Huang et al., 2021), MARS defines the second-order momentum $v_t$ as the exponential moving average of the squared norm of the corrected gradient $c_t$, rather than that of the raw stochastic gradient. This modification is critical for maintaining the correct update scale in each coordinate direction.
>
> Additionally, MARS introduces two further components: a tunable scaling factor $\gamma_t$ and gradient clipping on $c_t$, both of which are essential for stability and performance, especially in deep learning scenarios.
>
> Moreover, we introduces weight decay in both our implementation, as well as the theoretical analysis, while VRAdam does not.
>
> Lastly, MARS is designed as a flexible framework that extends beyond AdamW. Its design allows easy adaptation to other optimizers such as Lion and Shampoo, highlighting its broader applicability.
>
> We hope this clarifies the distinctions and contributions of our approach. We will add the above discussion in our revised manuscript.
>
> ---
>
> **Q4**: Open Source Implementation: The algorithms seem quite effective. Do you plan on providing an open-source implementation if the paper is accepted? I think it could be useful for researchers working in this area.
>
> **A4**: Thank you for your kind words and support for open research. We have included our implementation in the supplementary materials and plan to open-source the code upon acceptance. We also welcome feedback from the community to further improve it.

---

> > ### Comment · Reviewer_ZHwA · 2025-04-05
> >
> > I thank the authors for addressing all my comments/questions and for agreeing to add the suggested references. This is a good paper, and I maintain my view that it should be accepted.

---

> > > ### Author Response · Authors · 2025-04-08
> > >
> > > Thank you for your thoughtful review and for recognizing the strengths of our work! We're grateful for your support for acceptance and your valuable suggestions, which have helped improve the paper. If you feel the revisions have meaningfully enhanced the manuscript, we would greatly appreciate your support in raising your evaluation.

---

### Official Review · Reviewer_XvSa · 2025-03-14

**Overall Recommendation:** 5

**Summary:**

The paper presents an algorithm template for variance reduction that includes common optimizers such as AdamW and Lion as a special case. A convergence analysis is provided the smooth case, with an assumption on the pre-conditioning matrix. Extensive experiments are conducted on language modeling and vision tasks.

**Claims And Evidence:**

The theoretical claims have accompanying proofs and the experimental results are accompanied by both code and data.

**Essential References Not Discussed:**

None that I am aware of beyond the discussion above.

**Experimental Designs Or Analyses:**

The designs are overall valid, and I appreciate that wall clock time was used in some of the experiments. What is slightly concerning is that there seems to be no experiments/discussions surrounding FLOPs. I believe it is very important in this field to be aware of the exact computational cost at this scale. The authors seem to have access to industrial-level computing resources, making wall time a sometimes unhelpful performance measure for researchers, as the relative impact of additional FLOPs per iteration might change between differing resources (older GPUs, etc). This is my main concern overall, so it would be helpful for the authors to provide these numbers in the rebuttal.

**Methods And Evaluation Criteria:**

Overall, they do, as the most important evaluations are the optimization performance on respectably-sized deep learning models.

**Other Comments Or Suggestions:**

N/A

**Other Strengths And Weaknesses:**

The paper is generally well-written, and Section 3.2 is especially nice. My main concerns are some of the theoretical statements, experiments with FLOP count, and some discussions that would benefit the main text. I would raise my score to a 5 if all were addressed.

**Questions For Authors:**

None other than those brought up earlier.

**Relation To Broader Scientific Literature:**

Personally, I would be excited to finally see a variance reduction method actually become performant in deep learning, and much of the relevant literature in convex optimization is cited. That being said, I wish the authors were a bit more generous toward the stochastic simulation literature. The discussion on page 4 starting from line 206 seems to suggest that tunable variance reduction is a recent idea developed in deep learning optimization. Rather, nearly all of the (at least unbiased) variance-reduced gradient estimators are control variates, and setting $\gamma = 1$ has been a purposeful choice since SVRG. See [Asmussen & Glynn (2007)](https://link.springer.com/book/10.1007/978-0-387-69033-9), Chapter V Section 2 on control variates or [Mehta et. al. (2024)](https://openreview.net/pdf?id=TTrzgEZt9s) Eq. (6) for a simpler derivation than the one on page 4 of the submitted paper.

On a similar note, I feel as though the authors do not seriously discuss the schedule for $\gamma_t$ until Appx. F. Because this is one of the main components of the algorithm, the reader should be exposed to it earlier.

**Theoretical Claims:**

I have read the claims and the proofs briefly. While theory is generally not valued as much in deep learning optimization, I do believe there is some sloppiness in the presentation of the results that could be improved.

1. It appears from the theorem statements that they analyze an algorithm for which the parameters $\beta_1$ and $\beta_2$ may vary with time. I do not see this stated anywhere in the algorithm boxes and accompanying discussions, and the reader should not have to discover this. Similarly, in Lemma C.4, the problem constant $D$ is not formally introduced until Lemma D.2. Finally, the learning rate sequence is set, but is also accompanied by an upper bound. In general, this would be invalid as it placed an assumption on the problem constants, but because there is a free parameter $s$, the condition that $\eta_t \leq ...$ and $s \geq 1$ in Theorem B.4 and B.5 should be converted into a condition of the form $s \geq ...$.
2. I believe there should be some discussion about Assumption B.3. Is this common in the literature? Considering the extreme case in which one could compute or even approximate the Hessian as the pre-conditioner, this assumption is essentially assuming that the objective is strongly convex, which disagrees completely with the deep learning objective.
3. On a more superficial note, I believe the authors meant to write $F(\mathbf{x}_{T+1})$ in line 916. While the proofs are well-structured and easy to follow logically, the computations are quite heavy. If the authors are able, a bit of readability (using colors, etc) would be helpful. This does not factor into my score, but I still request this revision. See the analysis of SAGA in 5.4.4 of [Bach (2024)](https://www.di.ens.fr/~fbach/ltfp_book.pdf) for a reader-friendly Lyapunov stability analysis.

This said, it is laudable that the authors provided this analysis, which is likely a further step than similar papers in the area.

---

> ### Author Rebuttal · Authors · 2025-04-01
>
> We sincerely appreciate your careful review and constructive feedback. Your comments on both the theoretical and experimental aspects of our work have helped us further strengthen our contributions. We have carefully considered your suggestions and provide detailed responses below.
>
> ---
> **Q1**: Parameters $\beta_1$ and $\beta_2$ may vary with time. I do not see this stated anywhere in the algorithm boxes and accompanying discussions
>
> **A1**: Thank you for your question. Our use of time-varying $\beta$ follows the line of theoretical work on Adam, where $\beta$ is dependent on $\eta_t$. We will add a note that, for theoretical analysis, $\beta_{1, t}$ and $\beta_{2, t}$ are required. In practice, these parameters are typically chosen as constants.
>
> ---
> **Q2**: (1) In Lemma C.4, the problem constant D is not formally introduced until Lemma D.2. (2) The learning rate sequence is set, but is also accompanied by an upper bound
>
> **A2**: Thank you for your observation. (1) We will revise the definition of constant D into Lemma C.4 and Theorem B.5. (2) We will revise the upper bounds on $\eta_t$ in Theorems B.4 and B.5 by expressing them as conditions on the parameter $s$, avoiding any implicit assumptions on problem-specific constants. Specifically, we will require $s\ge 8L^3/\rho^3$ in Theorem B.4 and $s\ge \max(8L^3/\rho^3, 64\lambda^3)$ in Theorem B.5.
>
> ---
> **Q3**: I believe there should be some discussion about Assumption B.3. Is this common in the literature?
>
> **A3**: Thank you for your question. Assumption B.3 is not a strong convexity assumption, but rather a condition for ensuring computational stability. This assumption is common in the literature--for example, Assumption 3 in Huang et al., (2021) and AS.3 in [1]. It is more often used implicitly, as in [2], where $H_t = \eta / (\sqrt{\hat{v}_t} + \lambda)$ in Algorithm 1, and Theorem 4.1 dependent on $\lambda$. The same holds for Algorithm 2 in [3] and many others. We will include a more detailed discussion of this assumption and its role in related work in our revised manuscript.
>
> [1] Wang et al., Stochastic Quasi-Newton Methods for Nonconvex Stochastic Optimization
> [2] Li et al., Convergence of Adam Under Relaxed Assumptions
> [3] Zhuang et al., Adabelief Optimizer: Adapting Stepsizes by the Belief in Observed Gradients
>
> ---
> **Q4**: While the proofs are well-structured and easy to follow logically, the computations are quite heavy. If the authors are able, a bit of readability (using colors, etc) would be helpful
>
> **A4**: Thank you for the suggestion. While the rebuttal format limits manuscript changes, we will revise the proofs in the final version to follow the style of the analysis in Section 5.4.4 of Bach (2024).
>
> ---
> **Q5**: It would be helpful for the authors to provide FLOPs in the rebuttal
>
> **A5**: Thank you for raising this point. We now provide a detailed FLOPs analysis for our Transformer experiments (with GPT-2 small as an example).
>
> For a Transformer with batch size $B$, sequence length $s$, $l$ layers, hidden size $h$, and vocabulary size $V$, the total number of trainable parameters is $N = h(12lh + 13l + V)$ (approximately $1.24 \times 10^8 $). The forward pass requires $N_F=2Bsh(12lh + 2sl + V) $ FLOPs (around $1.40 \times 10^{14}$), and the backward pass roughly doubles that.
>
> For optimizer updates:
> - AdamW: 11N FLOPs (≈ $1.36 \times 10^9$)
> - MARS-AdamW (approx): 17N FLOPs (≈ $2.10 \times 10^9$), accounting for $c_t$ clipping
> - Muon: 11N + 12h^2l(30h + 20) FLOPs (≈ $1.96 \times 10^{12}$)
>
> We have added Figure 2 (https://anonymous.4open.science/r/M-E6D7/F.pdf) to illustrate loss vs. FLOPs across different optimizers.
>
> ---
> **Q6**: Related literature on control variates
>
> **A6**: We appreciate the reviewer’s insight and agree with the relevance to stochastic simulation and control variates literature. We will discuss the connection with the suggested papers in the final version. We would like to point out that, in methods like STORM, which are not strictly unbiased, the update does not follow the classical control variate form $\mathbb{E}||X-\gamma(Y-E[Y])||^2$. Instead, they control estimation error recursively through differences across consecutive iterates. When casting the STORM-style estimator in the simplified form, we define $Y=\nabla f(x_{t+1},\xi_{t+1})-\nabla f(x_t,\xi_{t+1})$, and the estimation error we aim to reduce becomes $\mathbb{E}||X+\gamma Y-E[Y]||^2 $(as shown on Page 4 of our manuscript, full formulation at Line 1172). Unlike standard control variates, the optimal $\gamma^*$ here cannot be directly represented in terms of $Y-E[Y]$, making the connection less explicit.
>
> ---
> **Q7**: The reader should be exposed to the schedule for $γ_t$ earlier
>
> **A7**:  Thank you for your suggestion. In the final version, we will include a more detailed discussion of the schedule for $\gamma_t$ in the methodology section.

---

> > ### Comment · Reviewer_XvSa · 2025-04-01
> >
> > Thank you for your work on the rebuttal. My concerns are addressed and I have raised my score.

---

> > > ### Author Response · Authors · 2025-04-03
> > >
> > > Thank you for taking the time to review our rebuttal and we truly appreciate your thoughtful feedback and support of our work!

---

### Official Review · Reviewer_L7rR · 2025-03-14

**Overall Recommendation:** 4

**Summary:**

This paper studies variance-reduction algorithm for large language models, and proposes a general framework, MARS, that combines variance-reduction and pre-conditioning. In particular, the authors add a control parameter $\gamma_t$ to the STORM momentum aggregate, i.e., $m_t = \beta m_{t-1} + (1-\beta) c_t$ where $c_t = \nabla f(x_t,\xi_t) + \gamma_t \frac{\beta}{1-\beta}(\nabla f(x_t,\xi_t) - \nabla f(x_{t-1},\xi_{t-1}))$. $\gamma_t$ provides interpolation between noise control and bias: setting $\gamma_t=1$ recovers the variance-reduction update of STORM, and setting $\gamma_t=0$ recovers standard SGDM. In addition, the authors also incorporate the pre-conditioning technique to the variance-reduced momentum, namely $x_{t+1} \approx x_t - \eta_t H_t^{-1}m_t$, where $H_t$ is some approximation (either full-matrix or diagonal) of the hessian $\nabla^2 F(x_t)$. Upon substituting popular pre-conditioning methods in the literature, the authors instantiate different algorithms of MARS. The authors provide theoretical analysis of the convergence guarantee of the proposed framework, and presented empirical evidence that MARS have better performance than the state-of-art optimizers on training language models.

**Claims And Evidence:**

yes

**Essential References Not Discussed:**

NA

**Experimental Designs Or Analyses:**

yes

**Methods And Evaluation Criteria:**

yes

**Other Comments Or Suggestions:**

- I think the name MARS-Shampoo is a little misleading. First, eq (3.24) is only true if we turn off the pre-conditioning aggregate of $L_t,R_t$ in Shampoo and use the instantaneous gradients. Also, the update rule of $m_t \mapsto UV^\top$ (either directly with SVD or approximately with Newton-Schulz) is more aligned with muon than shampoo.

- I believe the theorems prove convergence of MARS without clipping $c_t$ (which is not exactly the same algorithm in the pseudo-code)?

Typos:
- line 98: to reaches -> to reach
- 238 (right): line 2 -> line 5
- above Sec. 3.2.3: "we present a lemma..." is the lemma missing?
- 836: $\nabla f$ -> $\nabla F$
- eq (C.2): missing a )

**Other Strengths And Weaknesses:**

This paper has concrete results and shows significant novelty. Although variance-reduction algorithms have better theoretical convergence rate due to the slightly strong assumption (mean-square smoothness), they generally perform poorly on empirical tasks and lose to state-of-art optimizers such as Adam and its variants. Notably, MARS outperforms those algorithms on training language models, not only in sample complexity (number of tokens) but also in oracle complexity (as measured by wall-clock time). The novelty is thus showing that variance-reduction algorithms can be effective in practice. In addition, to my knowledge this is the first algorithm that incorporate preconditioning technique with variance-reduction algorithms. On the theory side, this paper also provides a theoretical justification of why properly choosing the control parameter $\gamma$ for the noise correction can improve the performance, and it also provides convergence guarantee of MARS. For these reasons, I'd recommend acceptance.

**Questions For Authors:**

1. Experiment results in Figure 1 suggests that muon has worse performance compared to AdamW and Lion; while in muon's report [1], muon outperformance these two optimizers. Could the authors briefly discuss this experiment difference? Is it due to different tasks with different scales? Also, have the authors tried the performance of MARS-Shampoo with different $\gamma_t$'s other than $1-\mu$ (which approximately recovers muon)?

2. I find the equivalence between nesterov update and momentum with noise-corrected gradients very interesting (Lemma D.1). Do authors know prior works stating the same result? Or is this a new finding?

3. How do Theorem B.4 and B.5 imply a convergence rate of $\tilde O(T^{-1/3})$ for $\mathbb{E}\\|\nabla F(x_t)\\|^2$? I was trying to directly apply the smooth descent lemma $F(x_{t+1})-F(x_t) \le \langle \nabla F(x_t), x_{t-1}-x_t\rangle + \frac{L}{2}\\|x_{t+1}-x_t\\|^2$, but then $x_{t+1}-x_t = -\eta_t H_t^{-1}m_t$ and $(F(x_{t+1})-F(x_t))/\eta_t \le - \langle \nabla F(x_t), \nabla F(x_t) - (F(x_t)-H_t^{-1}m_t)\rangle + \frac{L}{2\eta_t}\\|x_{t+1}-x_t\\|^2$, which requires a bound on $\mathbb{E}\\|\nabla F(x_t) - H_t^{-1}m_t\\|^2$ instead of $\mathbb{E}\\|\nabla F(x_t) - m_t\\|^2$.

4. I find the empirical choices of $\gamma_t$, shown in Figure 14, very interesting. (As a side note, I find it a bit hard to read the colors in the plot as many curves overlap with each other.) The optimal value of $\gamma=0.025$ is very close to the value of $1-\mu$, in which case the momentum with correct gradient is roughly momentum with nesterov update and MARS-Adamw should be close to AdamW with nesterov update. Have the authors tested the empirical performance of nesterov AdamW (or Adan) and compared it with MARS? Also, could the authors further elaborate on $\gamma\approx 1-\mu$? It makes me wonder whether variance-reduction is the right way to interpret MARS, since the noise-correction only plays a small role. Could it be possible that nesterov update (or optimistic update) is a better interpretation?

5. Have the authors tried MARS without preconditioning? I think the comparison of MARS-AdamW and MARS-SGDM (which is presumably significantly worse) would help motivating that both variance-reduction and preconditioning are necessary to achieve a good performance.

[1] Keller Jordan, "Muon: An optimizer for hidden layers in neural networks", https://kellerjordan.github.io/posts/muon/, 2024.

**Relation To Broader Scientific Literature:**

This paper connects to the subfield of variance-reduction and pre-conditioning. Notably, it proposes an empirically well-performing variance-reduction algorithm, while prior variance-reduction algorithms don't perform well in practice.

**Theoretical Claims:**

yes

---

> ### Author Rebuttal · Authors · 2025-04-01
>
> Thank you for your detailed and thoughtful review. Below, we provide detailed responses to each of your comments.
>
> ---
> **Q1**:The name MARS-Shampoo
>
> **A1**: The interpretation of Shampoo as a projection onto the closest semi-orthogonal matrix ($UV^\top$) originates from Proposition 5 of [1], which presents Shampoo as steepest descent under the spectral norm. While this interpretation is not directly proposed in Gupta et al., 2018, our MARS-Shampoo algorithm builds on (19) from [1]. Moreover, MARS-Shampoo employs Newton iteration, in contrast to the Newton-Schulz method used in Muon. We will clarify that this is an approximate version of Shampoo in our revision.
>
> [1] Bernstein et al., Old optimizer, new norm: An anthology
>
> ---
> **Q2**:The theorems prove convergence of MARS without clipping
>
> **A2**: You are correct. This discrepancy arises because our theoretical results rely on standard smoothness assumptions, which simplify the analysis but do not directly account for the clipped version.
>
> In practice, gradient clipping is employed as a heuristic to enhance stability, particularly in deep learning where gradients may explode. While our work does not theoretically justify the clipped variant, prior studies (e.g., [1]) have shown that clipping can handle scenarios where gradient smoothness conditions are violated or grow with gradient norms. We leave the extension of our analysis to incorporate clipping—and its interplay with momentum and adaptive gradient methods—as future work.
>
> [1] Zhang et al., Why gradient clipping accelerates training: A theoretical justification for adaptivity.
>
> ---
> **Q3**:Typos
>
> **A3**: Thank you, we have corrected them.
>
> ---
> **Q4**:Experimental difference between Muon's report and Figure 1 of our paper
>
> **A4**: The performance difference primarily stems from differences in experimental setups: while Muon’s speedrun combines architectural changes with its optimizer, our evaluation isolates the optimizer’s impact using Muon's open-sourced implementation on GPT-2 architecture. We rigorously tuned learning rates, but differences persist due to factors like dataset choice (FineWeb vs. OpenWebText), training scale (<10B vs.50B tokens), and learning rate schedulers (WSD vs. Cosine). Regarding the MARS-Shampoo algorithm, we added experiments on CV tasks with varying γ (see Table 1 in https://anonymous.4open.science/r/M-E6D7/F.pdf). For CIFAR-100, a larger value γ>1-μ (0.1) yields the best performance, while for CIFAR-10, a smaller value γ<1-μ (0.001) performs best.
>
> ---
> **Q5**:Is Lemma D.1 a new finding.
>
> **A5**: Thank you for your interest in Lemma D.1. While we independently derived this connection, we later found that similar ideas have appeared implicitly in prior work. A special case of the forms in (D.1) and (D.2) align with the implementation of Nesterov in PyTorch, and a special case of the 2nd form is stated as AGD-II in Lemma 1 of Adan [Xie et al. 2022]. However, to the best of our knowledge, Lemma D.1 is the first to present a unified formulation and theoretical framework that encompasses both classical momentum and noise-corrected gradients—capturing HB, NAG, Lion, and STORM within a single view.
>
> ---
> **Q6**: How do Theorem B.4 and B.5 imply a convergence rate
>
> **A6**: As a sketch of proof, after Jensen's inequality, we sum up $||x_{t+1} - x_t||/\eta_t = ||H_t^{-1} m_t|| $ with $||\nabla F(x_t)-m_t||/\rho \ge \frac{1}{||H_t||} ||\nabla F(x_t)-m_t||$. Triangle inequality (on $m_t$ and $\nabla F(x_t) - m_t$) and summing over $t$ leads to a bound of the form $\frac{1}{||H_t||} ||\nabla F(x_t)||\le O(T^{−1/3})$. While the final convergence requires upper boundedness of $||H_t||$, this is commonly satisfied in practice. We will include the detailed derivation in the final revision of our manuscript.
>
> ---
> **Q7**: Empirical performance of Nesterov AdamW (or Adan). Also, could the authors further elaborate on γ≈1-μ?
>
> **A7**: Thank you for the feedback. We compared MARS with Adan and found that while Adan achieves similar final performance under a cosine schedule, it consistently underperforms MARS at the same learning rate (see Figures 5–6 in the link above).
>
> For the choice of γ, we note that the performance of DL tasks can be influenced not only by theoretical factors but also by practical considerations such as smoothness and stability. These factors can hinder performance as γ increases. To validate the role of variance reduction, we examine it from two perspectives:
>
> (a) Comparing exact vs. approximate variance reduction (Figure 5). and
>
> (b) We explore γ values not strictly tied to 1-μ (e.g., γ=0.025 vs. μ=0.95).
>
> We added experiments for MARS-AdamW on CV tasks with varying γ (see Table 2 in the link above). For CIFAR-10, a larger value γ>1-μ (0.1) yields the best performance, while for CIFAR-100, a smaller value γ<1-μ (0.01) performs best.
>
> ---
> **Q8**: MARS-SGDM
>
> **A8**: We have now included a comparison between MARS-AdamW and MARS-SGDM in Figure 4 in the link above.

---

> > ### Comment · Reviewer_L7rR · 2025-04-01
> >
> > I thank the authors for the detailed response. My questions are all addressed, and I don't have any further questions. Overall, I think this paper has strong results, and I keep my current score.

---

> > > ### Author Response · Authors · 2025-04-03
> > >
> > > Thank you for your positive feedback! We're glad our rebuttal fully addressed your questions. If you believe this paper deserves a higher score, we would greatly appreciate your consideration in adjusting it.

---

### Official Review · Reviewer_8Y9B · 2025-03-14

**Overall Recommendation:** 4

**Summary:**

This paper proposes MARS, which is a framework that applies variance reduction to preconditioning methods for stochastic optimization. Namely, MARS takes a scaled update from STORM (which calculates momentum as an EMA combining the stochastic gradient and a gradient correction term) and applies preconditioning on the resulting variance-reduced object. The preconditioning can be achieved with different methods performing different approximations of the Hessian, leading to different instances of MARS-optimizers. The authors show that MARS has a superior convergence rate to AdamW. They empirically show that MARS-AdamW(-Approx) and MARS-Lion(-Approx) show competitive performance to Muon, Lion, and AdamW on GPT-2 trained on OpenWebText/FineWeb-Edu and ResNet-18 on CIFAR-10/100.

**Claims And Evidence:**

It is clear that MARS can be applied on any choice of adaptive gradient method, and the authors show that adding MARS on AdamW and Lion results in improved performance on language model pretraining at various scales (125m, 355m, 770m), language model finetuning (eg. Table 1) and computer vision tasks (Appendix E.4).

This may be pedantic, but the authors say numerous times that MARS is a framework which ‘unifies’ first-order and second-order optimization methods (eg. Lines 114-116, column 2). I’m not sure if ‘unified’ is the right word here– it seems that MARS is simply applying a scaled version of the STORM momentum, which you can then perform any preconditioning method on top. I wouldn’t say that it ‘unifies’ the first and second order methods, but is rather an addition that can be applied on top of the adaptive gradient method of your choice, because variance reduction is orthogonal to preconditioning.

**Essential References Not Discussed:**

None; see Questions for AdEMAMix citation.

**Experimental Designs Or Analyses:**

Experimental design and analyses are sound. I believe some additional analyses would strengthen the paper, given that the proposed framework combines existing variance reduction methods with existing adaptive gradient methods. More specifically, it was previously suggested that variance reduction can be ineffective in deep learning but the authors find that it can be beneficial here; are there characteristics of settings in which variance reduction should help, or are there characteristics of the adaptive gradient method that may yield greater empirical benefit with MARS as opposed to other adaptive gradient methods? Furthermore, the authors (please correct me if I'm wrong) use a fixed batch size across model scales, but an analysis between small-batch vs. large-batch training would be insightful, given that variance reduction intuitively should help more in smaller-batch or high-variance scenarios. Finally, the authors report multiple baselines (AdamW, Lion, Muon) but do not have other baselines which yield similar variance reduction effects (see Questions).

**Methods And Evaluation Criteria:**

The authors cover a variety of datasets tracking reasonable metrics for pretraining and computer vision tasks. For pretraining, they also show that downstream performance is also improved compared to other optimizers when controlling for number of pretraining steps. Implementing MARS in practice typically needs two forward passes on two different batches, but the authors conduct sweeps over hyperparameters (particularly learning rate, $\gamma$) and their -exact and -approx version (which replaces $\nabla f(x_{t-1}, \xi_t)$ with $\nabla f(x_{t-1}, \xi_{t-1})$ in the MARS update) to show that performance is comparable.

**Other Comments Or Suggestions:**

* Line 092, first column: “variance-reduction” -> “variance reduction”
* Line 163, first column: “precondtioned” -> “preconditioned”
* Line 1476: "grid research" -> "grid search"
* Figure 13: y-axis should be "Test Accuracy"?

**Other Strengths And Weaknesses:**

* The paper is generally well-written and presents prior background on variance reduction techniques well.
* The paper demonstrates that variance reduction techniques can be added as a separate dimension to preconditioning for stochastic optimization and yield further benefits in practical settings.

* I believe a lot of space is taken in the main paper simply reiterating the updates of known optimizers (AdamW, Lion, Shampoo) and replacing the momentum term with the scaled gradient correction term; there are very few results in the main paper, and many in the appendix. I would suggest to the authors that the paper would benefit bringing some of these results (such as ablating -approx vs -exact, one vision experiment, etc.) to the main paper and deferring details about the instantiation of MARS to the appendix.
* For novelty, the paper combines two known techniques (STORM modulo a scalar correction, and adaptive optimization methods), although I see the value in the empirical experiments showing that variance reduction techniques have potential in further improving language model performance which is compatible with preconditioning. Thus, I lean towards acceptance of the paper.

**Questions For Authors:**

1. What was the reasoning behind the choice in not including MARS-Shampoo for the language model pretraining (even just GPT-2 small) experiments?
2. Are there any insights regarding the learning rate stability of MARS-AdamW? If the authors could provide some insight about tuning the MARS-AdamW learning rate (eg. it seems that it is typically higher than the optimal LR for the adaptive method chosen) this would be a valuable addition to the paper.
3. Did the authors sweep across batch size for the pretraining experiments? Does MARS show less pronounced improvements in the large -batch setting (even on GPT-2 small?)
4. How do the variance reduction techniques employed by MARS, STORM, etc. compare to methods like AdEMAMix[1] which simply maintains multiple EMAs with longer retainment of the past?

[1] Pagliardini, Matteo, Pierre Ablin, and David Grangier. "The ademamix optimizer: Better, faster, older." arXiv preprint arXiv:2409.03137 (2024).

**Relation To Broader Scientific Literature:**

There is an established line of variance-reduced methods as mentioned by the authors (eg SVRG, SARAH, SPIDER, STORM) as well as adaptive gradient methods. The authors show that variance-reduction can indeed help even at scale—they cite prior negative results on variance reduction for deep neural nets (e.g., Defazio & Bottou’s arguments) and give evidence that in certain modern settings, variance reduction can indeed be beneficial.

**Theoretical Claims:**

I have not rigorously verified the convergence guarantees. It seems that it is a relatively straightforward extension of the STORM approach.

---

> ### Author Rebuttal · Authors · 2025-04-01
>
> Thank you for your thorough review and constructive feedback. We sincerely appreciate your recognition of the novelty of our proposed framework and the clarity of our writing. Additionally, your detailed questions have been invaluable in helping us refine our work, and we listed our responses to your questions below.
>
> ---
> **Q1**: I'm not sure if 'unified' is the right word here, because variance reduction is orthogonal to preconditioning
>
> **A1**: Thank you for your comment. We agree that variance reduction is orthogonal to preconditioning, and we appreciate the opportunity to clarify our use of the term. Our use of “unified” refers to two key aspects of the MARS framework: (1) The scaled variance-reduced momentum we propose not only generalizes STORM-style updates, but also recovers and accommodates various forms of momentum used in practice, including Nesterov, heavy-ball, Adan, and Lion. This highlights a shared underlying structure that our analysis brings to light. (2) The framework is designed to be applied on top of a wide range of preconditioning strategies, including those used in first-order methods like AdamW as well as second-order approaches like Shampoo. While variance reduction and preconditioning are orthogonal, MARS's integration enables theoretical analysis across both settings.
>
> To the best of our knowledge, this is the first framework that simultaneously accommodates optimizers as diverse as STORM and Muon under a common formulation. We will revise the relevant paragraphs in the manuscript to better reflect this intent.
>
> ---
> **Q2**: Are there characteristics of settings in which variance reduction should help
>
> **A2**: Thank you for your insightful question. It is well known that deep learning training—particularly for large language models—exhibits high gradient variance (e.g., McCandlish et al., 2018; RAdam, Liu et al., 2020). MARS introduces key design changes—such as a scaled, tunable correction mechanism—that make it more effective in this setting.
>
> We believe it is these design choices, rather than the setting itself, that drive the improved performance. For a detailed discussion of the algorithmic differences, please refer to our response to Reviewer ZHwA (A3).
>
> ---
> **Q3**: The paper would benefit by bringing some of the results to the main paper and deferring details about the instantiations
>
> **A3**: Thank you for your suggestions. Due to space constraints, we were limited in the amount of experimental results we could include in the main paper. In the final version, we will restructure the content to highlight more key results in the main paper.
>
> ---
> **Q4**: Typos
>
> **A4**: Thank you for spotting these typos. We have corrected them and will ensure they are reflected in the final version.
>
> ---
> **Q5**: What was the reasoning behind the choice in not including MARS-Shampoo for the language model pretraining experiments?
>
> **A5**: The time efficiency of Shampoo is too low due to the SVD decomposition. For CIFAR100 experiments, Shampoo requires 14.75 hours while AdamW only consumes 41 minutes for 200 epochs. For GPT-2 small models, we noticed that training with Shampoo is more than 5 times slower than AdamW experiments. While we acknowledge that accelerated implementations of Shampoo could mitigate this issue, we were unable to find a reliable open-source version to use in our experiments.
>
> ---
> **Q6**: Learning rate stability of MARS-AdamW
>
> **A6**: Thank you for the question. As shown in our learning rate ablation study (Figure 16), MARS-AdamW demonstrates greater stability compared to AdamW across a range of constant learning rates. Notably, its advantage becomes more pronounced at higher learning rates. This suggests that MARS-AdamW can tolerate and benefit from a larger maximum learning rate than AdamW.
>
> ---
> **Q7**: Sweep across batch size
>
> **A7**: We conducted experiments across multiple batch sizes (240, 480, 960) as suggested. Using the difference between AdamW and MARS-AdamW as a measure of improvement, we included the loss curve of varying batch sizes in in Figure 1 in https://anonymous.4open.science/r/M-E6D7/F.pdf. As shown below, MARS-AdamW consistently outperforms AdamW, with the performance gap widening at smaller batch sizes—supporting the intuition that variance reduction is especially effective in high-variance (small-batch) settings. This conclusion also aligns with our findings in Figure 16 (initial submission) under large step sizes.
>
> | AdamW - MARS-AdamW | Train loss | Valid Loss |
> | ------------------ | ---------- | ---------- |
> | 240                | 0.0477     | 0.0359     |
> | 480                | 0.0406     | 0.0327     |
> | 960                | 0.0384     | 0.0322     |
>
> ---
> **Q8**: Compare to methods like AdEMAMix
>
> **A8**: : We included tuned AdEMAMix as an additional baseline in Figure 3 in https://anonymous.4open.science/r/M-E6D7/F.pdf. While it benefits from longer-term averaging, it performs worse than MARS in both train and validation loss.

---

### Decision · Program_Chairs · 2025-05-01

**Decision:**

Accept (poster)

**Comment:**

The paper proposes applying a scaled version of the STORM momentum that has an extra control parameter before preconditioning, and shows that this captures a wide range of optimizers with momentum in practice while achieving better results with the extra knob. The reviewers all appreciate how the paper ties together existing works on variance reduction methods, though noting a relatively small change and wondering how it changes previously negative results. Some concerns on the experimental setting remain, for instance Muon being worse than Adam can be specific to this setting given the emerging literature that contradicts this conclusion (e.g., Liu et al., 2025). It also omits certain baselines due to computational reasons (e.g., the Shampoo variant). While limitations on novelty and experiments remain, the authors find the authors' rebuttal sufficient. Following the reviewers' recommendation, I'd advise the authors to reduce the main content on repeating known optimizers and focus on the experiments to answer the more important questions raised above (e.g., what are the conditions for variance reduction to work).